# CASMIR: Coupled Adaptive Feature–Target Smoothing with Density-Gated Mixture-of-Experts for Robust Imbalanced Tabular Regression

## Abstract

Although tabular data are central to many real-world applications, their target distributions are often imbalanced, as the majority of samples correspond to a narrow range of values. This imbalance severely degrades performance in sparse, few-shot regions. Prior work on imbalanced regression has typically relied on coarse binning of the target space, discarding fine-grained information, or on spatial-locality assumptions inherited from image domains. We introduce CASMIR (Coupled Adaptive Feature–Target Smoothing with Density-Gated Mixture-of-Experts), a framework tailored to deep imbalanced tabular regression. CASMIR combines two complementary mechanisms. (i) Coupled Adaptive Smoothing first identifies "true" neighbors by jointly considering similarities in both the feature and target spaces. Based on these neighbors, it then calibrates representations in sparse regions by scaling the smoothing strength according to each sample's continuous density. (ii) A Density-Gated Mixture-of-Experts (MoE) weights the contributions of specialized experts via a gate that predicts a density range from the original features. During training, experts take the calibrated features as input; at inference, the learned experts operate on the original features, yielding both higher accuracy and inference-efficient asymmetric design. Across 40 tabular benchmarks, CASMIR establishes state-of-the-art performance on balanced test sets by attaining the top average rank. Notably, it demonstrates exceptionally robust by minimizing performance loss on the original imbalanced test sets, consistently delivering balanced predictions that enhance few-shot accuracy without significantly sacrificing many-shot performance. **The implementation code is available in the supplementary material.**

## 1 Introduction

Tabular data are central to diverse real-world applications such as industrial control, finance, and healthcare, frequently matching or exceeding the importance of vision and text in industrial domains (Borisov et al., 2024; Jiang et al., 2025). However, real-world tabular datasets frequently suffer from imbalanced targets, where predictive accuracy on rare but critical cases—such as defects or rare diseases—is essential (Cartus et al., 2023; Kim et al., 2018). Despite its growing significance, imbalanced regression on tabular data remains a relatively under-explored domain.

However, current approaches to deep imbalanced regression face several key challenges. First, state-of-the-art (SOTA) methods such as Label Distribution Smoothing (LDS) and Feature Distribution Smoothing (FDS) (Yang et al., 2021), RankSim (Gong et al., 2022), and ConR (Keramati et al., 2024) were predominantly designed for and validated on image datasets. They leverage strong priors like spatial locality and feature continuity, which are often violated in heterogeneous tabular data where features can be independent, non-linear, and non-local (Somvanshi et al., 2024). Other common strategies simplify the problem by applying coarse binning to the continuous target space, treating all samples within a "rare" bin as identical. This approach, however, discards valuable, fine-grained information about the relationships between target values, limiting predictive precision. Second, many existing methods exhibit a performance trade-off, where improvements

in few-shot regions are often achieved at the cost of degraded performance in many-shot regions (Cai et al., 2021; Kang et al., 2020). By focusing heavily on minority data, strategies like aggressive re-weighting or oversampling can disrupt the learning of majority patterns, failing to achieve the holistic performance improvement required in real-world scenarios. Finally, the validation of existing methods is often limited and inconsistent, typically reporting results on a small handful of datasets. This raises concerns about reproducibility and potential overfitting and makes it difficult to assess true real-world efficacy. A systematic and extensive validation on diverse tabular benchmarks is therefore essential.

We posit that deep imbalanced tabular regression represents a unique *blind spot* where existing powerful paradigms become ineffective. Our empirical analysis reveals that while SOTA imbalanced regression methods (e.g., LDS, ConR) struggle to learn the manifold of heterogeneous tabular data (Fig. 2), tree-based models (e.g., XGBoost) fail to perform necessary interpolation in sparse few-shot regions (Table 1). CASMIR is explicitly designed to address this intersection where these two orthogonal challenges become entangled.

To address these challenges, we present CASMIR, an architecture engineered for robust regression performance across diverse data densities. The framework is founded on two key principles: Coupled Adaptive Smoothing (CAS) to refine sparse feature representations using a learned metric and continuous density, and a Density-Gated Mixture-of-Experts (MoE) which employs an asymmetric data path to foster expert specialization. This combination effectively balances predictive accuracy across the entire target spectrum.

Our main contributions are summarized as follows:

- **Core Contribution**: To address the important yet often overlooked problem of imbalanced tabular regression, we propose CASMIR, a framework that combines a novel adaptive smoothing technique that jointly considers feature and target spaces with a density-gated Mixture-of-Experts (MoE).
- **Technical Contribution**: We introduce two key innovations: (1) a Coupled Adaptive Smoothing (CAS) module tailored for tabular data, which utilizes continuous density and learnable feature weights; and (2) a Density-Gated MoE architecture that optimizes expert learning by asymmetrically utilizing original and calibrated features.
- **Empirical Contribution**: Across an extensive benchmark of 40 diverse datasets, CASMIR outperforms state-of-the-art methods, achieving the top average rank. It demonstrates exceptional robustness by significantly enhancing few-shot performance while minimizing performance degradation in many-shot regions.

## 2 RELATED WORK

Learning from imbalanced data is a well-studied challenge, though research has predominantly focused on classification (Zhang et al., 2023). In classification, the primary goal is to learn discriminative features that separate discrete classes (Buda et al., 2018). Imbalanced regression, however, presents a unique set of challenges. The continuous nature of regression targets requires models to learn a feature space that faithfully models the rich relationships—such as proximity and order—between target values. Consequently, methods designed for classification often fail when directly applied to regression tasks (Yang et al., 2021; Gong et al., 2022). To address these unique challenges, various approaches have been proposed. Our review structures the field into three primary methodologies: data-level, algorithm-level, and hybrid approaches.

**Data-Level Approaches.** Data-level strategies, which modify the training data distribution, are fundamental to imbalanced learning, especially for tabular data. The most traditional methods are resampling strategies. These include adaptations of classification techniques, such as SMOTER (Torgo et al., 2013), which applies the SMOTE (Chawla et al., 2002) interpolation logic to regression, and oversampling with Gaussian Noise. SMOGN (Branco et al., 2017) combines these ideas to generate synthetic samples for minority regions (Branco et al., 2018). More recent generative models often employ a Variational Autoencoder (VAE), which learns a probabilistic latent space from the data to generate new, high-quality synthetic samples, thereby densifying sparse regions of the distribution. Further advancing this, some variational methods (Wang & Wang, 2023) perform

probabilistic smoothing in the latent space, where a sample's representation is calibrated by borrowing information from neighbors with similar labels to counteract data sparsity. Another popular direction is interpolation-based augmentation. (Zhang et al., 2018; Yao et al., 2022) While powerful, these methods can be complex to tune and may not always preserve the intricate feature relationships of the original data.

**Algorithm-Level Approaches.**  Instead of modifying the data, algorithm-level approaches adapt the learning process itself.  This category includes cost-sensitive learning, where higher misprediction costs (Lin et al., 2017) are assigned to sparse regions, and methods that modify the training objective (Cui et al., 2019; Ren et al., 2020).  For instance, LDS and FDS employ kernel smoothing to encourage information sharing among local neighbors. Balanced MSE (Ren et al., 2022) reformulates the standard MSE loss to restore a balanced prediction distribution. Other methods focus on feature space regularization (Verma et al., 2019; Wang et al., 2021a). RankSim enforces an ordinal correspondence between feature-space and label-space similarities, while ConR uses a contrastive objective to prevent the feature collapse of minority samples (Cui et al., 2021; Zha et al., 2023).  While effective in vision tasks, their reliance on assumed spatial locality makes them less suitable for tabular data where feature relationships are complex and non-Euclidean.  A related and crucial concept is Metric Learning, which aims to learn a distance function tailored to the data (Gautheron et al., 2019; Gui & Zhang, 2021; Zeng et al., 2022). Our proposed CAS module is directly inspired by this principle, using learnable feature weights to form an adaptive, data-driven metric that is vital for heterogeneous tabular data.

**Hybrid and Ensemble Approaches.**  Hybrid methods combine data-level and algorithm-level techniques.  Ensemble-based strategies, which train multiple models, have also proven effective. The Mixture of Experts (MoE) framework (Jiang et al., 2024; Wang et al., 2021b; Zhou et al., 2020) is a particularly powerful type of ensemble. In classification, SADE (Zhang et al., 2022) trains specialized experts and dynamically aggregates their outputs at test-time. This concept was extended to tabular regression by MATI Wang et al. (2025), which partitions data using a GMM and weights experts dynamically.

Our framework, CASMIR, can be viewed as a advanced hybrid methodology. It is distinct from existing MoE approaches, as it fosters expert specialization during the training phase via a density-gated mechanism. Most critically, CASMIR introduces a novel asymmetric dual-pathway architecture: the gating network operates on the original features, while the expert networks receive features calibrated by our CAS module, which itself integrates principles of adaptive smoothing and metric learning. This design effectively combines data-level refinement with an advanced algorithm-level architecture, representing a novel synthesis of ideas for the imbalanced tabular regression problem.

## 3 METHODOLOGY

In this study, we propose CASMIR, a novel framework designed to address the problem of imbalanced tabular regression. CASMIR integrates two complementary modules: Coupled Adaptive Smoothing (CAS), which dynamically calibrates feature representations according to data density, and a Density-Gated Mixture-of-Experts (MoE) architecture, which leverages specialized experts to effectively handle both sparse and dense regions of the data distribution.

### 3.1 PROBLEM DEFINITION

Given a tabular dataset $D = \{(x_i, y_i)\}_{i=1}^{N}$, where $x_i \in \mathbb{R}^D$ is a $D$-dimensional feature vector and $y_i \in \mathbb{R}$ is a continuous target variable, we assume that the distribution of the target, $P(y)$, is imbalanced. The objective of our work is to learn a function $f : \mathbb{R}^D \to \mathbb{R}$ that generalizes effectively to a balanced target distribution. The primary goal is to improve the model's overall robustness by lifting accuracy in sparse, few-shot regions, without incurring a severe performance trade-off in the dense, many-shot regions.

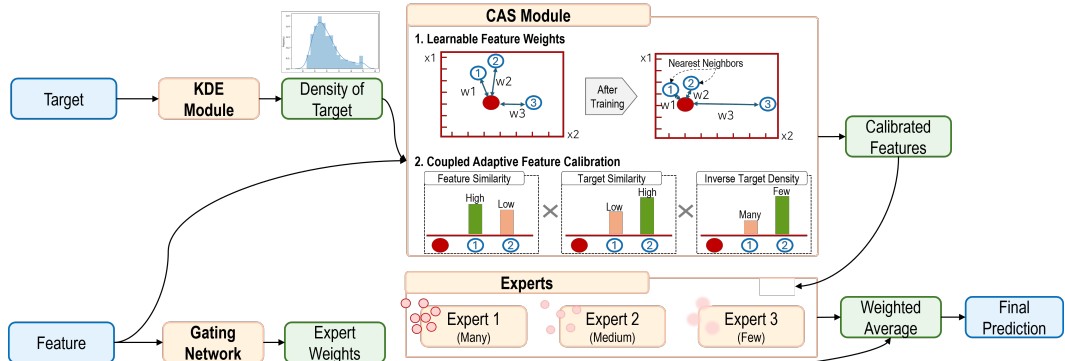

Figure 1: An overview of the proposed CASMIR architecture. The model employs an asymmetric dual-pathway design. (Gating Path) The gating network takes the original features as input to compute weights for the experts. (Expert Path) Simultaneously, the CAS module calibrates the input features. It first estimates the target density via KDE. The CAS module then performs 1. Learnable Feature Weighting to create a data-driven distance metric, followed by 2. Coupled Adaptive Feature Calibration, where features are smoothed based on neighbors selected by jointly considering feature similarity, target similarity, and inverse target density. The resulting Calibrated Features are fed into specialized experts (Many, Medium or Few-shots). The outputs of these experts are then combined in a Weighted Average using the weights from the gating network to yield the Final Prediction.

## 3.2 OVERALL ARCHITECTURE

The overall architecture of our proposed framework, CASMIR, is illustrated in Figure 1. It is composed of two main components: a Coupled Adaptive Smoothing (CAS) module for feature calibration and a Density-Gated Mixture-of-Experts (MoE) for specialized prediction. The model employs a novel dual-pathway design. For a given input, a gating network processes the original features to compute a set of weights. Simultaneously, the CAS module refines the features using density information to create a calibrated feature. This calibrated feature is then fed in parallel to all specialized expert networks. The final prediction is a weighted average of all expert outputs, using the weights from the gating network. This asymmetric architecture allows the model to intelligently combine expert predictions based on the original input signal, while each expert operates on a more stabilized and robust feature set. For a given input feature vector $x \in \mathbb{R}^D$ and its target $y \in \mathbb{R}$, we first apply a standard normalization to the input features, which we denote as $x_{\text{norm}}$. All subsequent modules in our framework then operate on this normalized feature vector.

## 3.3 COUPLED ADAPTIVE SMOOTHING (CAS)

The CAS module is a novel smoothing mechanism that adaptively calibrates the feature space according to data density. Its primary role is to stabilize volatile feature representations in sparse, low-density regions, thereby preventing poor generalization.

**Learnable Feature Weights.** To address the heterogeneity of tabular data, where features possess different scales and predictive importance, we introduce a learnable weight vector $\mathbf{w} \in \mathbb{R}^D$. This vector allows the model to learn a data-driven metric for the feature space. Unlike a standard Euclidean metric which treats all features equally, a data-driven metric learns to assign higher weights to features that are more relevant for the regression task, and lower weights to less informative or noisy features. This weight vector $\mathbf{w}$ is a set of model parameters within the CAS module, initialized and then optimized end-to-end via backpropagation. The model learns the optimal values for $\mathbf{w}$ by minimizing the overall loss function, $\mathcal{L}_{\text{total}}$. In effect, the model is guided to find a feature weighting scheme that best helps to identify "true" neighbors—those that are similar in both feature and target space. A better neighbor identification leads to more effective feature calibration ($\tilde{x}_i$), which in turn improves the experts' predictions ($\hat{y}_i$) and ultimately reduces the final regression loss. The weights are constrained to be positive via a Softplus function to ensure they behave as valid metric scaling factors.

**Coupled Adaptive Feature Calibration.** Each data sample, $x_i$, is calibrated based on its relationship with other samples within a batch.

1. *Weighted Distance Calculation*: The distance between two samples, $x_i$ and $x_j$, is computed using a weighted Euclidean metric. The squared distance is calculated as follows, where $\mathbf{w} = \{w_1, w_2, ..., w_D\}$ are the learnable, positive feature weights and $x_{i,d}$ is the $d$-th feature of sample $x_i$.

$$d(x_i, x_j)^2 = \sum_{d=1}^{D} w_d \cdot (x_{i,d} - x_{j,d})^2. \tag{1}$$

This allows the model to dynamically emphasize or de-emphasize the contribution of each feature to the distance calculation. Intuitively, $\mathbf{w}$ reshapes the feature space by expanding informative dimensions and compressing noisy ones. We provide a detailed discussion on how this acts as a data-driven metric learning mechanism in Appendix A.6.

2. *Neighbor Weight Calculation*: The influence of a neighboring sample (Ye et al., 2024), $x_j$, in the calibration of $x_i$ is determined by a weight, $\omega_{ij}$, which combines feature similarity with target-density similarity. This coupling ensures that neighbors are close on the joint feature-target manifold. This weighting is inversely proportional to the neighbor's density, ensuring that neighbors from sparser regions have a greater influence on the smoothing process.

$$\omega_{ij} \propto \exp\left(-\frac{d(x_i, x_j)^2}{2\sigma_x^2}\right) \times \exp\left(-\frac{(y_i - y_j)^2}{2\sigma_y^2}\right) \times \frac{\alpha}{d_j + \epsilon}. \tag{2}$$

Here, $\sigma_x$ and $\sigma_y$ are kernel bandwidth hyperparameters for the feature and target spaces, respectively. The hyperparameter $\alpha$ controls the strength of the inverse density weighting, where $d_j$ is the pre-computed KDE density of the neighboring sample $x_j$, and $\epsilon$ is a small constant for numerical stability. Note the distinction: $\mathbf{w}$ is a global, learnable feature weight vector, while $\omega_{ij}$ is a dynamic, pair-specific neighbor influence weight.

3. *Feature Calibration*: The final calibrated feature, $\tilde{x}_i$, is generated by blending the original feature $x_i$ with the weighted average of its neighbors, $\mu_{\text{smooth}}$.

$$\tilde{x}_i = (1 - s_i) \cdot x_i + s_i \cdot \mu_{\text{smooth}}, \quad \text{where} \quad \mu_{\text{smooth}} = \sum_{j \in \mathcal{N}(i)} \omega_{ij} x_j. \tag{3}$$

The smoothing strength, $s_i$, is inversely proportional to the density of the sample $x_i$, such that low-density samples undergo stronger regularization. This adaptive strength $s_i$ allows the model to actively navigate the bias-variance trade-off based on data density. A formal analysis of this mechanism and its connection to Vicinal Risk Minimization is detailed in Appendix A.6.

This entire procedure for CAS is detailed step-by-step in Algorithm 2 in the Appendix.

## 3.4 DENSITY-GATED MIXTURE-OF-EXPERTS (MOE)

The calibrated features from the CAS module are utilized by the expert networks within our Density-Gated MoE architecture.

**Asymmetric Input for Gating and Experts.** Our design intentionally provides asymmetric inputs to optimize distinct roles. The gating network receives the original features ($x_{\text{norm}}$) to exploit sharp decision boundaries for density classification. Conversely, the expert networks receive the CAS-calibrated features ($\tilde{x}$) to leverage stabilized representations for regression.

*Expert and Gating Networks*: Our MoE comprises a gating network and $K = 3$ specialized experts ($E_k$), each targeting a specific density region (many-, medium-, and few-shot). The gating network outputs a routing probability distribution, where the weight for expert $k$ is given by:

$$g_k(x_{\text{norm}}) = \text{Softmax}\left(\frac{\mathbf{W}_g x_{\text{norm}}}{\tau}\right)_k. \tag{4}$$

Here, $\tau$ is a learnable temperature parameter. Crucially, while experts are trained on $\tilde{x}$ to learn robust manifolds, they infer using raw $x_{\mathrm{norm}}$ to internalize the inductive bias of smoothing (detailed in Appendix A.6).

**Training Objectives.** The model is optimized by minimizing a combined objective $\mathcal{L}_{\mathrm{total}} = \mathcal{L}_{\mathrm{reg}} + \lambda_{\mathrm{aux}}\mathcal{L}_{\mathrm{aux}} + \lambda_{\mathrm{load}}\mathcal{L}_{\mathrm{load}}$. The primary regression loss, $\mathcal{L}_{\mathrm{reg}}$, is the Mean Absolute Error (MAE) between the final prediction $\hat{y}_i$ and the ground truth $y_i$:

$$\mathcal{L}_{\mathrm{reg}} = \frac{1}{N_b} \sum_{i=1}^{N_b} |\hat{y}_i - y_i|. \tag{5}$$

Given that we train on an *imbalanced* dataset while targeting *balanced* generalization, we introduce two auxiliary losses that play complementary roles for expert *specialization* and *survival*.

First, to enforce explicit specialization, we impose an **auxiliary supervision loss** $\mathcal{L}_{\mathrm{aux}}$. Unlike standard MoEs that rely on unsupervised routing, we use the pre-computed density label $c_i \in \{0, 1, 2\}$ as ground truth to guide the gate:

$$\mathcal{L}_{\mathrm{aux}} = -\frac{1}{N_b} \sum_{i=1}^{N_b} \sum_{k=1}^{K} \mathbb{I}(c_i = k) \log(g_k(x_{i,\mathrm{norm}})). \tag{6}$$

This explicitly supervises the gating network to guarantee correct semantic routing regardless of data frequency.

Second, to prevent "Expert Collapse" where majority-biased experts dominate the gradient, we employ a **load-balancing loss** $\mathcal{L}_{\mathrm{load}}$ (Shazeer et al., 2017):

$$\mathcal{L}_{\mathrm{load}} = (K - 1) \sum_{k=1}^{K} f_k P_k, \quad \text{where} \quad f_k = P_k = \frac{1}{N_b} \sum_{i=1}^{N_b} g_k(x_{i,\mathrm{norm}}). \tag{7}$$

Here, $f_k$ represents the average gating probability for expert $k$ over the batch. This loss acts as a regularizer to ensure the survival of few-shot experts during early training rather than forcing a mechanical balanced split. To prioritize semantic specialization ($\mathcal{L}_{\mathrm{aux}}$), we constrain $\lambda_{\mathrm{load}}$ to a small range ($0.01 \sim 0.2$).

### 3.5 Training and Inference Procedure

The training and inference procedures in CASMIR are intentionally asymmetric to maximize both learning capacity and deployment efficiency. During training, the model leverages the full architecture. The CAS module generates calibrated features for the experts, fostering specialization, while the gating network is guided by the auxiliary and load-balancing losses in addition to the main regression objective. This allows for a rich, density-aware representation learning process. During inference, however, the CAS module and the auxiliary loss calculations are deactivated. The final prediction, $\hat{y}$, is computed deterministically by passing the original (normalized) features through the trained gating and expert networks:

$$\hat{y} = \sum_{k=1}^{K} g_k(x_{\mathrm{norm}}) \cdot E_k(x_{\mathrm{norm}}). \tag{8}$$

This asymmetric design ensures that inference is fast and efficient, as the computationally intensive smoothing process is confined to the training stage. The detailed step-by-step training procedure is presented in Algorithm 1 in the Appendix. It allows the model to benefit from deep regularization during training without incurring additional overhead at test time, making it practical for real-world deployment.

## 4 Experiments

In this section, we describe the design and results of extensive experiments conducted to validate the effectiveness of our proposed methodology, CASMIR. We compare the performance of CASMIR

against multiple baselines, including state-of-the-art (SOTA) methods, across a large number of tabular datasets with diverse characteristics. Furthermore, we conduct an ablation study to analyze the impact of each component of CASMIR on its overall performance.

## 4.1 EXPERIMENTAL SETUP

**Datasets.** To evaluate the generalization performance and robustness of our model on the imbalanced tabular regression problem, we collected and utilized *40 public datasets* from well-known sources, including the UCI Machine Learning Repository, OpenML, Scikit-Learn, and the RE-BAGG benchmark. These datasets encompass a wide variety of characteristics in terms of the number of samples, feature dimensions, and degrees of imbalance, covering diverse domains such as 'grid_stability', 'forest_fires', 'Abalone', and 'superconductivity'.

For the continuous target variable (y) in each dataset, we analyzed its distribution and partitioned it into three discrete bins: *Few-shot* (bins with a sample count less than or equal to 'few threshold'), *Medium-shot* (count between 'few threshold' and 'many threshold'), and *Many-shot* (count greater than 'many threshold'). This binning is used for a fine-grained evaluation of model performance according to data density. To supervise the gating network, we similarly categorize the training data into three target density regions (Few, Medium, Many) based on the 33rd and 66th percentiles of the target density distribution. Detailed statistics for all 40 datasets are provided in Appendix A.1.

**Baselines.** To assess the performance of CASMIR, we conducted a comparative analysis against 11 strong baseline models, consisting of traditional tree-based models and recent deep learning-based imbalanced regression methods. While more complex tabular architectures exist (Arik & Pfister, 2021; Chen et al., 2024; Gorishniy et al., 2021; Holzmüller et al., 2024), we utilize a standard MLP for all deep learning baselines to objectively evaluate the contribution of the imbalanced learning frameworks themselves, rather than the backbone's capacity. We compared CASMIR against 13 strong baselines, including tree-based models (e.g., XGBoost), their oversampled variants (e.g., XG-Boost+SMOGN), a Vanilla MLP, and 8 state-of-the-art deep imbalanced regression methods (e.g., LDS, ConR). To ensure a fair comparison, all deep learning baseline methods and CASMIR shared the same standard MLP backbone. A complete list and further details are provided in Appendix A.2.

**Evaluation Process.** For a fair comparison, all algorithms underwent hyperparameter optimization (HPO) using *Optuna* with 50 trials for each of the 40 datasets. Neural network models were trained for a maximum of 300 epochs with an early stopping patience of 70. The models were trained on the original imbalanced training set, while a balanced set was used as the validation set for HPO. The final performance of the models was evaluated in two scenarios:

- *Balanced Test Set*: Performance was measured based on Mean Absolute Error (MAE) across the All, Few-shot, Medium-shot, and Many-shot regions. The *Average Ranking* across the 40 datasets was calculated to assess overall performance superiority without overfitting to specific datasets.
- *Original Imbalanced Test Set*: To investigate whether the imbalanced learning algorithms excessively sacrifice performance in many-shot regions (performance trade-off), we also evaluated and ranked their performance on a test set that follows the original imbalanced distribution.

The hyperparameter search spaces and final configurations for all models are detailed in Appendix A.1.

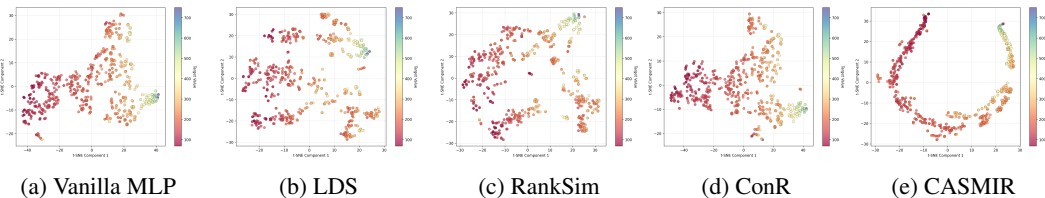

|               |             |              |           |             |
| :-----------: | :---------: | :----------: | :-------: | :---------: |
| (a) Vanilla MLP | (b) LDS   | (c) RankSim  | (d) ConR  | (e) CASMIR  |

Figure 2: A t-SNE visualization comparing learned feature representations on a maxTorque tabular dataset. The visualization displays the final hidden layer's feature representations for four models: (a) **Vanilla MLP**, (b) **LDS**, (c) **RankSim**, (d) **ConR**, and (e) **our proposed model, CASMIR**. Each point represents a sample from the test set, with its color determined by the continuous target value.

Table 1: Results for bike_sharing (Ranked by Overall MAE)

| Algorithm | MAE (Balanced) | | | | MAE (Original Imbalanced) | | | |
|---|---|---|---|---|---|---|---|---|
| | All | Many | Med | Few | All | Many | Med | Few |
| LightGBM | 149.85 | 105.51 | 68.08 | 275.95 | 86.59 | 78.82 | 57.16 | 275.95 |
| CatBoost | 128.41 | 123.32 | 79.27 | 182.64 | 104.76 | 101.89 | 88.86 | 182.64 |
| XGBoost | 118.97 | 127.79 | 50.33 | 178.79 | 88.54 | 86.58 | 54.08 | 178.79 |
| XGBoost+GN | 124.18 | 109.37 | 58.88 | 204.29 | 94.93 | 92.31 | 56.09 | 204.29 |
| XGBoost+SMOTER | 121.98 | 115.34 | 58.98 | 191.61 | 101.60 | 99.74 | 66.13 | 191.61 |
| Vanilla MLP | 209.91 | 128.18 | 156.85 | 344.68 | 138.29 | 126.09 | 150.01 | 344.68 |
| Simple Ensemble | 477.89 | 206.11 | 382.74 | 844.83 | 273.43 | 233.96 | 372.71 | 844.83 |
| MLP+SQRT_INV | 199.21 | 122.64 | 176.35 | 298.65 | 162.70 | 155.21 | 163.96 | 298.65 |
| MLP+LDS | 234.11 | 127.23 | 182.36 | 392.75 | 143.06 | 127.42 | 167.53 | 392.75 |
| MLP+RankSim | 211.25 | 115.26 | 187.84 | 330.65 | 130.75 | 116.18 | 174.44 | 330.65 |
| MLP+BMSE(GAI) | 208.12 | 144.00 | 174.97 | 305.38 | 133.10 | 119.95 | 177.60 | 305.38 |
| MLP+BMSE(BMC) | 144.78 | 123.09 | 143.77 | 167.47 | 103.36 | 95.49 | 154.89 | 167.47 |
| MLP+ConR | 208.50 | 119.34 | 165.24 | 340.92 | 134.73 | 120.55 | 169.79 | 340.92 |
| CASMIR | **73.81** | **26.45** | **47.06** | **147.93** | **35.35** | **27.90** | **51.05** | **147.93** |

## 4.2 RESULTS

Table 2 summarizes the average MAE rankings on the balanced test sets across the 40 datasets. Figure 3 plots the rank distribution for each algorithm, illustrating both median performance and consistency (see Appendix Table 12 for algorithm abbreviations). CASMIR exhibits the best median rank and a tight distribution, confirming its consistently strong performance across the datasets. Our proposed method, CASMIR, achieved the top average rank, outperforming all baseline models. This suggests that CASMIR demonstrates consistently superior generalization performance across diverse tabular data environments, not limited to a specific data domain or type of imbalance. Notably, CASMIR's superiority on balanced sets stems from its significant improvements in few-shot regions, without introducing the performance trade-off often seen in other methods. This is evidenced by its robust performance and minimal degradation on the original imbalanced test sets, highlighting its ability to learn from all data densities effectively.

Furthermore, CASMIR demonstrated exceptional robustness on the original imbalanced test sets. While tree-based baselines like XGBoost and CatBoost achieve higher rankings on the Original Test Set (Table 1), this performance is largely driven by their strong bias towards majority samples (Majority Bias). This is particularly notable in contrast to the common performance trade-off seen in many imbalanced learning algorithms, where improving few-shot accuracy comes at the cost of many-shot performance. CASMIR effectively mitigates this issue, maintaining a highly competitive rank by minimizing performance degradation in many-shot regions. This indicates its potential as a practical solution for reliable real-world applications. To provide a more detailed view, Tables 1 presents the results on the 'bike_sharing' dataset respectively. On 'bike_sharing', CASMIR reduces the Balanced MAE by approximately 65% and the Original MAE by over 74% compared to the Vanilla MLP. The complete performance tables for all 40 benchmarks are available in Appendix A.9.

Figure 2 qualitatively visualizes the learned feature representations via t-SNE. The (a) Vanilla MLP exhibits severe feature collapse, with samples of different target values (colors) being heavily mixed. While strong baselines like (b) LDS, (c) RankSim and (d) ConR offer some improvement, their representations remain fragmented. In contrast, (e) CASMIR learns a coherent and continuous manifold where target values are smoothly and semantically ordered, providing strong visual evidence for the regularization effect of our CAS module.

## 4.3 ABLATION STUDY

To rigorously disentangle the complexity of the CASMIR framework and isolate the contribution of each component, we conducted a detailed ablation study. We compared the **Full CASMIR** model against seven variants: (1) **MoE-Only** (standard MoE without CAS); (2) **MixUp-MoE** (MoE with standard MixUp augmentation); (3) **CAS-Only** (Single MLP with CAS); (4) **CAS-Feature-Only**

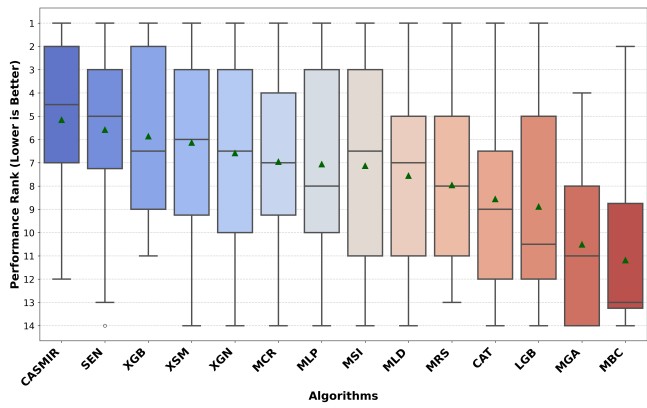

Figure 3: Algorithm Ranking on Balanced Datasets.

Table 2: Average Algorithm Ranking.

| Algorithm | Balanced Rank | Original Rank |
|---|---|---|
| CASMIR | **5.2** | 6.1 |
| Simple Ensemble | 5.6 | 5.3 |
| XGBoost | 5.8 | **4.5** |
| XGBoost+SMOTER | 6.1 | 7.9 |
| XGBoost+GN | 6.6 | 8.4 |
| MLP+ConR | 7.0 | 7.6 |
| Vanilla MLP | 7.0 | 7.2 |
| MLP+SQRT_INV | 7.1 | 10.4 |
| MLP+LDS | 7.5 | 7.5 |
| MLP+RankSim | 8.0 | 8.4 |
| CatBoost | 8.6 | 4.6 |
| LightGBM | 8.9 | 4.8 |
| MLP+BMSE(GAI) | 10.5 | 10.8 |
| MLP+BMSE(BMC) | 11.2 | 11.5 |

(Neighbor selection excluding target similarity); (5) **CAS-No-Learnable-Metric** (Using standard Euclidean distance); and (6–7) **Fixed Smoothness** (fixing $s_i = 0$ and $s_i = 1$) to validate adaptive regularization.

Table 3: Detailed ablation results on maxTorque benchmark

| Methods/Shots | MAE (Balanced) | | | | MAE (Original Imbalanced) | | | |
|---|---|---|---|---|---|---|---|---|
| | **All** | **Many** | **Med** | **Few** | **All** | **Many** | **Med** | **Few** |
| MoE-Only | 42.42 | 5.02 | 15.32 | 106.92 | 17.90 | 5.03 | 11.42 | 106.92 |
| MixUp-MoE | 44.15 | 4.86 | 13.84 | 113.74 | 18.42 | 5.04 | 10.65 | 113.74 |
| CAS-Only | 29.33 | 6.02 | 8.89 | 73.07 | 14.04 | 6.50 | 7.69 | 73.07 |
| CAS-Feature-Only | 35.61 | 17.17 | 32.62 | **57.03** | 24.47 | 13.90 | 34.23 | **57.03** |
| CAS-No-Learnable-Metric | 28.38 | 5.76 | 8.58 | 70.79 | 13.56 | 6.09 | 7.76 | 70.79 |
| Fixed Smoothness ($s_i = 0$) | 40.76 | 5.04 | 12.92 | 104.32 | 16.95 | 4.73 | 9.78 | 104.32 |
| Fixed Smoothness ($s_i = 1$) | 34.81 | 5.03 | 12.48 | 86.91 | 15.64 | 5.15 | 10.83 | 86.91 |
| **CASMIR (Ours)** | **25.39** | **4.79** | **6.64** | 64.74 | **11.74** | **4.71** | **6.58** | 64.74 |

The results on the maxTorque dataset are presented in Table 3. First, **CASMIR** outperforms both **MoE-Only** and **CAS-Only**, confirming that the combination of adaptive smoothing and expert specialization yields a significant synergistic effect. Notably, CASMIR surpasses **MixUp-MoE**, demonstrating that our proposed smoothing is more effective for imbalanced regression than generic augmentation. Second, the inferior performance of **CAS-Feature-Only** and **CAS-No-Learnable-Metric** highlights the necessity of the target-coupled mechanism and the learnable metric for handling tabular heterogeneity. Finally, regarding density adaptation, **Fixed Smoothness** ($s_i = 0$) suffered from overfitting in few-shot regions, while **Fixed Smoothness** ($s_i = 1$) lagged overall. CASMIR achieved the best balance across all densities, proving that dynamically navigating the bias-variance trade-off is crucial for robust performance.

Furthermore, to verify that our model performs context-aware dynamic routing rather than static averaging, we conducted a quantitative analysis of expert weights. The results confirm that CASMIR acts as a specialist via "Selection" in sparse regions and as a generalist via "Collaboration" in dense regions. Detailed routing patterns and statistical evidence are provided in Appendix A.10.

## 5 CONCLUSION

In this paper, we addressed the critical and under-explored challenge of imbalanced tabular regression. We proposed CASMIR, a novel framework that synergistically combines two components: Coupled Adaptive Smoothing (CAS), which calibrates features using joint feature-target similarity and continuous density, and a Density-Gated Mixture-of-Experts (MoE) that leverages an asymmetric design for optimal expert specialization.

Extensive experiments on 40 diverse tabular benchmarks demonstrate that CASMIR establishes a new state-of-the-art, achieving the top average rank on balanced test sets. Critically, it improves few-shot accuracy without the common performance trade-off in many-shot regions, demonstrating its exceptional robustness. Our work thus provides a principled and effective solution for real-world applications where predicting rare but critical outcomes is essential. Future work includes applying the CAS module to other modalities and exploring more advanced gating mechanisms.

ACKNOWLEDGMENTS

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

# A  APPENDIX

## A.1  DATASET DETAILS

This section provides detailed statistics for the 40 datasets used in our experiments. Table 4 outlines the fundamental properties of each dataset, while Tables 5 and 6 detail the distribution of samples across many-, medium-, and few-shot regions for the validation and test sets, respectively. The training set for each experiment consists of the remaining samples and fully retains the original imbalanced distribution.

In Table 4, **IR** (Imbalance Ratio) is the ratio of the number of samples in the most frequent bin to the least frequent bin, indicating the degree of imbalance. **Skew** and **Kurt** refer to the skewness and kurtosis of the target distribution. **Miss%** denotes the percentage of missing values in the dataset, while **Max Corr** and **Mean Corr** represent the maximum and mean absolute correlation between features, respectively.

The shot regions (many-, medium-, few-) were defined manually for each dataset to ensure a meaningful categorization. After splitting the data into training, validation, and test sets, we visually inspected the target distributions of the validation and test portions. We then selected appropriate thresholds to define the few-shot and many-shot boundaries on a per-dataset basis. Table 5 and Table 6 show the resulting distributions.

### HYPERPARAMETER SETTINGS

This section details the hyperparameter optimization process for all 14 algorithms across the 40 datasets. Our goal was to ensure that every method, including our own, was tuned to its optimal potential for a fair and rigorous comparison.

We comprehensively optimized hyperparameters for all 14 algorithms across 40 datasets. Using the Optuna framework with a TPE sampler, we ran 50 trials for each algorithm-dataset pair. The objective for each trial was to minimize the validation loss, averaged over three independent runs using different random seeds. The complete search space for every tuned parameter is detailed in Table 7, 9, 10.

- **Tree-Based Models (XGBoost, LightGBM):** We tuned all key parameters governing model structure (e.g., n_estimators, max_depth), the learning process (learning_rate), and regularization (subsample, colsample_bytree) to find the optimal balance between performance and complexity.
- **Samplers (SMOTER, GaussianNoise):** The sampling methods were optimized by tuning their relevance function (method, percentile), resampling strategy, and other specific settings like SMOTER's k_neighbors and the noise intensity for GaussianNoise.
- **MLP-Based Models:** All MLP variants were built upon a common search space for network architecture, optimization, and regularization. Additionally, the unique parameters of each specific method were fully optimized. This includes the specialized loss functions for MLP+RankSim, MLP+ConR, MLP+BMSE(GAI), and MLP+BMSE(BMC), as well as the distribution smoothing mechanisms for MLP+LDS.
- **CASMIR (Ours):** Our proposed model required tuning its two primary components. For the MoE architecture, we optimized the dimensions of expert and gate networks and the weights for the auxiliary and load-balancing losses. For the CAS module, we tuned all parameters that control its density-aware smoothing, including those for density estimation, feature and label bandwidths, and smoothing intensity.

## A.2  BACKBONE ARCHITECTURE DETAILS

To ensure a fair comparison, all deep learning baseline methods and the experts within our proposed CASMIR framework shared a standardized MLP backbone. This was not a vanilla MLP, but rather a modern architecture optimized for tabular data, inspired by recent advancements in deep learning for this domain.

Table 4: Dataset Basic Statistics

| Dataset | N Sample | N Features (X) | N Target (Y) | N Num. | Skew | Kurt | IR | Miss % | Max Corr | Mean Corr |
|---|---|---|---|---|---|---|---|---|---|---|
| Abalone | 4177 | 8 | 1 | 7 | 1.11 | 2.33 | 689.00 | 0.0 | 0.987 | 0.891 |
| airfoild | 1503 | 5 | 1 | 5 | -0.42 | -0.31 | 39.50 | 0.0 | 0.753 | 0.219 |
| availPwr | 1802 | 15 | 1 | 8 | 1.91 | 5.12 | 230.00 | 0.0 | 0.909 | 0.209 |
| bank32nh | 8192 | 32 | 1 | 32 | 1.97 | 3.76 | 4509.00 | 0.0 | 0.437 | 0.012 |
| bank8FM | 4499 | 8 | 1 | 8 | 1.08 | 0.62 | 973.00 | 0.0 | 0.029 | 0.011 |
| bike_sharing | 731 | 15 | 1 | 14 | -0.05 | -0.81 | 10.54 | 0.0 | 0.992 | 0.182 |
| combined_cycle_power_plant | 9568 | 4 | 1 | 4 | 0.31 | -1.05 | 328.00 | 0.0 | 0.844 | 0.453 |
| communities_crime | 1994 | 127 | 1 | 126 | 1.52 | 1.83 | 64.20 | 15.4 | 1.000 | 0.209 |
| concreteStrength | 1030 | 8 | 1 | 8 | 0.42 | -0.31 | 12.78 | 0.0 | 0.658 | 0.207 |
| cps88wages | 28155 | 6 | 1 | 2 | 6.50 | 165.16 | 13811.00 | 0.0 | 0.287 | 0.287 |
| cpuSm | 8192 | 12 | 1 | 12 | -3.42 | 12.73 | 907.00 | 0.0 | 0.881 | 0.231 |
| dAiler | 7129 | 5 | 1 | 5 | 0.29 | 2.92 | 1564.00 | 0.0 | 0.639 | 0.188 |
| diabetes | 442 | 10 | 1 | 10 | 0.44 | -0.88 | 13.33 | 0.0 | 0.897 | 0.322 |
| diamond_regression | 53940 | 9 | 1 | 6 | 1.62 | 2.18 | 90.70 | 0.0 | 0.975 | 0.464 |
| ecoli70 | 2000 | 45 | 1 | 45 | 0.03 | 0.15 | 178.00 | 0.0 | 0.978 | 0.215 |
| energy_efficiency | 768 | 9 | 1 | 9 | 0.36 | -1.25 | 9.35 | 0.0 | 0.992 | 0.288 |
| forest_fires | 517 | 12 | 1 | 10 | 12.85 | 194.14 | 429.00 | 0.0 | 0.682 | 0.173 |
| fps_benchmark | 24624 | 43 | 1 | 29 | 1.05 | 1.42 | 1035.00 | 6.4 | 1.000 | nan |
| fuelCons | 1764 | 37 | 1 | 25 | 1.14 | 2.90 | 194.00 | 0.0 | 0.908 | 0.127 |
| geographical_origin_of_music | 1059 | 116 | 1 | 116 | -0.99 | 0.70 | 25.07 | 0.0 | 1.000 | 0.266 |
| grid_stability_regression | 10000 | 12 | 1 | 12 | 0.02 | -0.91 | 546.00 | 0.0 | 0.585 | 0.036 |
| kin8nm | 8192 | 8 | 1 | 8 | 0.09 | -0.53 | 63.43 | 0.0 | 0.021 | 0.009 |
| kings_county | 21613 | 21 | 1 | 17 | 4.02 | 34.59 | 8630.00 | 0.0 | 0.877 | 0.203 |
| machineCpu | 209 | 6 | 1 | 6 | 3.89 | 19.25 | 160.00 | 0.0 | 0.758 | 0.461 |
| magic_irri | 2000 | 63 | 1 | 63 | -0.01 | -0.04 | 181.00 | 0.0 | 0.928 | 0.033 |
| maxTorque | 1802 | 32 | 1 | 19 | 1.64 | 4.55 | 234.00 | 0.0 | 0.995 | 0.172 |
| miami_housing_regression | 13932 | 15 | 1 | 15 | 3.22 | 13.28 | 842.00 | 0.0 | 0.792 | 0.192 |
| Moneyball | 1232 | 14 | 1 | 8 | 0.17 | -0.02 | 65.00 | 19.5 | 0.910 | 0.451 |
| nhanes_age | 2278 | 7 | 1 | 4 | 0.24 | -1.08 | 7.59 | 0.0 | 0.686 | 0.345 |
| online_news_popularity | 39644 | 60 | 1 | 59 | 33.96 | 1832.67 | 30014.00 | 0.0 | 1.000 | 0.070 |
| parkinsons_telemonitoring | 5875 | 21 | 1 | 21 | 0.08 | -0.94 | 65.67 | 0.0 | 1.000 | 0.411 |
| pumadyn32nh | 8192 | 32 | 1 | 32 | -0.05 | -0.04 | 669.00 | 0.0 | 0.034 | 0.009 |
| qsar_aquatic_toxicity | 546 | 8 | 1 | 8 | 0.32 | 0.45 | 39.50 | 0.0 | 0.860 | 0.374 |
| red_wine | 1599 | 11 | 1 | 11 | 0.22 | 0.30 | 68.10 | 0.0 | 0.683 | 0.200 |
| servo | 167 | 4 | 1 | 2 | 1.79 | 2.07 | 77.00 | 0.0 | 0.812 | 0.812 |
| socmob | 1156 | 5 | 1 | 1 | 4.43 | 25.10 | 531.00 | 0.0 | 0.000 | 0.000 |
| solar_flare | 1066 | 10 | 1 | 2 | 3.95 | 19.62 | 996.00 | 0.0 | 0.000 | nan |
| space_ga | 3107 | 6 | 1 | 6 | -1.02 | 8.43 | 229.00 | 0.0 | 0.994 | 0.482 |
| superconductivity | 21263 | 81 | 1 | 81 | 0.86 | -0.53 | 4339.00 | 0.0 | 0.998 | 0.345 |
| white_wine | 4898 | 11 | 1 | 11 | 0.16 | 0.22 | 439.60 | 0.0 | 0.839 | 0.178 |

Table 5: Validation Set Shot Distribution

| Dataset | N Samples | Few Samples | Med Samples | Many Samples | Few% | Med% | Many% |
|---|---|---|---|---|---|---|---|
| Abalone | 627 | 211 | 217 | 199 | 33.7 | 34.6 | 31.7 |
| airfoild | 226 | 14 | 52 | 160 | 6.2 | 23.0 | 70.8 |
| availPwr | 271 | 31 | 47 | 193 | 11.4 | 17.3 | 71.2 |
| bank32nh | 1229 | 96 | 149 | 984 | 7.8 | 12.1 | 80.1 |
| bank8FM | 675 | 95 | 140 | 440 | 14.1 | 20.7 | 65.2 |
| bike_sharing | 110 | 16 | 43 | 51 | 14.5 | 39.1 | 46.4 |
| combined_cycle_power_plant | 1436 | 342 | 517 | 577 | 23.8 | 36.0 | 40.2 |
| communities_crime | 300 | 68 | 81 | 151 | 22.7 | 27.0 | 50.3 |
| concreteStrength | 155 | 14 | 57 | 84 | 9.0 | 36.8 | 54.2 |
| cps88wages | 4224 | 374 | 1140 | 2710 | 8.9 | 27.0 | 64.2 |
| cpuSm | 1229 | 124 | 227 | 878 | 10.1 | 18.5 | 71.4 |
| dAiler | 1070 | 67 | 276 | 727 | 6.3 | 25.8 | 67.9 |
| diabetes | 67 | 4 | 39 | 24 | 6.0 | 58.2 | 35.8 |
| diamond_regression | 8091 | 1091 | 2533 | 4467 | 13.5 | 31.3 | 55.2 |
| ecoli70 | 300 | 31 | 70 | 199 | 10.3 | 23.3 | 66.3 |
| energy_efficiency | 116 | 27 | 35 | 54 | 23.3 | 30.2 | 46.6 |
| forest_fires | 78 | 12 | 21 | 45 | 15.4 | 26.9 | 57.7 |
| fps_benchmark | 3694 | 345 | 690 | 2659 | 9.3 | 18.7 | 72.0 |
| fuelCons | 265 | 15 | 97 | 153 | 5.7 | 36.6 | 57.7 |
| geographical_origin_of_music | 159 | 22 | 84 | 53 | 13.8 | 52.8 | 33.3 |
| grid_stability_regression | 1500 | 74 | 557 | 869 | 4.9 | 37.1 | 57.9 |
| kin8nm | 1229 | 81 | 387 | 761 | 6.6 | 31.5 | 61.9 |
| kings_county | 3242 | 698 | 978 | 1566 | 21.5 | 30.2 | 48.3 |
| machineCpu | 32 | 6 | 7 | 19 | 18.8 | 21.9 | 59.4 |
| magic_irri | 300 | 58 | 89 | 153 | 19.3 | 29.7 | 51.0 |
| maxTorque | 271 | 32 | 103 | 136 | 11.8 | 38.0 | 50.2 |
| miami_housing_regression | 2090 | 294 | 564 | 1232 | 14.1 | 27.0 | 58.9 |
| Moneyball | 185 | 25 | 44 | 116 | 13.5 | 23.8 | 62.7 |
| nhanes_age | 342 | 68 | 119 | 155 | 19.9 | 34.8 | 45.3 |
| online_news_popularity | 5947 | 227 | 653 | 5067 | 3.8 | 11.0 | 85.2 |
| parkinsons_telemonitoring | 882 | 39 | 366 | 477 | 4.4 | 41.5 | 54.1 |
| pumadyn32nh | 1229 | 153 | 431 | 645 | 12.4 | 35.1 | 52.5 |
| qsar_aquatic_toxicity | 82 | 18 | 23 | 41 | 22.0 | 28.0 | 50.0 |
| red_wine | 240 | 10 | 33 | 197 | 4.2 | 13.8 | 82.1 |
| servo | 25 | 7 | 6 | 12 | 28.0 | 24.0 | 48.0 |
| socmob | 174 | 41 | 38 | 95 | 23.6 | 21.8 | 54.6 |
| solar_flare | 160 | 11 | 18 | 131 | 6.9 | 11.2 | 81.9 |
| space_ga | 466 | 66 | 105 | 295 | 14.2 | 22.5 | 63.3 |
| superconductivity | 3190 | 582 | 824 | 1784 | 18.2 | 25.8 | 55.9 |
| white_wine | 735 | 61 | 132 | 542 | 8.3 | 18.0 | 73.7 |

Table 6: Test Set Shot Distribution

| Dataset | N Samples | Few Samples | Med Samples | Many Samples | Few% | Med% | Many% |
|---|---|---|---|---|---|---|---|
| Abalone | 836 | 177 | 183 | 476 | 21.2 | 21.9 | 56.9 |
| airfoild | 301 | 12 | 34 | 255 | 4.0 | 11.3 | 84.7 |
| availPwr | 361 | 38 | 43 | 280 | 10.5 | 11.9 | 77.6 |
| bank32nh | 1639 | 75 | 189 | 1375 | 4.6 | 11.5 | 83.9 |
| bank8FM | 900 | 108 | 98 | 694 | 12.0 | 10.9 | 77.1 |
| bike_sharing | 147 | 7 | 11 | 129 | 4.8 | 7.5 | 87.8 |
| combined_cycle_power_plant | 1914 | 175 | 359 | 1380 | 9.1 | 18.8 | 72.1 |
| communities_crime | 399 | 66 | 133 | 200 | 16.5 | 33.3 | 50.1 |
| concreteStrength | 206 | 14 | 34 | 158 | 6.8 | 16.5 | 76.7 |
| cps88wages | 5631 | 313 | 467 | 4851 | 5.6 | 8.3 | 86.1 |
| cpuSm | 1639 | 101 | 256 | 1282 | 6.2 | 15.6 | 78.2 |
| dAiler | 1426 | 89 | 265 | 1072 | 6.2 | 18.6 | 75.2 |
| diabetes | 89 | 4 | 42 | 43 | 4.5 | 47.2 | 48.3 |
| diamond_regression | 10788 | 1114 | 2416 | 7258 | 10.3 | 22.4 | 67.3 |
| ecoli70 | 400 | 26 | 91 | 283 | 6.5 | 22.8 | 70.8 |
| energy_efficiency | 154 | 24 | 61 | 69 | 15.6 | 39.6 | 44.8 |
| forest_fires | 104 | 17 | 22 | 65 | 16.3 | 21.2 | 62.5 |
| fps_benchmark | 4925 | 355 | 714 | 3856 | 7.2 | 14.5 | 78.3 |
| fuelCons | 353 | 24 | 72 | 257 | 6.8 | 20.4 | 72.8 |
| geographical_origin_of_music | 212 | 32 | 48 | 132 | 15.1 | 22.6 | 62.3 |
| grid_stability_regression | 2000 | 96 | 180 | 1724 | 4.8 | 9.0 | 86.2 |
| kin8nm | 1639 | 72 | 257 | 1310 | 4.4 | 15.7 | 79.9 |
| kings_county | 4323 | 691 | 578 | 3054 | 16.0 | 13.4 | 70.6 |
| machineCpu | 42 | 6 | 13 | 23 | 14.3 | 31.0 | 54.8 |
| magic_irri | 400 | 48 | 88 | 264 | 12.0 | 22.0 | 66.0 |
| maxTorque | 361 | 37 | 107 | 217 | 10.2 | 29.6 | 60.1 |
| miami_housing_regression | 2787 | 353 | 580 | 1854 | 12.7 | 20.8 | 66.5 |
| Moneyball | 247 | 22 | 32 | 193 | 8.9 | 13.0 | 78.1 |
| nhanes_age | 456 | 23 | 110 | 323 | 5.0 | 24.1 | 70.8 |
| online_news_popularity | 7929 | 297 | 875 | 6757 | 3.7 | 11.0 | 85.2 |
| parkinsons_telemonitoring | 1175 | 39 | 210 | 926 | 3.3 | 17.9 | 78.8 |
| pumadyn32nh | 1639 | 103 | 449 | 1087 | 6.3 | 27.4 | 66.3 |
| qsar_aquatic_toxicity | 110 | 17 | 21 | 72 | 15.5 | 19.1 | 65.5 |
| red_wine | 320 | 17 | 43 | 260 | 5.3 | 13.4 | 81.2 |
| servo | 34 | 2 | 15 | 17 | 5.9 | 44.1 | 50.0 |
| socmob | 232 | 44 | 66 | 122 | 19.0 | 28.4 | 52.6 |
| solar_flare | 214 | 14 | 17 | 183 | 6.5 | 7.9 | 85.5 |
| space_ga | 622 | 60 | 75 | 487 | 9.6 | 12.1 | 78.3 |
| superconductivity | 4253 | 578 | 734 | 2941 | 13.6 | 17.3 | 69.2 |
| white_wine | 980 | 66 | 176 | 738 | 6.7 | 18.0 | 75.3 |

Table 7: Hyperparameter Search Space for Tree-Based Algorithms

| Algorithm | Hyperparameter | Type | Search Space / Distribution |
|---|---|---|---|
| XGBoost | max_depth | Integer | 3 to 10 |
| | learning_rate | Real | Log-uniform ($10^{-4}$ to $10^{-1}$) |
| | n_estimators | Integer | 50 to 300 |
| | min_child_weight | Integer | 1 to 7 |
| | subsample | Real | Uniform (0.6 to 1.0) |
| | colsample_bytree | Real | Uniform (0.6 to 1.0) |
| | gamma | Real | Log-uniform ($10^{-8}$ to 1.0) |
| | early_stopping_rounds | Integer | 5 to 30 |
| LightGBM | num_leaves | Integer | 20 to 100 |
| | learning_rate | Real | Log-uniform ($10^{-4}$ to $10^{-1}$) |
| | n_estimators | Integer | 50 to 300 |
| | min_child_samples | Integer | 5 to 100 |
| | subsample | Real | Uniform (0.6 to 1.0) |
| | colsample_bytree | Real | Uniform (0.6 to 1.0) |
| | early_stopping_rounds | Integer | 5 to 30 |
| SMOTER | relevance_method | Categorical | sigmoid, pdf |
| | relevance_percentile | Real | Uniform (60 to 90) |
| | resampling_strategy | Categorical | balance, average, extreme |
| | sigmoid_cl | Real | 10th to 45th percentile of y_train |
| | sigmoid_ch | Real | 55th to 90th percentile of y_train |
| | pdf_bandwidth | Real | Log-uniform (0.1 to 2.0) |
| | k_neighbors | Integer | 3 to 10 |
| GaussianNoise | relevance_method | Categorical | sigmoid, pdf |
| | relevance_percentile | Real | Uniform (60 to 90) |
| | resampling_strategy | Categorical | balance, average, extreme |
| | delta | Real | Log-uniform (0.01 to 0.2) |
| | sigmoid_cl | Real | 10th to 45th percentile of y_train |
| | sigmoid_ch | Real | 55th to 90th percentile of y_train |
| | pdf_bandwidth | Real | Log-uniform (0.1 to 2.0) |

The core of the backbone is a residual block structure. Each block consists of a Linear layer followed by pre-activation Layer Normalization, a GELU activation function, and Dropout. This design (e.g., LayerNorm before activation, GELU over ReLU) is known to improve training stability and performance on tabular data. Furthermore, we incorporate a residual connection that adds a scaled projection of the block's input to its output, facilitating gradient flow and enabling deeper models. The specific layer-by-layer configuration is detailed in Table 8.

Table 8: The standardized MLP backbone architecture. $B$ denotes the batch size, $D_{in}$ the input feature dimension, and $H_1, H_2$ the hidden dimensions.

| Layer | Details | Output Shape |
|---|---|---|
| Input | - | $(B, D_{in})$ |
| *Block 1* | | |
| Linear | Xavier Init. | $(B, H_1)$ |
| LayerNorm | Pre-activation | $(B, H_1)$ |
| GELU | - | $(B, H_1)$ |
| Dropout | $p$ = dropout_rate | $(B, H_1)$ |
| *Block 2* | | |
| Linear | Xavier Init. | $(B, H_2)$ |
| LayerNorm | Pre-activation | $(B, H_2)$ |
| GELU | - | $(B, H_2)$ |
| Dropout | $p$ = dropout_rate | $(B, H_2)$ |
| Output Linear | Xavier Init. (gain=0.1) | $(B, 1)$ |
| Residual Path | Linear Projection from Input | $(B, 1)$ |
| **Final Output** | Output Linear + $0.1 \times$ Residual Path | $(B, 1)$ |

Table 9: Hyperparameter Search Space for MLP-based Algorithms

| Algorithm | Hyperparameter | Type | Search Space / Distribution |
|---|---|---|---|
| MLP | hidden_dim1 | Categorical | 128, 256, 512 |
| | hidden_dim2 | Categorical | 64, 128, 256 |
| | dropout | Real | Uniform (0.1 to 0.4) |
| | batch_size | Categorical | 32, 64, 128, 256* |
| | optimizer | Categorical | Adam, AdamW |
| | learning_rate | Real | Log-uniform ($10^{-5}$ to $10^{-2}$) |
| | weight_decay | Real | Log-uniform ($10^{-6}$ to $10^{-3}$) |
| | base_loss | Categorical | L1, MSE |
| MLP_RankSim | ranksim_lambda_val | Real | Log-uniform (0.1 to 10.0) |
| | ranksim_alpha | Real | Log-uniform (0.1 to 10.0) |
| MLP_ConR | conr_distance_threshold | Real | Uniform (0.1 to 5.0) |
| | conr_temperature | Real | Log-uniform (0.01 to 1.0) |
| | conr_pushing_power | Real | Log-uniform (0.001 to 0.1) |
| | conr_alpha | Real | Log-uniform (0.1 to 10.0) |
| | conr_mse_weight | Real | Log-uniform (0.1 to 10.0) |
| MLP_LDS | lds_ks | Integer | 3 to 9 (odd numbers) |
| | lds_sigma | Real | Log-uniform (0.5 to 5.0) |

## A.3 DETAILED ABLATION STUDY AND COMPONENT ANALYSIS

To rigorously disentangle the complexity of the CASMIR framework and isolate the contribution of each component, we conducted a detailed ablation study on the Diabetes dataset. We compared the **Full CASMIR** model against seven variants to validate specific sub-components:

- **MoE-Only**: A baseline using only the raw features with the MoE architecture, removing the CAS module to measure the independent performance of the MoE structure.

Table 10: Hyperparameter Search Space for CASMIR(Ours)

| Algorithm | Hyperparameter | Type | Search Space / Distribution |
|---|---|---|---|
| CASMIR | expert_dim1 | Categorical | 128, 256, 512 |
| | expert_dim2 | Categorical | 64, 128, 256 |
| | gate_dim1 | Categorical | 64, 128, 256 |
| | num_experts | Integer | Fixed (typically 3) |
| | batch_size | Categorical | 32, 64, 128, 256* |
| | k_neighbors | Integer | 5 to 15 |
| | feature_bw | Real | Log-uniform (0.5 to 5.0) |
| | label_bw | Real | Log-uniform (1.0 to 50.0) |
| | density_factor | Real | Uniform (0.0 to 0.5) |
| | strength_base | Real | Uniform (0.1 to 0.9) |
| | density_c | Real | Log-uniform (1.0 to 50.0) |
| | optimizer | Categorical | Adam, AdamW |
| | learning_rate | Real | Log-uniform ($10^{-5}$ to $10^{-2}$) |
| | expert_dropout | Real | Uniform (0.1 to 0.4) |
| | weight_decay | Real | Log-uniform ($10^{-6}$ to $10^{-3}$) |
| | lambda_aux | Real | Uniform (0.1 to 1.0) |
| | lambda_load | Real | Uniform (0.01 to 0.2) |

- **MixUp-MoE**: A model trained with standard MixUp augmentation instead of CAS, used to demonstrate that CAS is more effective for imbalanced regression than generic augmentation techniques.

- **CAS-Only**: A model using a single MLP predictor with the full CAS module, validating that expert specialization via MoE is essential for performance improvement.

- **CAS-Feature-Only**: A variant where neighbors are selected solely based on feature similarity (excluding Target Similarity) to analyze the importance of the Feature-Target 'Coupled' mechanism.

- **CAS-No-Learnable-Metric**: A model using standard Euclidean distance (1) instead of learnable weights ($w$) to verify the utility of the learned metric for handling tabular feature heterogeneity.

- **Fixed Smoothness ($s_i = 0$)**: Smoothing strength is fixed to 0 (no smoothing), used to check for increased variance and overfitting in sparse regions.

- **Fixed Smoothness ($s_i = 1$)**: Smoothing strength is fixed to 1 (using only neighbor information), used to check for increased bias and underfitting in many-shot regions.

- **CASMIR**: The final model integrating all proposed components (Coupled Adaptive Smoothing, Density-Gated MoE, Asymmetric Training), optimizing the bias-variance trade-off.

Table 11: Detailed Ablation Study Results on Diabetes Dataset

| Methods/Shots | MAE (Balanced) | | | | MAE (Original Imbalanced) | | | |
|---|---|---|---|---|---|---|---|---|
| | **All** | **Many** | **Med** | **Few** | **All** | **Many** | **Med** | **Few** |
| MoE-Only | 59.41 | 57.32 | 37.34 | 83.57 | 56.90 | 59.74 | 49.76 | 83.57 |
| MixUp-MoE | 55.63 | 56.90 | 22.83 | 87.16 | 53.57 | 63.28 | 38.71 | 87.16 |
| CAS-Only | 62.78 | 77.06 | 21.07 | 90.22 | 57.32 | 69.82 | 39.93 | 90.22 |
| CAS-Feature-Only | 74.37 | 84.74 | 50.42 | 87.95 | 62.89 | 73.65 | 48.46 | 87.95 |
| CAS-No-Learnable-Metric | 59.82 | 66.54 | 29.07 | 83.86 | 50.30 | 55.78 | 39.48 | 83.86 |
| Fixed Smoothness ($s_i = 0$) | 56.31 | **50.61** | 28.18 | 90.13 | 51.43 | **54.64** | 41.92 | 90.13 |
| Fixed Smoothness ($s_i = 1$) | 63.06 | 76.15 | 25.40 | 87.65 | 60.45 | 69.60 | 47.19 | 87.65 |
| **CASMIR (Ours)** | **50.01** | 53.52 | **18.87** | **77.65** | **49.27** | 56.02 | **38.12** | **77.65** |

The experimental results on the Diabetes dataset, presented in Table 11, clearly demonstrate the distinct contribution of each module within the CASMIR framework to the overall performance improvement.

- **Effect of MoE Structure (MoE-Only vs. CASMIR):** Compared to using the MoE architecture alone (**MoE-Only**), **CASMIR**, which integrates the CAS module, achieved an approximate 15.8% improvement ($59.41 \rightarrow 50.01$) in Balanced MAE. Notably, while MoE-Only struggled in the **Few-shot** region (83.57), CASMIR significantly reduced the error to **77.65**, proving that the CAS module effectively compensates for data sparsity in the tail regions.

- **Importance of 'Coupled' and 'Learnable' Mechanisms:** The superior performance of **CAS-Only** over **CAS-Feature-Only** ($74.37 \rightarrow 62.78$) suggests that relying solely on feature similarity is insufficient; considering *Target Similarity* is essential for accurate neighbor selection. Furthermore, the performance gap between **CASMIR** and **CAS-No-Learnable-Metric** ($59.82 \rightarrow 50.01$) validates the necessity of the learnable weight vector $w$ to handle the heterogeneous feature importance inherent in tabular data.

- **Role of Adaptive Smoothing Strength** ($s_i$)**:** Neither **Fixed Smoothness** ($s_i = 0$), which corresponds to no smoothing, nor **Fixed Smoothness** ($s_i = 1$), which corresponds to full smoothing, achieved optimal performance. In contrast, **CASMIR**, which dynamically adjusts $s_i$ based on data density, achieved balanced performance across both Many-shot and Few-shot regions. This confirms its capability to effectively navigate the bias-variance trade-off by reducing variance in sparse regions while preserving information in dense regions.

A.4 COMPUTATIONAL AND MEMORY EFFICIENCY ANALYSIS

Our proposed CASMIR framework employs an *asymmetric design* where heavy computations, such as KNN search and feature calibration in the CAS module, are intentionally front-loaded into the training phase. During inference, the CAS module is deactivated, allowing the model to operate as a standard feed-forward network.

As detailed in Table 13, this strategy proves highly effective. Although CASMIR exhibits higher latency relative to a Vanilla MLP, its absolute inference time consistently remains under 1.0ms across all datasets (e.g., 0.90ms on the large-scale fps_benchmark), ensuring practical real-time feasibility. Furthermore, the structural complexity imposes minimal memory overhead, with only an $\approx$11–13% increase in RAM and $\approx$5% increase in GPU usage compared to a single MLP. This demonstrates that CASMIR offers a highly practical trade-off, delivering significant performance gains with negligible impact on deployment latency and hardware resources.

Table 12: Algorithm Abbreviations

| Abbreviation | Full Name |
|---|---|
| CAT | CatBoost |
| LGB | LightGBM |
| XGB | XGBoost |
| XGN | XGBoost+Gaussian Noise |
| XSM | XGBoost+SMOTER |
| SEN | Simple Ensemble |
| MLP | Vanilla MLP |
| MSI | MLP+SQRT_INV |
| MLD | MLP+LDS |
| MRS | MLP+RankSim |
| MGA | MLP+BMSE(GAI) |
| MBC | MLP+BMSE(BMC) |
| MCR | MLP+ConR |
| CASMIR | CASMIR |

Table 13: Detailed Analysis of Computational and Memory Complexity (Vanilla MLP vs. Simple Ensemble vs. CASMIR)

| Dataset | Model | Training Time (s) | Inference Latency (ms) | Peak RAM (MB) | Peak GPU (MB) |
|---|---|---|---|---|---|
| **diabetes** (Small) | Vanilla MLP | 4.75 | 0.25 | 346.7 | 65.5 |
| | Simple Ensemble | 12.61 | 0.68 | 321.2 | 68.0 |
| | **CASMIR (Ours)** | 20.65 | 0.81 | 394.2 | 68.1 |
| **parkinsons_ telemonitoring** (Medium) | Vanilla MLP | 62.37 | 0.25 | 357.8 | 67.2 |
| | Simple Ensemble | 184.72 | 0.67 | 348.2 | 66.5 |
| | **CASMIR (Ours)** | 267.15 | 0.80 | 399.7 | 66.6 |
| **fps_benchmark** (Large) | Vanilla MLP | 259.68 | 0.35 | 350.8 | 76.0 |
| | Simple Ensemble | 764.24 | 0.83 | 341.2 | 79.8 |
| | **CASMIR (Ours)** | 1115.49 | 0.90 | 394.7 | 80.2 |

Training time measured on GPU (Nvidia 4090) with single-threaded execution.
Dataset sizes (Train/Test): Diabetes (286/21), Parkinsons (3818/117), FPS Benchmark (16005/1065).
RAM and GPU usage indicate peak consumption during the process.

## A.5 ADDITIONAL FEATURE VISUALIZATION

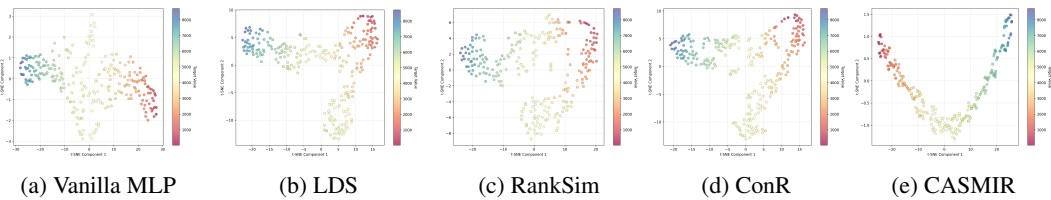

    (a) Vanilla MLP      (b) LDS      (c) RankSim      (d) ConR      (e) CASMIR

Figure 4: A t-SNE visualization comparing learned feature representations on a bike_sharing tabular dataset. The visualization displays the final hidden layer's feature representations for four models: (a) **Vanilla MLP**, (b) **LDS**, (c) **RankSim**, (d) **ConR**, and (e) **our proposed model, CASMIR**. Each point represents a sample from the test set, with its color determined by the continuous target value.

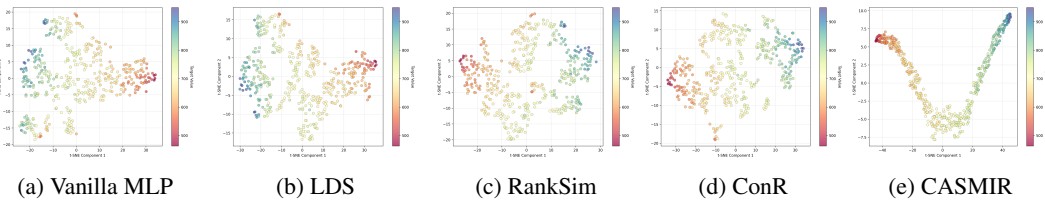

    (a) Vanilla MLP      (b) LDS      (c) RankSim      (d) ConR      (e) CASMIR

Figure 5: A t-SNE visualization comparing learned feature representations on a Moneyball tabular dataset. The visualization displays the final hidden layer's feature representations for four models: (a) **Vanilla MLP**, (b) **LDS**, (c) **RankSim**, (d) **ConR**, and (e) **our proposed model, CASMIR**. Each point represents a sample from the test set, with its color determined by the continuous target value.

To supplement the t-SNE visualization in the main paper (Figure 2), this section provides additional feature visualizations for two more datasets: bike_sharing (Figure 4) and Moneyball (Figure 5). While baseline methods often yield fragmented or collapsed representations where the target structure is unclear, CASMIR consistently learns a coherent and continuous manifold that clearly preserves the ordinal structure of the target variable across different datasets.

## A.6 THEORETICAL ANALYSIS AND JUSTIFICATION

The strong empirical performance of CASMIR is not incidental but stems from a principled design rooted in established machine learning theory. In this section, we provide a unified theoretical

analysis, linking our framework to **Vicinal Risk Minimization (VRM)** and providing mathematical intuition for our core components and asymmetric design.

### A.6.1 THEORETICAL FOUNDATION: ADAPTIVE VICINAL RISK MINIMIZATION

Standard model training based on Empirical Risk Minimization (ERM) is prone to overfitting on sparse and noisy samples. To overcome this, Vicinal Risk Minimization (VRM) was proposed to improve generalization by minimizing risk not only on the training points but also in their local *vicinity* (Chapelle et al., 2000; Zhang et al., 2018; Huang & Wei, 2022).

Our CAS module functions as an advanced, adaptive implementation of this principle. For each sample $x_i$, the calibrated feature $\tilde{x}_i$ acts as a **principled vicinal sample**:

$$\tilde{x}_i = (1 - s_i) \cdot x_i + s_i \cdot \mu_{\text{smooth}}, \quad \text{where} \quad \mu_{\text{smooth}} = \sum_{j \in \mathcal{N}(i)} \omega_{ij} x_j. \tag{9}$$

The key innovation of CASMIR is the **adaptivity** of the smoothing strength $s_i$, which is inversely proportional to the target density. This allows the model to intelligently navigate the **bias-variance trade-off**:

- **In dense, many-shot regions** ($s_i \to 0$): $\tilde{x}_i \approx x_i$. The model effectively performs ERM, trusting the abundant data and preserving fine-grained details (low bias).
- **In sparse, few-shot regions** ($s_i \to 1$): The model performs a strong form of VRM. This pulls the representation of a noisy sample towards the stable mean of its neighbors, introducing a controlled bias to effectively suppress the high variance that causes overfitting.

### A.6.2 MATHEMATICAL INTUITION OF CORE COMPONENTS

To ensure that the VRM process is effective on heterogeneous tabular data, we introduce specific mechanisms for neighbor selection and routing.

**Data-Driven Metric Learning (Revisiting Eq. 1).** The weighted distance metric $d(x_i, x_j)^2 = \sum_{d=1}^{D} w_d \cdot (x_{i,d} - x_{j,d})^2$ addresses the varying importance of tabular features. The learnable weight vector $w$ **reshapes the feature space**, expanding informative axes and compressing noisy ones. This ensures neighbors are selected based on *semantic distance* rather than simple geometric distance.

**Coupled Neighbor Weighting (Revisiting Eq. 2).** Similarity in feature space alone is insufficient. We enforce a **coupling constraint** where neighbors must share similarity in both feature space and target density context. Additionally, the inverse density weighting ($\frac{\alpha}{d_j + \epsilon}$) ensures that neighbors from sparse regions carry higher information value, preventing the majority class from dominating the smoothing process.

**High-Frequency Preservation for Gating (Revisiting Eq. 4).** The gating network uses raw features $x_{\text{norm}}$ instead of smoothed features. This is because density classification requires clear decision boundaries. **Sharp high-frequency details** contained in the raw data are essential for accurate routing; smoothing would blur these boundaries and degrade expert specialization.

### A.6.3 JUSTIFICATION FOR ASYMMETRIC DESIGN: INDUCTIVE BIAS TRANSFER

A distinctive feature of CASMIR is its asymmetric data flow: training experts on calibrated features $\tilde{x}$ while inferring on raw features $x$. This strategy is theoretically grounded in **Inductive Bias Transfer**.

**Training as Data Augmentation.** Using $\tilde{x}$ during training acts as a form of *Training-time Data Augmentation* and *Implicit Regularization*, analogous to Mixup (Zhang et al., 2018) or Dropout (Srivastava et al., 2014). In sparse regions where the conditional distribution $P(y|x)$ is unstable, $\tilde{x}$ provides a stabilized signal, increasing the Signal-to-Noise Ratio (SNR).

**Inference via Internalized Structure.** Crucially, although the experts process $\tilde{x}$, the network parameters $\theta$ are updated to minimize the loss on the underlying target manifold. Through this process, the experts **internalize the smoothed geometric structure**, effectively learning to filter high-frequency noise from raw inputs $x$. The superior inference performance on raw features (Table 2) and the coherent manifold visualization (Figure 2) empirically validate that this regularization effect is successfully transferred to the model weights, allowing the model to bypass the costly smoothing process at inference time while retaining its benefits.

A.7 ALGORITHM

To provide a clear and reproducible definition of our framework, we present the pseudocode for CASMIR. Algorithm 1 outlines the overall training procedure, highlighting the asymmetric data flow for the gating and expert networks. Algorithm 2 provides a detailed step-by-step description of the core CAS module.

---

**Algorithm 1** CASMIR: Training Procedure

---

**Require:** Training dataset $\mathcal{D} = \{(x_i, y_i)\}_{i=1}^{N}$, Model parameters $\Theta = \{\theta_g, \theta_{e_k}, \mathbf{w}, \tau\}$.
**Ensure:** Trained model parameters $\Theta$.

  **procedure** PREPROCESSING
     Pre-calculate target densities $\{d_i\}$ and thresholds $d_{\text{low}}, d_{\text{high}}$ using KDE on $\{y_i\}_{i=1}^{N}$.
  **end procedure**

  **procedure** TRAIN
    **for** each epoch **do**
      **for** each batch $\mathcal{B} = \{(x_i, y_i, d_i)\}_{i=1}^{N_b}$ in $\mathcal{D}$ **do**
        $\{x_{i,\text{norm}}\} \leftarrow \text{Normalize}(\{x_i\})$
        $\{\tilde{x}_i\} \leftarrow \text{CAS}(\{x_{i,\text{norm}}\}, \{y_i\}, \{d_i\}; \mathbf{w})$         $\triangleright$ See Algorithm 2 for details
                                                $\triangleright$ Asymmetric forward pass
        $\{g_i\} \leftarrow \text{Softmax}(G(\{x_{i,\text{norm}}\})/\tau)$         $\triangleright$ Gate uses original features
        $\{\hat{y}_i\} \leftarrow \sum_{k=1}^{K} g_{i,k} \cdot E_k(\{\tilde{x}_i\})$         $\triangleright$ Experts use calibrated features
                                                  $\triangleright$ Combined loss (see Eq. 5)
        $\mathcal{L}_{\text{total}} \leftarrow \mathcal{L}_{\text{reg}}(\hat{y}_i, y_i) + \lambda_{\text{aux}}\mathcal{L}_{\text{aux}} + \lambda_{\text{load}}\mathcal{L}_{\text{load}}$
        Update $\Theta$ by descending the gradient $\nabla\mathcal{L}_{\text{total}}$
      **end for**
    **end for**
  **end procedure**

---

**Algorithm 2** Coupled Adaptive Smoothing (CAS) Module

---

**Require:** Batch of normalized features $\{x_i\}$, labels $\{y_i\}$, densities $\{d_i\}$, feature weights $\mathbf{w}$.
**Ensure:** Calibrated features $\{\tilde{x}_i\}$.
  **procedure** CAS
    **for** $i = 1$ to $N_b$ **do**
      Find $k$-nearest neighbors $\mathcal{N}_i$ for $x_i$ using the weighted distance metric from Eq. (1).
      Calculate neighbor influence weights $\{\omega_{ij}\}$ for all $j \in \mathcal{N}_i$ as defined in Eq. (2).
      Determine the adaptive smoothing strength $s_i$ based on the sample's own density $d_i$.
      Compute the final calibrated feature $\tilde{x}_i$ by blending $x_i$ and the neighbors' average as in Eq. (3).
    **end for**
    **return** $\{\tilde{x}_i\}_{i=1}^{N_b}$
  **end procedure**

---

Table 14: Novelty Comparison with Key Imbalanced Regression Methods

| Method | Dual-Path MoE | Feature Calibration | Adaptive Smoothing | Coupled Smoothing | Density Weighting | Learnable Weights | Continuous Handling | Tabular Design | Test-Time Adaptation |
|---|---|---|---|---|---|---|---|---|---|
| **CASMIR (Ours)** | ✓ | ✓ | ✓ | ✓ | ✓ | ✓ | ✓ | ✓ | ✗ |
| MATI | ✗ | ✓ | ✓ | ✗ | ✓ | ✗ | Binning | ✓ | ✓ |
| SQRT_INV | ✗ | ✗ | ✗ | ✗ | ✓ | ✗ | ✓ | ✗ | ✗ |
| LDS | ✗ | ✓ | ✓ | ✗ | ✓ | ✗ | Binning | ✗ | ✗ |
| FDS | ✗ | ✓ | ✓ | ✗ | ✓ | ✗ | Binning | ✗ | ✗ |
| LDS+FDS | ✗ | ✓ | ✓ | ✗ | ✓ | ✗ | Binning | ✗ | ✗ |
| RankSim | ✗ | ✗ | ✗ | ✓ | ✗ | ✗ | ✓ | ✗ | ✗ |
| ConR | ✗ | ✗ | ✗ | ✓ | ✗ | ✗ | ✓ | ✗ | ✗ |
| BMSE(GAI) | ✗ | ✗ | ✗ | ✗ | ✓ | ✗ | ✓ | ✗ | ✗ |
| BMSE(BMC) | ✗ | ✗ | ✗ | ✗ | ✓ | ✗ | ✓ | ✗ | ✗ |

## A.8 ANALYSIS OF NOVELTY AND COMPARISON WITH RELATED METHODS

In this appendix, we aim to clearly delineate the core novelties of our proposed CASMIR methodology by performing a detailed functional comparison against key MLP-based baselines and relevant state-of-the-art methods in the field of imbalanced regression. Table 14 summarizes whether each methodology incorporates specific innovative mechanisms. (For conciseness, the full column headers in Table 14 are abbreviated as follows: `Dual-Path MoE` to `DP-MoE`.

### A.8.1 OVERALL ANALYSIS: THE MULTIFACETED INNOVATIVENESS OF CASMIR

As evidenced in Table 14, CASMIR comprehensively and systematically integrates various key mechanisms for addressing imbalanced regression, offering distinctive contributions particularly in aspects such as Dual-Path MoE, Feature-Target Coupled Smoothing, Learnable Feature Weights, and its explicit design for tabular data. While most baseline methodologies focus on specific imbalance resolution strategies (e.g., sample weighting, feature/label smoothing), CASMIR tackles the imbalance problem head-on across the entire process, from input feature calibration (Direct Feature Calibration) to the model architecture (Dual-Path MoE), and learning optimization, all while explicitly accounting for the unique characteristics of tabular data.

### A.8.2 CORE NOVELTY ANALYSIS (CASMIR-EXCLUSIVE FEATURES)

- **Dual-Path MoE**: CASMIR introduces an **asymmetric dual-path Mixture-of-Experts (MoE) architecture where the Gating Network and Experts receive different inputs (original features and calibrated features, respectively)**. This novel design allows the Gating Network to accurately identify density regions from comprehensive features, while Experts process the precisely calibrated features from the CAS module to enhance prediction accuracy. This unique architecture is not found in existing MoE-based methods and plays a critical role in optimizing expert specialization for imbalanced datasets.

- **Feature-Target Coupled Smoothing**: While RankSim and ConR learn relationships with targets within the feature embedding space, CASMIR's CAS module **directly smooths (calibrates) the input features themselves** by simultaneously considering feature space similarity, target space similarity, and inverse target density. This approach mitigates the imbalance problem before model training, preventing the underestimation of samples in sparse regions.

- **Learnable Feature Weights**: The CAS module in CASMIR **dynamically adjusts the importance of each feature during similarity calculations through learnable weights (w)**. This effectively handles the heterogeneous nature of tabular data and optimizes the similarity metric in a data-driven manner, thereby improving performance. In contrast to existing methods that typically use fixed feature weights or simple distance metrics, CASMIR enhances adaptivity by allowing the model to learn these weights.

- **Designed for Tabular Features**: While all methods are applied to tabular data, CASMIR is explicitly designed to leverage and mitigate the challenges inherent in tabular datasets. Its Learnable Feature Weights explicitly address feature heterogeneity and varying importance, which is critical for mixed-type tabular features. Furthermore, the Feature-Target Coupled Smoothing within the CAS module is tailored to refine feature representations by understanding inter-feature relationships and their correlation with target values, a common

challenge in tabular learning. This contrasts with methods primarily adapting deep learning techniques from other domains (e.g., computer vision) without specific tabular-centric architectural considerations beyond basic MLP backbones.

### A.8.3 Evolved Integration and Differentiation (Shared Features with Advanced Implementation)

CASMIR shares some conceptual similarities with other methodologies, but it integrates or develops these concepts in a more sophisticated and effective manner.

- **Direct Feature Calibration & Adaptive Smoothing**: LDS, FDS, and MATI also exhibit indirect or direct feature calibration effects by smoothing(or synthesizing) feature or label distributions. However, CASMIR's CAS Module **adaptively and finely tunes the smoothing intensity for each sample** based on continuously estimated target densities using KDE. This approach better captures the subtle structures of imbalanced data compared to methods relying on discrete bin-based density estimation or fixed kernels.

- **Density-Aware Weighting**: Methods like SQRT_INV, LDS, FDS, and BMSE family also leverage target density to adjust loss or sample weights. CASMIR similarly utilizes target density, but it integrates this information into the **smoothing weight calculation** of the CAS module and the **density region prediction** of the Gating Network, creating a synergistic effect within the architecture. Specifically, using KDE for continuous target density estimation allows for smoother weight adjustments than discrete bin-based methods.

- **Continuous Density/Distance Handling (KDE & Metric)**: Vanilla MLP, SQRT_INV, RankSim, ConR, and BMSE handle features continuously or utilize target values directly in their loss functions. CASMIR extends this by **continuously estimating target density via KDE** and calculating feature-target similarity through a **learned metric-based distance**, which is then used for smoothing. This differentiates it from discrete bin-based approaches (LDS/FDS/MATI) and enables sensitivity to subtle changes in target values.

### A.8.4 Test-Time Adaptation and CASMIR's Strategy

Some advanced methodologies, such such as MATI, enhance prediction performance through additional adaptation processes applied at test time. In contrast, CASMIR is designed to operate without explicit Test-Time Adaptation. This approach suggests that CASMIR's innovative architecture, including its CAS module and Dual-Path MoE, aims to address the inherent imbalance problem primarily during the training phase. Consequently, CASMIR is intended to deliver robust and stable prediction performance without incurring the additional computational overhead or data dependency often associated with Test-Time Adaptation during inference.

### A.8.5 Conclusion

The comparison in Table 14 suggests that CASMIR makes significant strides beyond existing solutions in imbalanced regression by systematically combining and advancing several innovative mechanisms. Its distinct contributions, which include the Dual-Path MoE, Feature-Target Coupled Smoothing, Learnable Feature Weights, and its explicit design for tabular data, alongside sophisticated imbalance resolution strategies such as Direct Feature Calibration and Continuous Density Handling, are posited as key drivers contributing to CASMIR's ability to achieve competitive state-of-the-art performance.

## A.9 Additional Experimental Results

This section provides the complete, per-dataset performance results that were summarized in the main paper. We report the Mean Absolute Error (MAE) for all 14 models on the 40 datasets. Table 15 shows the results on the **balanced test set**, while Table 16 shows the results on the **original imbalanced test set**.

In all result tables, lower MAE values indicate better performance. For each dataset (row), the **best-performing** model is marked in **bold**, and the **second-best** is underlined. Table 12 provides a legend for the algorithm abbreviations used in the result tables.

Table 15: Performance Results across All Datasets (MAE Overall Balanced)

| Dataset | Tree-based | | | Enhanced Tree | | Ensemble | MLP-based Methods | | | | | | | Ours |
|---|---|---|---|---|---|---|---|---|---|---|---|---|---|---|
| | **LGB** | **CAT** | **XGB** | **XGN** | **XSM** | **SEN** | **MLP** | **MSI** | **MLD** | **MRS** | **MGA** | **MBC** | **MCR** | **CASMIR** |
| Abalone | 1.70 | 1.76 | 1.68 | 1.68 | 1.70 | 1.66 | 1.65 | 1.65 | 1.70 | **1.64** | 2.00 | 1.87 | 1.70 | 1.66 |
| Moneyball | 21.10 | 19.06 | 19.33 | 20.75 | 19.47 | 18.70 | 18.01 | 4.87K | 18.82 | 18.08 | 18.84 | 133.13 | 17.06 | **16.22** |
| airfoild | 2.03 | 1.66 | **1.29** | 1.66 | 1.73 | 1.49 | 1.43 | 1.53 | 1.47 | 1.50 | 1.46 | 2.08 | 1.41 | 1.80 |
| availPwr | 3.78 | 3.15 | **2.67** | 3.19 | 6.72 | 5.19 | 6.37 | 7.34 | 6.27 | 6.06 | 5.76 | 10.94 | 7.68 | 5.81 |
| bank32nh | 0.12 | 0.13 | 0.12 | 0.11 | 0.10 | 0.11 | 0.10 | **0.09** | 0.11 | 0.12 | 0.48 | 0.60 | 0.10 | 0.11 |
| bank8FM | 0.03 | 0.03 | 0.03 | 0.03 | 0.03 | 0.03 | 0.03 | 0.03 | 0.03 | 0.03 | 0.31 | 0.32 | 0.03 | **0.03** |
| bike_sharing | 149.85 | 128.41 | 118.97 | 124.18 | 121.98 | 477.89 | 209.91 | 199.21 | 234.11 | 211.25 | 208.12 | 144.78 | 208.50 | **73.81** |
| combined_cy... | 2.54 | 2.86 | 2.40 | 2.39 | **2.38** | 3.19 | 3.41 | 3.36 | 3.29 | 3.52 | 3.33 | 3.62 | 3.27 | 3.51 |
| communities:. | **0.11** | 0.12 | 0.11 | 0.11 | 0.12 | 0.11 | 0.12 | 0.12 | 0.12 | 0.12 | 0.36 | 0.27 | 0.12 | 0.11 |
| concreteStre... | 3.49 | 3.23 | 3.05 | 3.81 | 2.98 | 3.15 | 2.97 | 3.18 | 2.90 | **2.90** | 3.02 | 3.00 | 2.98 | 3.44 |
| cps88wages | 477.73 | 479.74 | 443.20 | 435.94 | 422.07 | 458.92 | 433.36 | **394.07** | 469.83 | 435.48 | 430.21 | 429.84 | 431.67 | 397.21 |
| cpuSm | 2.98 | 2.90 | **2.67** | 2.75 | 2.76 | 2.68 | 2.69 | 2.78 | 3.76 | 2.84 | 2.81 | 3.09 | 2.72 | 2.85 |
| dAiler | 0.00 | 0.00 | 0.00 | 0.00 | 0.00 | **0.00** | 0.00 | 0.00 | 0.00 | 0.00 | 0.24 | 0.23 | 0.00 | 0.00 |
| diabetes | **58.55** | 59.35 | 66.33 | 66.53 | 61.65 | 59.49 | 67.62 | 64.81 | 61.29 | 65.40 | 64.63 | 66.55 | 62.96 | 60.21 |
| diamond_reg... | 500.63 | 515.81 | 508.98 | 515.71 | 511.42 | **494.34** | 10.2K | 438.1K | 2.4M | 3.26K | 138.3K | 286.9K | 6.81K | 502.08 |
| ecoli70 | 0.29 | 0.30 | 0.28 | 0.28 | 0.29 | 0.28 | 0.26 | 0.28 | 0.28 | 0.54 | 2.45 | 3.49 | **0.26** | 0.27 |
| energy_effi... | 0.48 | **0.37** | 0.45 | 0.47 | 0.42 | 0.40 | 0.45 | 0.56 | 0.49 | 0.48 | 0.88 | 2.17 | 0.41 | 0.40 |
| forest_fires | **17.78** | 17.99 | 19.30 | 26.20 | 18.26 | 19.18 | 19.27 | 20.57 | 19.25 | 19.83 | 20.32 | 18.80 | 18.66 | 19.43 |
| fps_benchmark | 2.41 | 1.23 | 1.38 | 1.58 | 1.77 | **0.60** | 1.52 | 1.68 | 1.38 | 1.24 | 1.43 | 1.91 | 1.09 | 0.87 |
| fuelCons | 0.37 | 0.35 | **0.29** | 0.30 | 0.34 | 0.38 | 0.44 | 0.52 | 0.43 | 0.52 | 2.30 | 51.41 | 0.42 | 0.35 |
| geographical... | 17.24 | 17.54 | 16.13 | 15.84 | 15.91 | 16.86 | 16.12 | 15.56 | **15.26** | 15.78 | 16.15 | 15.43 | 16.04 | 15.84 |
| grid_stabil... | 0.01 | 0.01 | 0.01 | 0.01 | 0.01 | **0.00** | 0.01 | 0.01 | 0.01 | 0.01 | 0.43 | 0.40 | 0.01 | 0.00 |
| kin8nm | 0.12 | 0.10 | 0.12 | 0.12 | 0.12 | **0.05** | 0.06 | 0.06 | 0.06 | 0.06 | 0.72 | 0.69 | 0.06 | 0.06 |
| kings_county | 92.0K | 94.1K | **89.7K** | 91.9K | 91.8K | 106.2K | 102.3K | 106.5K | 106.9K | 109.0K | 105.3K | 101.6K | 741.0K | 99.0K |
| machineCpu | 76.91 | 72.32 | 59.28 | 55.36 | 54.95 | 72.32 | **48.55** | 50.90 | 51.67 | 52.14 | 54.40 | 72.28 | 60.30 | 71.65 |
| magic_irri | 7.86 | 7.89 | 7.69 | **7.28** | 7.33 | 7.54 | 7.62 | 7.44 | 7.48 | 7.44 | 7.54 | 7.83 | 7.78 | 7.41 |
| maxTorque | 4.57 | 9.02 | 3.69 | 4.39 | 6.10 | 4.29 | 94.71 | 96.93 | 96.84 | 94.04 | 95.16 | 79.27 | 96.53 | **3.29** |
| miami_housi... | 76.3K | 76.9K | **71.2K** | 76.8K | 72.2K | 75.8K | 79.9K | 80.1K | 86.8K | 81.6K | 79.2K | 77.8K | 570.4K | 76.8K |
| nhanes_age | 15.08 | 15.37 | 14.95 | 15.48 | 15.29 | 14.94 | 14.77 | 14.87 | 14.48 | 15.09 | 14.84 | 15.02 | 15.00 | **13.91** |
| online_news... | 12.3K | 12.2K | 11.3K | 11.2K | **10.6K** | 11.6K | 11.1K | 11.1K | 11.8K | 11.2K | 11.2K | 11.2K | 11.0K | 11.4K |
| parkinsons_... | 0.25 | 0.52 | **0.14** | 0.21 | 0.19 | 0.17 | 0.34 | 0.41 | 0.38 | 0.31 | 0.43 | 0.98 | 0.36 | 0.25 |
| pumadyn32nh | 0.02 | 0.02 | 0.02 | **0.02** | 0.02 | 0.02 | 0.02 | 0.02 | 0.02 | 0.04 | 0.37 | 0.35 | 0.02 | 0.02 |
| qsar_aquati... | 1.02 | 0.80 | 0.88 | 0.89 | 0.90 | 0.85 | 0.88 | 0.80 | **0.79** | 0.83 | 2.21 | 2.20 | 0.83 | 0.85 |
| red_wine | 0.81 | 0.75 | 0.65 | **0.61** | 0.65 | 0.75 | 0.84 | 0.70 | 0.71 | 0.76 | 3.61 | 3.83 | 0.77 | 0.77 |
| servo | 0.24 | 0.27 | 0.36 | 0.26 | 0.27 | **0.22** | 0.25 | 0.23 | 0.26 | 0.26 | 1.46 | 1.29 | 0.40 | 0.23 |
| socmob | 8.70 | 8.62 | 8.04 | 8.93 | 10.48 | 5.84 | **5.48** | 7.61 | 6.00 | 6.59 | 7.48 | 6.88 | 6.96 | 5.80 |
| solar_flare | 1.42 | 1.43 | 1.30 | 1.21 | **1.15** | 1.42 | 1.31 | 1.26 | 1.42 | 1.30 | 1.50 | 1.33 | 1.27 | 1.31 |
| space_ga | 0.11 | 0.11 | 0.10 | 0.11 | 0.10 | **0.08** | 0.09 | 0.09 | 0.09 | 0.10 | 0.77 | 0.53 | 0.09 | 0.08 |
| superconduct... | 7.55 | 8.41 | **7.06** | 7.21 | 7.87 | 7.37 | 9.28 | 8.90 | 8.67 | 8.82 | 8.87 | 9.12 | 8.80 | 7.55 |
| white_wine | 0.79 | 0.74 | 0.68 | **0.65** | 0.70 | 0.70 | 0.71 | 0.70 | 0.72 | 0.76 | 3.19 | 3.11 | 0.74 | 0.72 |

Table 16: Performance Results across All Datasets (MAE Overall Original Imbalanced)

| Dataset | Tree-based | | | Enhanced Tree | | Ensemble | MLP-based Methods | | | | | | | Ours |
|---|---|---|---|---|---|---|---|---|---|---|---|---|---|---|
| | LGB | CAT | XGB | XGN | XSM | SEN | MLP | MSI | MLD | MRS | MGA | MBC | MCR | CASMIR |
| Abalone | 1.48 | 1.51 | 1.49 | 1.58 | 1.50 | **1.46** | 1.48 | 2.06 | 1.68 | 1.62 | 1.89 | 1.80 | 1.53 | 1.46 |
| Moneyball | 18.00 | 16.72 | 16.80 | 18.30 | 17.84 | 17.57 | 16.63 | 1.32K | 42.54 | 17.05 | 17.62 | 71.09 | 17.03 | **15.85** |
| airfoild | 1.40 | 1.33 | **1.22** | 1.45 | 1.47 | 1.29 | 1.44 | 1.67 | 1.47 | 1.37 | 1.43 | 1.79 | 1.28 | 1.63 |
| availPwr | 2.26 | 2.04 | **1.60** | 2.27 | 4.71 | 3.73 | 4.82 | 5.34 | 5.45 | 4.31 | 4.08 | 14.68 | 5.30 | 4.33 |
| bank32nh | 0.05 | 0.05 | 0.06 | 0.06 | 0.07 | 0.05 | 0.06 | 0.06 | **0.05** | 0.07 | 0.44 | 0.49 | 0.06 | 0.05 |
| bank8FM | 0.02 | 0.02 | 0.02 | 0.02 | 0.02 | 0.02 | 0.02 | 0.02 | **0.02** | 0.02 | 0.24 | 0.27 | 0.02 | 0.02 |
| bike_sharing | 86.59 | 104.76 | 88.54 | 94.93 | 101.60 | 273.43 | 138.29 | 162.70 | 143.06 | 130.75 | 133.10 | 103.36 | 134.73 | **35.35** |
| combined_cy... | 2.38 | 2.63 | **2.29** | 2.51 | 2.40 | 3.01 | 3.10 | 3.14 | 9.63 | 3.14 | 3.08 | 3.63 | 3.02 | 3.29 |
| communities:. | **0.09** | 0.09 | 0.09 | 0.09 | 0.09 | 0.09 | 0.09 | 0.10 | 0.09 | 0.10 | 0.29 | 0.24 | 0.10 | 0.09 |
| concreteStre... | **2.51** | 2.70 | 2.58 | 3.14 | 2.86 | 2.88 | 2.85 | 3.08 | 3.09 | 2.78 | 2.92 | 2.82 | 2.73 | 3.38 |
| cps88wages | 217.34 | **216.45** | 226.28 | 228.27 | 250.98 | 218.12 | 224.94 | 245.90 | 217.90 | 224.91 | 226.65 | 227.11 | 225.23 | 251.89 |
| cpuSm | 2.12 | 2.06 | **1.90** | 2.05 | 2.04 | 2.03 | 2.16 | 2.30 | 3.10 | 2.29 | 2.31 | 2.56 | 2.18 | 2.19 |
| dAiler | **0.00** | 0.00 | 0.00 | 0.00 | 0.00 | 0.00 | 0.00 | 0.00 | 0.00 | 0.00 | 0.24 | 0.22 | 0.00 | 0.00 |
| diabetes | 48.74 | **48.11** | 50.17 | 51.14 | 50.92 | 49.41 | 50.86 | 51.91 | 49.58 | 50.01 | 51.17 | 52.06 | 48.94 | 51.83 |
| diamond_reg... | 270.79 | 280.35 | 277.71 | 284.34 | 283.15 | **269.19** | 3.29K | 136.1K | 745.4K | 1.14K | 43.3K | 98.1K | 2.24K | 271.22 |
| ecoli70 | 0.24 | 0.23 | **0.23** | 0.24 | 0.24 | 0.28 | 0.27 | 0.26 | 0.25 | 0.38 | 1.95 | 2.48 | 0.25 | 0.26 |
| energy_effi... | 0.51 | 0.40 | 0.44 | 0.50 | 0.42 | 0.42 | 0.46 | 43.39 | 0.48 | 0.48 | 0.83 | 1.64 | 0.43 | **0.38** |
| forest_fires | 10.37 | 10.96 | 15.60 | 26.28 | 12.57 | 11.21 | 12.27 | 14.01 | 11.04 | 24.84 | 12.59 | **9.79** | 10.37 | 10.63 |
| fps_benchmark | 1.47 | 0.69 | 1.04 | 1.37 | 1.56 | **0.37** | 1.20 | 1.20 | 1.04 | 1.00 | 1.12 | 1.14 | 0.87 | 0.59 |
| fuelCons | 0.28 | 0.28 | **0.26** | 0.30 | 0.31 | 0.33 | 0.36 | 0.39 | 0.35 | 0.39 | 1.79 | 13.28 | 0.34 | 0.33 |
| geographical... | 12.17 | 12.44 | 12.47 | 12.53 | 13.72 | 11.24 | 11.52 | 11.88 | 10.84 | 11.71 | 11.82 | 11.37 | 11.46 | **10.83** |
| grid_stabil... | 0.01 | 0.01 | 0.01 | 0.01 | 0.01 | 0.00 | 0.00 | 0.00 | 0.00 | 0.01 | 0.37 | 0.34 | 0.00 | **0.00** |
| kin8nm | 0.09 | 0.08 | 0.09 | 0.10 | 0.10 | **0.05** | 0.06 | 0.06 | 0.06 | 0.06 | 0.76 | 0.73 | 0.06 | 0.05 |
| kings_county | 62.3K | 63.7K | **62.3K** | 64.1K | 63.9K | 73.5K | 69.9K | 72.2K | 71.3K | 74.4K | 71.4K | 70.5K | 540.2K | 68.0K |
| machineCpu | 47.13 | 45.08 | 39.98 | 37.91 | 39.52 | 46.88 | **34.31** | 35.51 | 34.46 | 35.16 | 37.04 | 46.65 | 41.71 | 48.84 |
| magic_irri | 5.69 | 5.67 | **5.64** | 5.72 | 5.84 | 6.05 | 6.00 | 6.20 | 6.04 | 6.01 | 6.39 | 6.73 | 6.18 | 6.17 |
| maxTorque | 2.31 | 4.52 | **1.74** | 2.52 | 3.36 | 2.48 | 76.06 | 76.36 | 76.13 | 75.89 | 75.13 | 587.07 | 76.00 | 1.90 |
| miami_housi... | 45.5K | 45.3K | **42.7K** | 46.5K | 44.1K | 44.7K | 49.2K | 50.4K | 50.2K | 49.5K | 47.2K | 47.2K | 399.4K | 45.6K |
| nhanes_age | **13.11** | 13.13 | 13.43 | 13.85 | 13.60 | 13.41 | 13.29 | 13.36 | 13.16 | 13.27 | 13.19 | 13.39 | 13.49 | 13.35 |
| online_news... | **2.30K** | 2.35K | 3.06K | 4.14K | 5.13K | 2.74K | 3.32K | 3.63K | 2.63K | 3.51K | 3.41K | 3.27K | 3.37K | 3.95K |
| parkinsons_... | 0.21 | 0.44 | **0.12** | 0.23 | 0.16 | 0.16 | 0.31 | 0.40 | 0.36 | 0.30 | 0.42 | 0.65 | 0.32 | 0.23 |
| pumadyn32nh | **0.02** | 0.02 | 0.02 | 0.02 | 0.02 | 0.02 | 0.02 | 0.02 | 0.02 | 0.03 | 0.37 | 0.35 | 0.02 | 0.02 |
| qsar_aquati... | 0.80 | **0.68** | 0.69 | 0.78 | 0.77 | 0.76 | 0.74 | 0.75 | 0.71 | 0.74 | 2.07 | 1.92 | 0.73 | 0.74 |
| red_wine | 0.44 | 0.42 | **0.39** | 0.43 | 0.49 | 0.45 | 0.58 | 0.53 | 0.52 | 0.49 | 3.09 | 3.37 | 0.54 | 0.46 |
| servo | 0.20 | 0.21 | 0.27 | 0.26 | 0.23 | **0.13** | 0.18 | 0.20 | 0.20 | 0.21 | 1.52 | 1.33 | 0.23 | 0.15 |
| socmob | 5.59 | 5.59 | 5.39 | 6.04 | 6.97 | 3.93 | **3.88** | 5.77 | 4.20 | 4.57 | 5.17 | 5.45 | 4.84 | 3.93 |
| solar_flare | **0.32** | 0.34 | 0.44 | 0.61 | 0.83 | 0.33 | 0.44 | 0.49 | 0.34 | 0.46 | 0.93 | 0.74 | 0.59 | 0.59 |
| space_ga | 0.08 | 0.08 | 0.08 | 0.09 | 0.08 | **0.07** | 0.07 | 0.07 | 0.07 | 0.08 | 0.71 | 0.51 | 0.07 | 0.07 |
| superconduct... | 5.69 | 6.74 | **5.32** | 5.77 | 5.96 | 5.47 | 7.58 | 7.36 | 6.91 | 7.08 | 7.11 | 7.53 | 7.02 | 5.75 |
| white_wine | 0.46 | 0.45 | **0.43** | 0.47 | 0.44 | 0.49 | 0.50 | 0.52 | 0.50 | 0.52 | 2.87 | 3.01 | 0.50 | 0.49 |

## A.10 ANALYSIS OF EXPLICIT ROUTING PATTERNS

To address the effectiveness of the architecture compared to a simple ensemble and the operational mechanics of the routing mechanism, we conducted a quantitative analysis of routing patterns on the test sets of two representative datasets: socmob and Abalone.

### A.10.1 EXTREME SPECIALIZATION IN SPARSE REGIONS (SOCMOB DATASET)

The socmob dataset represents an environment where Few-shot data is extremely sparse. detects this condition and demonstrates **"Specialization"** by assigning a weight of over **0.92** to a specific expert (Expert 0) in the Few-shot region, effectively minimizing the interference of other experts.

Table 17: Analysis on socmob Dataset. (Left) Average Expert Weights. (Right) Dominant Expert Counts.

| Shot Type | Exp 0 (Few) | Exp 1 (Med) | Exp 2 (Many) | Shot Type | Exp 0 (Few) | Exp 1 (Med) | Exp 2 (Many) |
|---|---|---|---|---|---|---|---|
| Few | **0.9228** | 0.0532 | 0.0240 | Few | **43** | 1 | 0 |
| Med | **0.4267** | 0.3514 | 0.2219 | Med | 32 | **33** | 1 |
| Many | 0.0836 | 0.4276 | **0.4888** | Many | 0 | 36 | **86** |

*Note: Bold indicates the highest value, Underline indicates the second highest.*

**Specialization and Interference Blocking:** A major weakness of simple ensembles is that predictions from models biased toward majority classes are mixed into predictions for sparse data. As shown in Table 17, in the Few-shot region, assigns an average weight of 0.9228 to Expert 0 while suppressing others. This demonstrates a **Switching** operation, relying almost exclusively on the specialized expert.

### A.10.2 DYNAMIC COLLABORATION IN CONTINUOUS DISTRIBUTIONS (ABALONE DATASET)

The Abalone dataset exhibits a relatively continuous data distribution. In this context, demonstrates a **"Collaboration"** pattern by appropriately blending the outputs of multiple experts. Unlike a simple average (e.g., 1/3), it identifies the optimal mixing ratio based on data density.

Table 18: Analysis on Abalone Dataset. (Left) Average Expert Weights. (Right) Dominant Expert Counts.

| Shot Type | Exp 0 (Few) | Exp 1 (Med) | Exp 2 (Many) | Shot Type | Exp 0 (Few) | Exp 1 (Med) | Exp 2 (Many) |
|---|---|---|---|---|---|---|---|
| Few | **0.4616** | 0.2625 | 0.2760 | Few | **137** | 8 | 32 |
| Med | 0.3113 | **0.3744** | 0.3143 | Med | 61 | **81** | 41 |
| Many | 0.2769 | 0.3588 | **0.3643** | Many | 95 | 129 | **252** |

**Flexible Collaboration:** Referring to Table 18, while a dominant expert exists for each shot region, the weights are distributed around the 0.3–0.4 range. This implies that performs a **Soft Handoff**, blending expert knowledge in the most appropriate ratio tailored to input characteristics.

### A.10.3 CONCLUSION OF ROUTING ANALYSIS

The expert selection frequencies analyzed above prove that the initial design goal of "density-specific expert mapping" has been realized. The Gating Network distributes traffic to the most suitable expert based on data density without explicit rule-based controls. performs **"Selection"** when data is sparse and **"Collaboration"** when data is complex. This dynamic mechanism is the driving force behind its superior performance compared to simple ensembles.

## A.11 USE OF LARGE LANGUAGE MODELS

In adherence to ICLR 2026 policy, we report the minor use of a large language model as a general-purpose assistant. Its application was strictly limited to the following supportive roles:

- **To aid or polish writing:** Used for minor grammatical corrections and sentence refinement to improve clarity and readability.

- **For retrieval and discovery:** Occasionally used to help summarize or rephrase concepts related to existing work during the literature review process.

The LLM did not contribute to research ideation or the generation of experimental results. The authors take full responsibility for all scientific claims and content presented in this paper.

