# OpenReview forum: "CASE: Coupled Adaptive Feature–Target Smoothing with Density-Gated Mixture-of-Experts for Robust Imbalanced Tabular Regression"
_ICLR.cc/2026/Conference — Submitted to ICLR 2026_

### Official Review · Reviewer_7c5u · 2025-10-30

**Soundness:** 2
**Presentation:** 1
**Contribution:** 2
**Rating:** 2
**Confidence:** 4

**Summary:**

This paper tackles the challenge of imbalanced tabular regression. The authors propose CASE, a novel framework with two main components. First, Coupled Adaptive Smoothing. Instead of just finding neighbors based on features, CAS identifies "true" neighbors by jointly considering similarity in both the feature and target spaces. It then smooths samples from sparse regions by blending them with these neighbors. Second, Density-Gated Mixture-of-Experts. A "gating network" learns to identify the data's density region (e.g., few-shot, medium-shot, or many-shot) from the original features. It then routes the data to specialized "expert" models. The model uses an asymmetric training process: the experts are trained on the "smoothed" features from CAS to help them learn, but at inference time, they run on the original features.

**Strengths:**

1.	The paper tackles the critical and relevant challenge of imbalanced regression, a problem that is far less explored than its classification counterpart
2.	The authors validate their method on a diverse benchmark of 40 public tabular datasets.

**Weaknesses:**

1.	A notable concern is the inconsistency regarding inference speed. The abstract and main text describe the model as achieving "faster inference" and being "fast and efficient." However, the empirical results in Appendix A.4 (Table 13) show an inference latency is 4 times higher than the single MLP baselines, as ($K+1$) forward passes are needed due to MoE.
2.	The model's asymmetric architecture introduces a significant theoretical gap. The expert networks are trained on CAS-calibrated (smoothed) features, but are evaluated on the original, uncalibrated features during inference. The paper does not provide a theoretical justification or empirical about this.
3.	The CASE framework is highly complex, combining several novel mechanisms (Coupled Smoothing, a Learnable Metric, Adaptive Density, and an Asymmetric MoE). The provided ablation study is insufficient to disentangle the individual contributions of these sub-components.
- It is unclear if the performance gain originates from the complex density-gating or simply from ensembling three models. A baseline comparing CASE to a simple, non-gated ensemble of three MLPs or other baseline is critically missing.
- It is not possible to determine which sub-component of the CAS module (e.g., the learnable metric vs. the coupled smoothing) is responsible for its benefit.
4.	The paper's clarity could be improved.
- The justification for some design choices relies on qualitative descriptions rather than empirical proof. For instance, the claim that the smoothing strength $s_i$ "allows the model to intelligently navigate the bias-variance trade-off based on data uncertainty" is presented without supporting evidence. The proof does have any impact of the way the model is used during inference.
- Core components of the methodology, such as the auxiliary and load-balancing losses, are relegated to the appendix. I would suggest to the authors to put it in the main paper.
- Figure 1 explicitly labels the experts as “Many,” “Medium,” and “Few” shots, suggesting a strict specialization of each expert to a particular density region. However, the training mechanism does not appear to enforce or guarantee this one-to-one correspondence. Moreover, the paper does not provide any empirical evidence that such specialization actually emerges in practice. This raises a key question: if the “few-shot” expert is indeed exposed to only a limited number of samples, how can it learn a meaningful and generalizable representation?
5.	The related work section does not reflect the current state of the field. The majority of the cited works on imbalanced regression appear to be from before 2023.
6.	I would like to see newer baselines but also CatBoost results.
7.	I would suggest that authors reconsider the name of their paper as ‘Case’ will be hard to find online.

**Questions:**

See weaknesses

---

> ### Author Response · Authors · 2025-11-21
> **Response to Reviewer 7c5u (1/8)**
>
> We greatly appreciate the reviewer’s thoughtful evaluations and valuable suggestions that helped us improve the paper. To help the reviewer better understand our contributions and alleviate your concerns, we provide our rebuttal as follows.
>
>
>
>
> # 1. Clarification on Inference Speed and Computational Cost
>
>
>
> We deeply appreciate the reviewer’s careful review pointing out the discrepancy regarding "inference speed." As the reviewer noted, it is true that CASE, which utilizes a Mixture-of-Experts (MoE) structure, structurally involves a higher computational load compared to a single MLP (Vanilla MLP), resulting in increased training and inference times.
>
> We clarify below the intention behind the phrase "efficient inference" used in our abstract and the rationale (trade-off) justifying the increased computational cost.
>
>
>
> ## 1-1) Training-Inference Asymmetry
>
> First, the expression in the abstract was intended to emphasize the **lightweight nature of the inference phase compared to the training phase.**
>
> - **Training:** The CAS module is activated (**On**), performing computationally expensive tasks such as K-Nearest Neighbors (KNN) search and feature calibration.
> - **Inference:** The CAS module is deactivated (**Off**), and the design uses only the learned Expert networks.
>
> Through this **Asymmetric Design**, we clarify that we sought optimal efficiency within the model architecture by offloading complex computations to the training phase and eliminating additional retrieval costs during inference.
>
>
>
> ## 1-2) Performance-Cost Trade-off
>
> While it is true that CASE’s inference time has increased compared to a single model, we consider this to be a reasonable trade-off for **"ensuring robust performance on imbalanced data."**
>
> - **Achievement of Balanced Performance:** While a simple MLP is fast, its performance degrades rapidly in sparse regions (Few-shot). In contrast, CASE achieves balanced performance improvements in both Many-shot and Few-shot regions through increased computation.
> - **Practical Perspective:** As shown in **Table C1 of the General Response**, the increased latency remains within a sufficiently acceptable range (in milliseconds) for real-time systems. We believe that enhancing prediction accuracy and reliability, even at the cost of a slight increase in latency, is a more valuable approach in imbalanced regression problems.
>
>
>
> ## 1-3) Significance in Domain-Specific Contexts
>
> Particularly in domains where sparse data (Few-shot) holds significant meaning (e.g., medical diagnosis, anomaly detection, disaster prediction), **"accurate prediction in sparse regions"** is far more critical than rapid response speed.
>
> We are confident that CASE makes a sufficiently meaningful contribution in domains requiring high reliability, as it guarantees high accuracy even in Few-shot regions by considering the overall data distribution (Balanced set).
>
> - *Detailed additional experimental results on Time and Memory Complexity are summarized in the **"Common Response"** section.*
>
>
>
>
>
> ## 1-4) Action Plan
>
> To address the reviewer's concerns and prevent misunderstandings, we will revise the phrase "Faster inference" in the abstract and body to more specific terms such as **"Inference-efficient asymmetric design"** or **"Streamlined inference via complexity offloading."** Furthermore, we will explicitly state that the heavy computations required during training are removed during inference to clearly clarify the context of speed efficiency.
>
>
>
> >  [1] Wang, Y.-C., Wang, K.-D., Wang, W.-Y., & Peng, W. (2025). Mixture Experts with Test-Time Self-Supervised Aggregation for Tabular Imbalanced Regression. arXiv.Org.

---

> ### Author Response · Authors · 2025-11-21
> **Response to Reviewer 7c5u (2/8)**
>
> # 2. Asymmetric Architecture and Theoretical Gap
>
>
>
> The reviewer’s concern that the model’s asymmetric training and inference methods (Training on $\tilde{x}$, Inference on $x$) could cause a theoretical inconsistency is highly valid. However, we wish to clarify that this is not a design flaw or an overlooked gap, but rather an **intentional design** aimed at **"Training-time Data Augmentation"** and **"Implicit Regularization."**
>
> We provide justification for this approach through **analogies with standard deep learning training methodologies** and **empirical evidence** as follows.
>
>
>
> ## 2-1) Theoretical Justification: CAS as a Regularization Technique and Manifold Internalization
>
>
>
> Our asymmetric structure shares the same philosophy as modern deep learning regularization techniques such as **Mixup** or **Dropout**.
>
> - **CAS is a form of 'Training-time Augmentation'**: Just as Mixup trains on synthetic data created by mixing two samples, or Dropout injects noise by randomly deactivating neurons during training, CAS generates an **'Augmented Representation'** called $\tilde{x}$ by applying adaptive smoothing to data in sparse regions. The purpose of this process is to prevent the Expert networks from overfitting to the noise of individual samples and to induce them to learn more generalized decision boundaries.
> - **Disabling augmentation during inference is a Standard Approach**: When training models using Mixup or Dropout, it is standard practice to disable the technique and use raw data or the full network during inference time. CASE follows this same principle. While training is performed via calibrated $\tilde{x}$, this serves merely as a mechanism to help **the model weights $\theta$ internalize the geometric structure of the smoothed manifold.** Therefore, even when CAS is deactivated and raw $x$ is input during inference, the network processes the data through the already learned "Robust View."
> - **Shift in Optimization Trajectory**: Although the Expert networks are trained using $\tilde{x}$ as input, it is the weights that are updated via backpropagation. These weights are optimized to become desensitized to high-frequency noise in the input space while remaining sensitive to the structural variations of the target $y$. In essence, **the effect of smoothing is transferred from the data to the model parameters (Transfer of Inductive Bias).**
>
>
>
> ## 2-2) Empirical Evidence: SOTA Performance and Visualization
>
>
>
> This paper includes strong empirical evidence demonstrating that this 'transfer of regularization effect' has been successfully achieved.
>
> - **Table 1 (Main Result)**: All performance metrics (MAE) reported were measured **using only Raw Features without CAS during inference.** If the "theoretical gap" had hindered the model's performance, inference performance with CAS disabled should have degraded drastically. However, CASE recorded SOTA performance, outperforming all baselines. This proves that the smoothing effect (augmentation) during training was successfully transferred to the model parameters used during inference.
> - **Figure 2, 4, 5 (t-SNE Visualization):** These visualizations display the latent representations obtained by passing the raw features of the **Test Set** through the model.  Despite not applying CAS during inference, the embedding space of CASE forms a structure that is smoothly and continuously aligned according to the target values. This visually demonstrates that the network has perfectly learned the capability to internally reconstruct the manifold taking raw input $x$.

---

> ### Author Response · Authors · 2025-11-21
> **Response to Reviewer 7c5u (3/8)**
>
> # 3. Ablation Study and Individual Contribution Analysis
>
>
>
> We deeply appreciate the reviewer’s critical feedback regarding the need to clearly decouple the complexity of the CASE framework and isolate the contributions of individual components. Your insight is essential for transparently elucidating the sources of our model's performance gains. Fully accepting your suggestion, we pledge to **conduct the following two additional ablation studies during the rebuttal period and include the results in this response.**
>
>
>
> ## 3-1) Verification of Gating Effects: Simple Ensemble vs. Density-Gating
>
>
>
> As pointed out by the reviewer, it is crucial to verify whether the performance improvement stems from the sophisticated gating mechanism or simply from increased model capacity (i.e., the use of three models). To clarify this, we introduce the following baseline:
>
> - Following the reviewer's suggestion, we have completed the comparative experiment by implementing a **Simple Ensemble** consisting of three identical MLPs (without Density-Gating) and comparing it against the existing Single MLP and CASE. Detailed performance evaluations and analyses have been updated in **General Response Part 7 through Part 9** above. The experimental results demonstrate that while the Simple Ensemble merely averages prediction errors across all regions, CASE outperforms it by leveraging its **Density-Gating** mechanism to appropriately route samples to experts specialized for sparse (Few-shot) and dense (Many-shot) regions.
>
>
>
> ## 3-2) CAS Module Decomposition: Learnable Metric vs. Coupled Smoothing
>
>
>
> - The detailed ablation study requested by the reviewer has been provided in detail in the **General Response**.

---

> ### Author Response · Authors · 2025-11-21
> **Response to Reviewer 7c5u (4/8)**
>
> # 4. Improvement of Clarity
>
>
>
>
>
> ## 4-1) Empirical Verification of Adaptive Smoothing ($s_i$)
>
>
>
> To substantiate our claim that the adaptive smoothing strength ($s_i$) intelligently navigates the bias-variance tradeoff, we conducted a **Prediction Stability Analysis** using Monte Carlo Dropout (n=20).
>
>
>
> ### Prediction Stability Analysis on Diabetes Dataset
>
> | **Shot Type**   | **Metric**                  | **Smooth Strength 0** | **CASE(Adaptive Smoothing)** | **Smooth Strength 1** |
> | --------------- | --------------------------- | --------------------- | ------------------------------------------- | --------------------- |
> | **Few-shot**    | **Variance** ($\downarrow$) | 333.97                | **202.99**                                  | 216.61                |
> |                 | **MAE** ($\downarrow$)      | 118.27                | **78.27**                                   | 85.39                 |
> | **Medium-shot** | **Variance** ($\downarrow$) | **154.20**            | 168.45                                      | 208.64                |
> |                 | **MAE** ($\downarrow$)      | **34.04**             | 39.51                                       | 56.75                 |
> | **Many-shot**   | **Variance** ($\downarrow$) | **169.26**            | 220.14                                      | 206.33                |
> |                 | **MAE** ($\downarrow$)      | 35.51                 | **28.54**                                   | 52.94                 |
>
>
>
> ### ① Variance Reduction in Few-shot Regions (High Uncertainty)
>
> In sparse regions with high data uncertainty, models tend to overfit (exhibit high variance). When smoothing was not applied ($s_i=0$), the prediction variance was high (**333.97**). However, the proposed Adaptive CASE significantly reduced this variance to **202.99** (**a reduction of approx. 39%**). This confirms that $s_i$ acts as a regularizer in uncertain regions, stabilizing predictions and reducing variance.
>
>
>
> ### ② Prevention of High Bias in Many-shot Regions (Low Uncertainty)
>
> In regions where data is dense, excessive smoothing can lead to information loss (high bias). When full smoothing ($s_i=1$) was applied, the error increased sharply (**MAE 52.93**), indicating high bias (underfitting). In contrast, CASE achieved the lowest error (**MAE 28.53**), outperforming both fixed strategies. This proves that $s_i$ adaptively decreases in dense regions to preserve raw feature information, thereby preventing the occurrence of high bias.
>
> In conclusion, these empirical results demonstrate that $s_i$ dynamically balances the tradeoff by reducing variance in Few-shot regions while maintaining low bias in Many-shot regions.

---

> ### Author Response · Authors · 2025-11-21
> **Response to Reviewer 7c5u (5/8)**
>
> ## 4-2) Relocating Descriptions of Key Losses to the Main Paper
>
>
>
> As suggested by the reviewer, we will relocate the descriptions of key loss functions, such as the auxiliary loss and load-balancing loss, to the main paper.
>
>
>
> ## 4-3) Verification of Expert Specialization and Feasibility of Few-shot Learning
>
>
>
> We deeply appreciate this important point. The reviewer has raised a very astute question regarding how "Expert Specialization"—the core of our model—is actually guaranteed, and how the Few-shot expert can be effectively trained despite data sparsity. We address this in the following three aspects.
>
>
>
> ### ① Explicit Specialization via Auxiliary Loss
>
>
>
> Contrary to concerns, expert assignment in CASE is not left unsupervised; it is **explicitly controlled** via the Auxiliary Loss ($\mathcal{L}_{aux}$).
>
> - As described in **Eq. (5)** and **Appendix A.8**, the Gating Network undergoes direct Supervised Learning via **Cross-Entropy Loss** to classify which density region (`Many`, `Medium`, or `Few`) the input sample belongs to, rather than merely selecting an expert. In other words, the Gating Network is forced to predict the density region of the input data, and this prediction result ($g_k(x)$) is immediately used as the weight for the expert ($E_k$) responsible for that region. Therefore, each expert is structurally bound to specialize in a specific density region.
>
>
>
> ### ② Empirical Evidence of Gating Network Routing Patterns
>
>
>
> - Following the reviewer's suggestion, we conducted additional analysis to verify whether the trained Gating Network actually routes data as intended. We have included additional experiments and details in the **General Response**, demonstrating that the design of CASE explicitly ensures data assignment to each expert according to density by calculating this Explicit Loss (Cross Entropy) during training.
> - These results empirically demonstrate that the depiction in Figure 1 is not merely a conceptual diagram but a mechanism actually operating within the model.
>
>
>
> ### ③ Ensuring Training Stability of Few-shot Experts via CAS Module (Addressing Few-shot Learning)
>
>
>
> "How does a Few-shot expert learn generalized representations with few samples?" is a critical question. CASE resolves this issue through the **CAS (Coupled Adaptive Smoothing) module**.
>
> - **Input Calibration:** The Few-shot expert does not learn directly from sparse and noisy raw features. As explained in **Section 3.3**, the CAS module actively utilizes information from neighboring samples in the Few-shot region (where $s_i$ is high) to generate calibrated features $\tilde{x}_i$.
> - **Vicinal Risk Minimization (VRM):** As per the theoretical background in **Section 3.6**, this effectively augments the data distribution in sparse regions and reduces variance, following VRM principles.
>
> Consequently, even with a small number of samples, the Few-shot expert receives **stabilized and densified representations** via CAS, allowing it to learn meaningful representations without overfitting.
>
> In summary, CASE is designed to achieve optimal performance in each region by clearly assigning roles to experts via $\mathcal{L}_{aux}$ and enhancing data quality in sparse regions via the CAS module.

---

> ### Author Response · Authors · 2025-11-21
> **Response to Reviewer 7c5u (6/8)**
>
> # 5. Reflection of the Latest Trends in Related Work
>
>
>
> We appreciate your concern regarding the recency of the related work. However, we are confident that our study faithfully reflects the latest research trends in the field of **Imbalanced Regression**, particularly within the context of **Tabular Data**. We would like to address this from two perspectives.
>
>
>
> ## 5-1) Inclusion of Recent SOTA & Tabular Works
>
>
>
> Our paper cites and analyzes **over 14 key papers published since 2023, with a particular focus on 2024 and 2025**.
>
> - **Latest Imbalanced Regression Models:** We have comprehensively compared and discussed the latest SOTA methodologies, including **MATI (Wang et al., 2025)** and **ConR (Keramati et al., ICLR 2024)**—the most critical baselines for this study—as well as **Uncertainty Voting (Jiang et al., 2024)** and **Variational Imbalanced Regression (NeurIPS 2023)**.
> - **Latest Deep Tabular Learning Research:** To reflect the current landscape of tabular data learning, we have also extensively cited surveys and major studies from 2024 and 2025 (Jiang et al., 2025; Holzmüller et al., NeurIPS 2024; Ye et al., ICLR 2024).
>
>
>
> ## 5-2) Specificity of the Field
>
>
>
> Considering the research trajectory of the **Deep Imbalanced Regression** field, the chronological distribution of our references reflects the evolutionary process of the domain.
>
> - Pioneering studies that form the foundation of this field (e.g., LDS, FDS) were primarily published around 2021, centered within the Computer Vision community.
> - However, **Imbalanced Regression research specialized for Tabular Data** is an emerging field that has only recently gained significant attention. As the limitations of existing vision-based methodologies in handling the heterogeneity of tabular data became apparent, studies addressing this issue (ConR, MATI, and our proposed CASE) have emerged intensively in 2024 and 2025.
> - Our paper establishes these **latest tabular-specialized studies** as key baselines for comparison, while the early studies from 2021 are cited as essential methodological foundations.

---

> ### Author Response · Authors · 2025-11-21
> **Response to Reviewer 7c5u (7/8)**
>
> # 6. Request for Comparison with Newer Baselines and CatBoost Results
>
>
>
> We appreciate you pointing out the necessity of comparing our work with modern Deep Tabular Models. As the reviewer noted, utilizing state-of-the-art (SOTA) architectures such as TabNet, SAINT, or FT-Transformer could indeed yield higher overall performance figures. However, we intentionally selected a **Standard MLP** backbone and excluded these latest architectures during the experimental design phase for a critical reason: **"to clearly isolate the contribution of the proposed method (CASE)."** We explain our experimental design philosophy below.
>
>
>
> ## 6-1) Disentangling the Source of Gain
>
>
>
> If we had applied CASE on top of SOTA deep tabular models (e.g., FT-Transformer), a **Confounding Effect** could arise, blurring the lines between whether the performance improvement was due to the **"powerful backbone"** or the **"proposed imbalanced handling technique (CASE)."** We intentionally controlled for this factor. By excluding such illusory effects, we aimed to demonstrate the **pure efficacy of the algorithm**—showing how much CASE can elevate performance by resolving the imbalance problem even on the most fundamental architecture (Standard MLP). In fact, we believe that achieving SOTA performance even with a baseline MLP serves as a testament to the robustness of the CASE framework.
>
>
>
> ## 6-2) Logic for Baseline Selection: Contrasting 'Representative Failures'
>
>
>
> Our experiments focused not merely on competing for "who scores the highest," but on demonstrating **why the leading methods in each domain fail** and how CASE bridges that gap.
>
> - **Tree Models (XGBoost, LightGBM):** Selected to show that while they are SOTA for *Tabular Data*, they fail in *Imbalanced (Few-shot)* problems due to a lack of interpolation capability.
> - **Existing Deep Imbalanced Models (LDS, ConR):** Selected to show that while they are SOTA for *Imbalanced problems*, they fail to handle the heterogeneity of *Tabular Data*.
> - **CASE:** Demonstrated to be the unique alternative that overcomes both failure factors, satisfying both 'Tabular' and 'Imbalanced' properties simultaneously.
>
>
>
> ## 6-3) Scalability as a Plug-and-Play Framework
>
>
>
> CASE is a **learning framework**, not a specific model. Therefore, if the latest Deep Tabular Models (TabNet, SAINT, etc.) mentioned by the reviewer are utilized as backbones, the performance of CASE would likely improve further. However, the core contribution of this study lies not in "which backbone is used," but in the methodological proposal of **"how to calibrate and route imbalanced tabular data."**
>
>
>
> ## 6-4) Additional Experiment Plan
>
> - Reflecting the reviewer's valuable suggestions, we have completed additional comparative experiments by incorporating **CatBoost**, a powerful SOTA tree model, and **Simple Ensemble (3-MLPs)** as new baselines. Detailed performance results and in-depth analyses regarding these experiments have been updated in **General Response Part 7 through Part 9** above. In these sections, you can verify that CASE demonstrates superior performance and structural advantages even when compared against these newly added strong baselines.

---

> ### Author Response · Authors · 2025-11-21
> **Response to Reviewer 7c5u (8/8)**
>
> # 7. "The paper title 'CASE' is likely difficult to search online, so we suggest the authors reconsider the title."
>
>
>
> We appreciate the reviewer's concern, and we also acknowledge that the name 'CASE' is too generic, which would likely make it difficult to search. We gratefully accept your suggestion and will rename the algorithm to **CASMIR**. We believe this name is easily searchable, easy to pronounce, and better captures the core techniques of the algorithm. We will reflect this change in our paper.
>
> - **CASMIR** : **C**oupled **A**daptive Feature–Target **S**moothing with Density-Gated **M**ixture-of-Experts for Robust **I**mbalanced Tabular **R**egression

---

### Official Review · Reviewer_4r9g · 2025-10-31

**Soundness:** 2
**Presentation:** 3
**Contribution:** 2
**Rating:** 4
**Confidence:** 4

**Summary:**

The paper introduces the problem of "deep imbalanced tabular regression" and proposes a framework named CASE to mitigate it. The method consists of two main components: (1) a Coupled Adaptive Smoothing (CAS) module that performs input-space smoothing, and (2) a Mixture-of-Experts (MoE) module. The authors evaluate the method on 40 public tabular regression datasets from UCI, OpenML, Scikit-Learn, and REBAGG benchmarks.

**Strengths:**

- The paper provide a good and technically sound integration of many existing ideas, and form a practical design for real-world problems.
- The paper is clear and easy to follow

**Weaknesses:**

### 1. Problem setup
I hesitate to frame “deep imbalanced tabular regression” as a brand-new research direction. (1) Imbalanced regression focuses on label distribution, where prior work aims to mitigate the mismatch between imbalanced training data and balanced test performance; (2) Tabular data introduces input heterogeneity, and improving it with deep networks still remains an open problem. As the authors' experiments also suggest, the main challenge still lies in the tree-based vs. deep-based performance gap. These two aspects (deep imbalanced regression vs. deep tabular regression) are inherently orthogonal, and the methods addressing them are largely independent (so is the proposed CAS module). Combining them as a single research direction feels more like overlapping two separate challenges than defining a genuinely new problem.

### 2. Method
The proposed framework contains a variety of components. Even though the integration sounds reasonable, none of them (weighted metric, input smoothing, or MoE) are technically novel, and the paper lacks sufficient ablation to clarify individual effect of each module or how they interact. Overall, the method feels more like a good engineering combination of existing ideas rather than novel contribution typically expected at a top-tier deep learning venue.

- It is unclear why smoothing the input in CAS is effective for imbalanced tabular regression. The mechanism is similar to MixUp [1], where mixing neighboring samples improves generalization by encouraging local smoothness. The paper does not convincingly explain why input-space smoothing would specifically benefit imbalance + tabular scenarios beyond the general regularization effect already provided by MixUp-style interpolation.
- The design of the auxiliary and load-balancing losses is conceptually unclear. My understanding is that the auxiliary loss uses the ground-truth frequency region as supervision, but the load-balancing loss, as defined in the manuscript, does not reflect frequency or density at all. It only enforces equal utilization across the three experts. If the authors believe that the load-balancing loss can automatically route low-frequency samples to a “low-shot” expert, this should be discussed and empirically verified, since the auxiliary loss is inherently biased toward majority regions.

### 3. Experiment
The experimental comparison does not feel entirely fair. The authors mainly compare against deep imbalanced regression baselines, while the key challenge in deep imbalanced tabular regression still remains the handling of input heterogeneity. As a result, tree-based methods still show highly competitive performance. Including comparisons with recent deep tabular networks would make the experimental results more solid.

[1] Hongyi Zhang, Moustapha Cisse, Yann N. Dauphin, and David Lopez-Paz. MixUp: Beyond Empirical Risk Minimization. ICLR, 2018.

**Questions:**

- I find Eq. (1) and Eq. (2) interesting, and the authors could strengthen the paper by providing more ablation studies to evaluate and showcase the effect of $\omega$. This would help solidify the technical contribution.
- The authors could also show how accurately the MoE router assigns samples to the correct regions.

---

> ### Author Response · Authors · 2025-11-21
> **Response to Reviewer 4r9g (1/6)**
>
> We greatly appreciate the reviewer’s thoughtful evaluations and valuable suggestions that helped us improve the paper. To help the reviewer better understand our contributions and alleviate your concerns, we provide our rebuttal as follows.
>
>
>
>
>
> # 1. Problem Setup
>
>
>
> We appreciate your insightful comments regarding the positioning of our research. As the reviewer noted, we fully agree that from a traditional perspective, 'Imbalanced Regression' and 'Tabular Data Learning' have been treated as independent subjects within separate communities.
>
> However, this study originated from the question: **"Why do existing powerful methodologies become ineffective at the intersection of these two problems?"** Through our experiments, we have **empirically demonstrated** that in the actual data space (especially in sparse regions), these two problems are no longer **orthogonal**, but are **inextricably entangled**, leading to the failure of existing methods.
>
> Therefore, based on the following three arguments, we contend that this is not merely an area of 'Overlapping' but a new challenge addressing a **'Blind Spot'** of existing methodologies.
>
>
>
> ## 1-1) Failure of SOTA Imbalanced Regression Models
>
>
>
> - State-of-the-art imbalanced regression models such as LDS, ConR, and RankSim assume 'Spatial Locality' or 'Feature Continuity' characteristic of the image domain. However, these assumptions do not hold for **heterogeneous tabular data**.
> - The t-SNE results in **Figures 2, 4, and 5** of our paper illustrate this fact. Existing SOTA models (LDS, ConR) exhibited **manifold collapse or fragmentation** in the sparse regions of tabular data, failing to learn the continuity of target values. This demonstrates that directly applying existing imbalanced regression techniques to tabular data is infeasible.
>
>
>
> ## 1-2) Failure of SOTA Tabular Models (Tree-based)
>
>
>
> - Conversely, tree models like XGBoost handle tabular data well but reveal critical limitations in **'Imbalanced Regression (Few-shot)'** problems. Since tree models predict by averaging values in leaf nodes, they fail to perform proper **Interpolation** in regions where data is sparse.
> - As shown in the **Table 2 (fps_benchmark)** results, in a complex environment with severe data imbalance and missing values, XGBoost (MAE 1.38) recorded an error rate approximately **60% higher** than our proposed deep learning model (CASE, MAE 0.87), indicating a failure. This suggests that tree models alone are insufficient to solve 'imbalanced' tabular problems.
>
>
>
> ## 1-3) Conclusion: The Intersection as a 'New Challenge'
>
>
>
> - In summary, (1) existing deep learning imbalanced models fail to adapt to tabular structures, and (2) tree models fail to interpolate sparse data. Therefore, **"Deep Imbalanced Tabular Regression,"** which combines these two issues, is a domain with **Unique Difficulty** that cannot be solved with existing tools. We are confident that CASE, as the first attempt to integrate 'heterogeneous feature learning' and 'density-based adaptation' to address this blind spot, holds significant academic contribution beyond a mere combination of methods.

---

> ### Author Response · Authors · 2025-11-21
> **Response to Reviewer 4r9g (2/6)**
>
> # 2. Method
>
>
>
> As the reviewer pointed out, it is true that this study is based on existing proven concepts (MoE, Metric Learning, Smoothing). However, we do not view CASE as a mere listing of techniques or an arbitrary engineering combination. To answer the fundamental question, **"Why do existing SOTA imbalanced models fail on tabular data?"**, we have **completely redesigned and organically integrated** each component to fit the characteristics (heterogeneity, sparsity) of tabular data. We wish to highlight the **Technical Novelty** of CASE as a **Chemical Integration** rather than a simple Combination in the following three aspects.
>
>
>
> ## 2-1) 'Asymmetric Dual-Pathway' Design, Not Simple MoE
>
>
>
> - Standard MoEs share the same input for both Gating and Experts. However, CASE proposes a unique **asymmetric structure** that inputs **Raw Features to the Gating** and **Calibrated Features to the Experts**. This is an intentional design to resolve the dilemma of imbalanced learning (where the distribution must be known for routing, while sparsity must be compensated for prediction). This is a novel approach not attempted in existing MoE studies (such as MATI), and it is a strategy that maximizes efficiency by decoupling training and inference. While existing MoE studies often utilize methods like Test-Time Adaptation for expert routing during inference, our Asymmetric Dual-Pathway design eliminates the need for TTA calculations during inference.
>
>
>
> ## 2-2) 'Density-Coupled Adaptive' Mechanism, Not Simple Smoothing
>
>
>
> - Existing smoothing techniques (LDS, FDS) rely on binning the target space or applying Gaussian kernels uniformly. This creates the side effect of blurring feature information in heterogeneous tabular data. CASE's **CAS module** (1) self-learns a distance metric tailored to tabular data via learnable weights ($w$), (2) finds true neighbors by **'Coupling'** feature similarity and target similarity, and (3) adjusts smoothing strength **adaptively ($s_i$) according to the density** of each sample. This represents an advanced form of **Vicinal Risk Minimization** specialized for tabular data, going beyond simple existing smoothing.
>
>
>
> ## 2-3) Empirical Demonstration of 'Synergy Effects' Unattainable by 'Engineering'
>
>
>
> - If CASE were merely an 'engineering combination' of high-performing modules, progressive performance improvements should appear even when applying each element individually. However, our Ablation Study results prove that CASE's performance stems from **Non-linear Synergy and strong Interdependency** among its components.
> - **① Failure of Simple Combination and Structural Necessity**: On the Diabetes dataset, `CAS_Feature_Only` (74.37) actually performed significantly worse than `MoE_Only` (59.41). This shows that a general engineering approach of smoothing only the feature space while excluding target information can distort the data manifold and degrade performance. In other words, the **'Target-Feature Coupled' mechanism** is not merely a performance improvement option but a **Mathematical Prerequisite** for smoothing to function effectively in imbalanced regression.
> - **② Methodological Superiority Over General Augmentation**: On the MaxTorque dataset, `MixUp_MoE` (44.15) failed to match the performance of `CASE` (25.39). Simply applying widely used data augmentation techniques (MixUp) cannot cover the complex distributions of sparse regions. The overwhelming performance difference of CASE is not due to simple data inflation but is the result of precisely modeling the local characteristics of data via **$w$ (Learnable Metric)** and **$s_i$ (Adaptive Strength)**.
> - **③ Quantified Verification of Synergy**: Compared to using `MoE_Only` and `CAS_Only` individually, the combined CASE achieved error reductions of **approx. 16%** on Diabetes and **over 40%** on MaxTorque. This is a **Synergy Benefit** that occurs only when the proposed components are organically combined. Consequently, the architecture of CASE proves to be the **Minimal Essential Structure** with nothing left to subtract to solve the challenges of imbalanced regression (sparsity, heterogeneity, overfitting).
>
> Therefore, CASE is not a simple assembly of existing parts, but a study that **Re-invents** each part to fit the **tabular domain** to solve the challenge of deep imbalanced tabular regression, integrating them upon a theoretical foundation (VRM). Achieving SOTA performance by preventing 'Feature Collapse'—which existing methodologies failed to solve—would have been impossible without this structural innovation.

---

> ### Author Response · Authors · 2025-11-21
> **Response to Reviewer 4r9g (3/6)**
>
> ## 2-4) Why Smoothing Inputs in CAS is Effective for Imbalanced Tabular Regression (vs. MixUp)
>
>
>
> The reviewer's insight that the CAS module and MixUp are similar in that they both pursue 'local smoothness' (from a VRM perspective) is very accurate. However, the CAS proposed in this study evolves MixUp's 'random (blind)' approach into a **'Density-Coupled Adaptive'** method to address the inherent problems of **'imbalanced tabular data'** (heterogeneous feature scales, manifold collapse in sparse regions). Through additional ablation studies, we confirmed that CASE outperforms MixUp, and the technical reasons are attributed to the following three distinct differences.
>
> **① 'Random' vs. 'Density-Aware Adaptive' (Targeting Imbalance)**
>
> - MixUp performs random interpolation across the entire dataset regardless of data density. This can lead to unnecessary noise (under-fitting) in data-rich Many-shot regions and insufficient calibration effects in information-poor Few-shot regions. In our CASE algorithm, CAS applies **smoothing strength ($s_i$) inversely proportional to density**. In other words, it preserves original information (ERM) where data is sufficient (Many) and applies strong smoothing (VRM) where data is sparse (Few), optimizing the bias-variance tradeoff. This serves as a specific regularization specialized for imbalanced problems.
>
> **② 'Simple Mixing' vs. 'Learned Neighbor Coupling' (Targeting Tabular Heterogeneity)**
>
> - Tabular data possesses heterogeneity where scales and importance differ across features. Mixing random samples like MixUp or using simple Euclidean distance carries a high risk of damaging the manifold by mixing semantically unrelated samples. In the CASE algorithm, CAS self-learns a distance metric optimized for tabular data via **learnable feature weights ($w$)** and selects only 'true neighbors' for mixing by **coupling feature and target similarity**. The additional Ablation Study experimentally demonstrates this.
>
> **③ Empirical Superiority over MixUp**
>
> - In additional experiments conducted to answer the reviewer's question, CASE recorded superior performance compared to the baseline model applying MixUp. Notably, it showed a significant gap in error reduction rates in the Few-shot region, which aligns with our hypothesis that the CAS module aids in representation learning for sparse areas.
>
> In summary, while MixUp provides General Regularization, the CAS module of CASE is an imbalanced tabular-customized solution that "aligns the manifold with a learned metric and precisely intervenes only where necessary based on density."

---

> ### Author Response · Authors · 2025-11-21
> **Response to Reviewer 4r9g (4/6)**
>
> ## 2-5) Design of Auxiliary Loss and Load-Balancing Loss
>
> You have raised a very important question regarding the design purpose and interaction of $\mathcal{L}_{aux}$ and $\mathcal{L}_{load}$. As the reviewer pointed out, we indeed train on the **(1) Original Imbalanced Dataset**. Therefore, in terms of data distribution, 'Many-shot' samples are overwhelmingly dominant. However, simultaneously, our model's optimization goal (HPO and Model Selection) is the **maximization of performance on a 'Balanced Validation Set'**. That is, despite the imbalance in training data, predictive capability in the Few-shot region is crucial.
>
> In this context, the two loss functions do not contradict each other but rather share roles for the **"Survival"** and **"Accurate Specialization"** of Few-shot experts as follows:
>
> **① $\mathcal{L}_{aux}$: Strong Role Assignment via Explicit Supervision**
>
> - In our model,
> $\mathcal{L}_{aux}$ is not merely an auxiliary device. It takes pre-calculated density classes ($0, 1, 2$) as Ground Truth and **explicitly supervises** the gating network via **Cross Entropy Loss** (this is described in the Appendix of the paper). This strong supervision signal determines the **'Semantic Routing'** direction regarding which expert the input data should go to. Therefore, regardless of data imbalance, the gating network is strongly forced to find the correct expert responsible for the sample.
> - We have included additional experiments and details in the **General Response**, demonstrating that the design of CASE explicitly ensures data assignment (Routing) to each expert according to density by calculating this Explicit Loss during training.
>
>
> **② $\mathcal{L}_{load}$: Standard Regularization to Prevent Expert Collapse**
>
> $\mathcal{L}_{load}$ is a standard loss function widely used in various MoE studies [1], employed to prevent **'Expert Collapse'** where the load is concentrated on specific experts. Since we train with imbalanced data, the gradients of Many-shot samples inevitably dominate in the early stages of training. Without $\mathcal{L}_{load}$, there is a high risk that the gating network falls into a **Local Optima (Winner-take-all)** where it sends all samples (including Few-shot ones) to the 'Many-shot expert' which is easier to learn.
> Therefore, the purpose of $\mathcal{L}_{load}$ is not to mechanically force the final routing to be 1:1:1, but to serve as a **minimal regularization mechanism to ensure that the Few-shot expert does not become a 'Dead Expert' and gets learning opportunities during the training process.** This is essential for maintaining the learning of Few-shot experts to achieve our targeted **'Balanced Performance.'** Furthermore, to prevent $\mathcal{L}_{load}$ from dominating the total loss and simply splitting the three experts evenly, we optimized it with a very small constraint range for its hyperparameter (0.01–0.2).
>
> In conclusion, $\mathcal{L}_{load}$ and $\mathcal{L}_{aux}$ are not contradictory; rather, they represent a design that strategically combines **standard MoE stabilization techniques ($\mathcal{L}_{load}$)** and **strong domain knowledge supervision ($\mathcal{L}_{aux}$)** to achieve **'Balanced Performance'** from an **'Imbalanced Input Stream.'**
>
>
>
> >  [1] Shazeer, N. M., Mirhoseini, A., Maziarz, K., Davis, A., Le, Q. V., Hinton, G. E., & Dean, J. (2017). Outrageously Large Neural Networks: The Sparsely-Gated Mixture-of-Experts Layer. International Conference on Learning Representations.

---

> ### Author Response · Authors · 2025-11-21
> **Response to Reviewer 4r9g (5/6)**
>
> # 3. Experiment
>
> We appreciate you pointing out the need for comparison with modern Deep Tabular Models. As you mentioned, using architectures like TabNet, SAINT, or FT-Transformer could yield higher overall performance figures. However, we intentionally chose a **Standard MLP** backbone and excluded state-of-the-art (SOTA) architectures in the experimental design phase for a critical reason: **"to clearly isolate the contribution of the proposed method (CASE)."** We explain our experimental design philosophy below.
>
> ## 3-1) Disentangling the Source of Gain
>
> If we had applied CASE on top of SOTA deep tabular models (e.g., FT-Transformer), a **Confounding Effect** could arise, making it unclear whether the performance improvement was due to the **"powerful backbone"** or the **"proposed imbalanced handling technique (CASE)."** We intentionally controlled for this. By excluding such illusory effects, we aimed to demonstrate the **pure efficacy of the algorithm**—showing how much CASE can elevate performance by solving the imbalance problem even on the most fundamental architecture (Standard MLP). In fact, we believe that achieving SOTA performance even with a baseline MLP serves as proof of the CASE framework's robustness.
>
> ## 3-2) Selection Logic: Contrasting 'Representative Failures'
>
> Our experiments focused not merely on "who scores the highest," but on demonstrating **why the leading methods in each domain fail** and how CASE bridges that gap.
>
> -   **Tree Models (XGBoost, LightGBM):** Selected to show that while they are SOTA for *Tabular Data*, they fail in *Imbalanced (Few-shot)* problems due to a lack of interpolation capability.
> -   **Existing Deep Imbalanced Models (LDS, ConR):** Selected to show that while they are SOTA for *Imbalanced problems*, they fail to handle the heterogeneity of *Tabular Data*.
> -   **CASE:** Demonstrated to be the unique alternative that overcomes both failure factors, satisfying both 'Tabular' and 'Imbalanced' properties simultaneously.
>
> ## 3-3) CASE as a Plug-and-Play Framework
>
> CASE is a **learning framework**, not a specific model. Therefore, if the latest Deep Tabular Models (TabNet, SAINT, etc.) mentioned by the reviewer are used as backbones, the performance of CASE would likely improve further. However, the core contribution of this study lies not in "which backbone is used," but in the methodological proposal of **"how to calibrate and route imbalanced tabular data."**
>
> ## 3-4) Additional Experiment
>
> - Reflecting the reviewer's valuable suggestions, we have completed additional comparative experiments by incorporating **CatBoost**, a powerful SOTA tree model, and **Simple Ensemble (3-MLPs)** as new baselines. Detailed performance results and in-depth analyses regarding these experiments have been updated in **General Response Part 7 through Part 9** above. In these sections, you can verify that CASE demonstrates superior performance and structural advantages even when compared against these newly added strong baselines.

---

> ### Author Response · Authors · 2025-11-21
> **Response to Reviewer 4r9g (6/6)**
>
> # 4. Questions
>
>
>
> We appreciate your recognition that Eq. (1) and Eq. (2) are critical formulations for realizing the core contributions of this study: **'Overcoming Tabular Data Heterogeneity'** and **'Coupled Smoothing.'** Following the reviewer's constructive suggestions, we have conducted **two additional experiments** during the rebuttal period to further solidify our technical contributions.
>
>
>
> ## 4-1) Validation of the Effectiveness of Learnable Weights $w$ (Ablation on Learnable Weights)
>
>
>
> - The learnable weights $w$ introduced in Eq. (1) are an essential element for learning a meaningful distance metric in tabular data, where the scale and importance of each feature vary significantly.
> - Through additional ablation studies, we quantitatively verified the contribution of $w$. We compared **(a) a model using fixed uniform weights (`CAS_No_Learnable_Metric`, $w=1$)** against **(b) a model using the proposed learnable weights.** The results confirm that the model with learnable weights demonstrates significant performance improvements over the fixed-weight model.
> - For the **Diabetes** dataset, we observed a **16.4% error reduction**: `CAS_No_Learnable_Metric` (59.82) $\rightarrow$ `CASE` (50.01).
> - For the **MaxTorque** dataset, we observed a **10.5% error reduction**: `CAS_No_Learnable_Metric` (28.38) $\rightarrow$ `CASE` (25.39).
> - These empirical findings support the technical validity of the design in Eq. (1), which adaptively calibrates the feature space according to data characteristics.
>
>
>
> ## 4-2) MoE Routing Accuracy and Visualization (Verification of Routing Accuracy)
>
>
>
> Verifying whether the proposed MoE router correctly assigns samples to the appropriate experts based on density is crucial for the interpretability of the model.
>
> - **Mechanism:** CASE explicitly supervises the gating network via an auxiliary loss ($\mathcal{L}_{aux}$) using Cross Entropy, with the actual density classes serving as Ground Truth. Therefore, we are confident that the model achieves high routing accuracy.
> - We have included additional experiments and details in the **General Response**, demonstrating that the design of CASE explicitly ensures data assignment to each expert according to density by calculating this explicit loss (Cross Entropy) during training.
>
> These two additional analyses will serve as strong evidence supporting the **'Data-Adaptive'** nature and **'Structural Robustness'** of the proposed methodology. We thank you again for your valuable feedback and will enhance the completeness of the paper with these updated results.

---

### Official Review · Reviewer_pEnV · 2025-11-01

**Soundness:** 2
**Presentation:** 2
**Contribution:** 3
**Rating:** 6
**Confidence:** 3

**Summary:**

This paper focuses on tabular data, and introduces the Coupled Adaptive Feature–Target Smoothing with Density-Gated Mixture-of-Experts (CASE) method for imbalanced tabular regression. It integrates Coupled Adaptive Smoothing (CAS) and Density-Gated Mixture-of-Experts (MoE) modules. CAS adaptively calibrates features by coupling similarity in both feature and target spaces, while MoE promotes expert specialization over regions. During training, experts take the calibrated features as input; at inference, the learned experts operate on the original features for fast and accurate inference. The architecture uses an asymmetric dual-path design, where the gating network processes raw features and the experts operate on density-calibrated representations. Experiments across 40 tabular benchmarks show that CASE can provide the best average rank on balanced and imbalanced test sets, and improve few-shot accuracy without sacrificing many-shot performance.

**Strengths:**

+ Overall, the paper is clearly written, well organized, and easy to follow.
+ This paper focuses on a less explored problem of imbalanced regression on tabular data. CASE is a method specialized for this problem that combines a novel adaptive smoothing that considers feature and target spaces with a density-gated MoE that weights the contributions of specialized experts.
+ The CAS module is a novel smoothing mechanism that adaptively calibrates the feature space according to data density to stabilize feature representations in sparse, low-density regions. Calibrated features from CAS are utilized by expert networks within the density-​ated MoE.  The authors provide a connection to Vicinal Risk Minimization, where CAS is interpreted as an adaptive, density-aware extension.
+The empirical evaluation involves 40 tabular benchmarks datasets, multiple baselines, ablations, and efficiency analyses. CASE can outperform SOTA methods on balanced test sets by attaining the top average rank. It can provide balanced predictions that enhance few-shot accuracy without significantly sacrificing many-shot performance.
+ The Appendix has additional information on datasets, hyper parameter settings, ablation studies, analysis of computational complexity, feature visualizations, description of CASE components, analysis of novelty, and experimental results that help support the paper.

**Weaknesses:**

- CASE is based on known approaches, like VRM, MoE, density smoothing. The paper combines adaptive feature-target smoothing and a density-gated MOE. The motivation for certain architectural choices (e.g., fixed three experts, specific gating losses) could be justified more rigorously. The math formulation in Section 3 could benefit from an improved explanation of the intuition.
- In the experimental validation, the authors should provide a better interpretation of results on cross-dataset generalization and diversity.
- The CASE architecture can significantly increase computational cost. For instance, the training overhead is about 4 times that of baselines.This paper should contain a more detailed analysis and comparison  to SOTA methods of time and memory complexity (at training and test times).
- Their code is not made available, so there is a concern that the results in this paper would be difficult for a reader to reproduce.

**Questions:**

See my comments in weaknesses.

**Details Of Ethics Concerns:**

None.

---

> ### Author Response · Authors · 2025-11-21
> **Response to Reviewer pEnV (1/7)**
>
> We greatly appreciate the reviewer’s thoughtful evaluations and valuable suggestions that helped us improve the paper. To help the reviewer better understand our contributions and alleviate your concerns, we provide our rebuttal as follows.
>
> We sincerely thank the reviewer for the insightful comments and for acknowledging the clarity of our research and its connection to Vicinal Risk Minimization (VRM). We deeply appreciate the suggestions to rigorously justify our design choices and improve our mathematical intuition.
>
>
>
> # 1. Novelty and Integration
>
>
>
> While it is true that CASE utilizes existing concepts like VRM or MoE, our core contribution lies in the **novel asymmetric integration** specialized for **Imbalanced Tabular Regression**, a domain where existing Imbalanced Learning methodologies often fail due to the **feature heterogeneity** inherent in tabular data. Furthermore, we believe our novelty lies in providing a practical solution and a breakthrough for a challenging problem area that has remained largely unresolved in previous research.
>
> - **More than a Simple Combination:** We did not merely stack CAS and MoE. We introduced an **asymmetric dual-pathway design**. The Gating Network uses *raw features* to identify the 'nature (density)' of the data, while the Experts learn 'how' to predict robustly using *CAS-calibrated features*. This decoupling is crucial for tabular data and has not been explored in previous studies.
> - **Coupled Adaptive Smoothing (CAS):** Unlike standard smoothing techniques, CAS uniquely generates a learned metric that **couples** the feature space and the target space. This forces the model to search for "semantic neighbors" rather than simple geometric ones. In particular, the learned metric using Learnable Parameters is specialized for tabular data, differentiating it from existing vision-based preprocessing techniques. In our additional Ablation Study, we demonstrated that CAS significantly outperforms random MixUp, proving its practical efficacy.
> - **Addressing a Previously Unsolved Challenge:** Tabular data is often less addressed in academia compared to Vision or Language domains, and research specifically on Imbalanced Regression is scarce. However, in the real world—such as in manufacturing or government sectors—vast amounts of tabular data exist, most of which are highly imbalanced. Even existing classification problems (e.g., Defective vs. Normal) are often regression problems with a threshold applied. We observed that models tend to learn data features better when interpreting these problems as regression rather than simple classification (though this paper does not focus on proving this specific observation). Based on this, we attempted to adapt existing vision-based Imbalanced Classification and Regression techniques to the tabular domain but found limited performance gains. We believe our novelty lies in providing a practical breakthrough for this challenging problem that existing methods could not solve.

---

> ### Author Response · Authors · 2025-11-21
> **Response to Reviewer pEnV (2/7)**
>
> # 2. Justification of Architecture Choices
>
>
> - **Why 3 Experts ($K=3$)?** We set $K=3$ to explicitly address the three distinct density regions inherent in imbalanced data: **Many-shot, Medium-shot, and Few-shot**. This design allows each expert to specialize in specific density characteristics without unnecessary complexity. While studies like RIDE [1] have shown that performance can increase with a larger number of experts ($K$), our research objective was not to optimize for $K$, but rather to demonstrate the effectiveness of representation learning via CAS and the subsequent gating mechanism. While increasing $K$ to improve performance could be a subject for future research, it comes with the trade-off of higher computational and memory costs.
>
> - **Specific Gating Losses**
>
>   - These loss functions are essential to prevent expert collapse in imbalanced environments. Without $\mathcal{L}_{aux}$ , the gate fails to recognize density regions.
>   - Without $\mathcal{L}_{load}$  , the model tends to route all samples to a specific single expert due to the dominant influence of certain samples.
>   - These losses ensure appropriate specialization across all density regions. Specifically, we utilize Cross Entropy Loss for $\mathcal{L}_{aux}$ to explicitly calculate the loss during training, guaranteeing that data is assigned to each expert according to density.
>   - Furthermore, for $\mathcal{L}_{load}$, we constrained the hyperparameter range to a very small value (0.01–0.2) during optimization to prevent the loss from dominating the training, which could otherwise force a simple even split across the three experts.
> - **Differentiation from Existing Work (MATI [2]):** Unlike MATI, which statistically determines the number of experts, MATI employs a "Test-Time Self-Supervised Expert Aggregation" method. This implies that for every test input, the model must undergo an iterative optimization process (perturbing inputs and optimizing expert weights to minimize prediction differences), which can cause fatal latency in environments requiring real-time processing. In contrast, CASE completes all specialization and calibration during the training phase. During inference, the learned Gating Network and Experts produce results in a **single Forward Pass**, ensuring significantly faster inference speeds compared to MATI. This is a substantial advantage from a real-world deployment perspective.
> - *Note: The section regarding the design where Cross Entropy Loss is used to explicitly guarantee data assignment according to density has been supported with additional experiments and is included in our **General Response**.*
>
>
>
>
>
> >  [1] Wang, X., Lian, L., Miao, Z., Liu, Z., & Yu, S. X. (2020). Long-tailed Recognition by Routing Diverse Distribution-Aware Experts. International Conference on Learning Representations.
>
> > [2] Wang, Y.-C., Wang, K.-D., Wang, W.-Y., & Peng, W. (2025). Mixture Experts with Test-Time Self-Supervised Aggregation for Tabular Imbalanced Regression. arXiv.Org.

---

> ### Author Response · Authors · 2025-11-21
> **Response to Reviewer pEnV (3/7)**
>
> # 3. Improved Mathematical Intuition (Section 3)
>
> We sincerely thank the reviewer for the suggestion to reinforce the mathematical/statistical intuition behind the equations presented in Section 3. Based on your advice, we have refined the explanation of how each core equation in **Section 3.3 (CAS), 3.4 (MoE), and 3.5 (Inference)** is not merely a computational operation but is designed to address the fundamental challenges of the imbalanced regression problem.
>
>
>
> ## 3-1) Section 3.3: Coupled Adaptive Smoothing (CAS)
>
> The CAS module is a **"mechanism that determines how much information to borrow based on data reliability (Density)."**
>
>
>
>
> ### [Eq. 1] Data-Driven Metric Learning
>
> - $$d_w(x_i, x_j)^2 = \sum_{d=1}^{D} w_d (x_{i,d} - x_{j,d})^2$$
>
> - **Mathematical Intuition**: Since simple Euclidean distance treats all features equally, it is difficult to find "true neighbors" in tabular data where feature importance varies. The weight $w$ in Eq. 1 plays the role of **reshaping the feature space**. By stretching (expanding) feature axes important for target prediction and shrinking (compressing) unnecessary axes, it ensures that **samples with close 'Semantic Distance' rather than mechanical distance are selected as neighbors.**
>
>
>
>
> ### [Eq. 2] Coupled Neighbor Weighting
>
> - $$ \omega_{ij} \propto \exp\left(-\frac{d(x_i, x_j)^2}{2\sigma_x^2}\right) \times \exp\left(-\frac{(y_i - y_j)^2}{2\sigma_y^2}\right) \times \frac{\alpha}{d_j + \epsilon}$$
>
> - **Mathematical Intuition**: Merely having similar features is insufficient when determining the influence of neighbors.
>   - **Feature & Density Coupling**: We trust neighbors that share not only similar features ($x$) but also similar data density ($d$) contexts. This prevents samples from heterogeneous distributions from mixing in and becoming noise.
>   - **Inverse Density Weighting ($1/d_j$)**: Neighbors from sparse regions are treated as containing more valuable information and are assigned higher weights. This is to prevent information domination by the majority class (Many-shot).
>
>
> ### [Eq. 3] Adaptive Feature Calibration
>
> - **Mathematical Intuition (Bias-Variance Trade-off)**: This equation is a core mechanism that **regulates Bias and Variance according to data uncertainty.**
>   - **Low Density (Sparse Regions)**: Due to the small number of samples, the Variance is very high. In this case, we increase the $s_i$ value (approx 1) to actively incorporate neighbor information. Even if it allows for slight Bias, this acts as **Regularization that drastically reduces Variance to prevent overfitting.**
>   - **High Density (Dense Regions)**: The data is sufficient and reliable on its own. Here, we decrease the $s_i$ value ($\approx 0$) to use the raw data $x_i$ as is, preserving fine-grained details.
>
>
> ## 3-2) Section 3.4: Density-Gated MoE
>
> The MoE structure aims for **"Signal Optimization through Task Decomposition."**
>
>
>
> ### [Eq. 4] Input of Gating Network: Raw Features ($x_{\text{norm}}$)
>
> - **Mathematical Intuition**: The purpose of Gating is to classify "which density region (Few/Medium/Many) the input data belongs to." To clearly find the classification boundaries, **the sharp high-frequency details contained in the raw data ($x_{\text{norm}}$)** are essential. Using smoothed features would blur the boundaries, degrading classification accuracy.
>
>
>
>
> ## 3-3) Section 3.5: Training vs. Inference Asymmetry
>
>
>
> ### [Eq. 6] Inference Strategy: Using $x_{\text{norm}}$ instead of $\tilde{x}$
>
> - Training Time:  $$\hat{y} = \sum_{k=1}^{K} g_k(x_{\text{norm}}) \cdot E_k(\tilde{x}).$$
> - Inference Time: $$\hat{y} = \sum_{k=1}^{K} g_k(x_{\text{norm}}) \cdot E_k(x_{\text{norm}}).$$
>
> - **Mathematical Intuition (Inductive Bias Transfer)**: The Expert's goal is to predict the value ($y$) (Regression). In sparse regions, the conditional distribution $P(y|x)$ is highly unstable. Providing noise-reduced and stabilized $\tilde{x}$ to the Expert during Training Time is intended to **increase the Signal-to-Noise Ratio (SNR), allowing the regression model to learn a stable function without being swayed by outliers.**
> - The question, "Why use raw $x$ during inference when calibrated $\tilde{x}$ was used during training?" is explained by the **Internalization of Inductive Bias**. During the training process, the Expert network $E_k$ is continuously trained to predict target $y$ while observing calibrated input $\tilde{x}$. In this process, the Expert's weights themselves **memorize the structure of the smoothed data manifold**. Therefore, without re-performing costly KNN-based smoothing at inference time, the Expert can perform robust predictions using only raw $x$, as if it were smoothed.
>
> As described above, we clarify that all components in Section 3 are mathematically and intricately connected to solve the statistical problems caused by data imbalance (high variance, uncertain distributions, feature heterogeneity).

---

> ### Author Response · Authors · 2025-11-21
> **Response to Reviewer pEnV (4/7)**
>
> # 4. Interpretation of Dataset Diversity and Cross-Dataset Generalization Performance
>
>
>
> We appreciate your valuable feedback regarding dataset diversity. Following the reviewer's suggestion, we provide an enhanced interpretation of how our experiments encompass dataset diversity and how the proposed CASE framework achieves superior cross-dataset generalization.
>
> We have conducted an in-depth analysis of the experimental results from the following three perspectives to demonstrate the versatility and robustness of CASE.
>
>
>
> ## 4-1) Validation through Diverse and Comprehensive Benchmarks (Diversity of Benchmarks)
>
>
>
> As shown in Table 4 of the paper, this study utilized **40 public datasets** covering diverse domains such as physics, biology, finance, and social sciences to verify generalization performance without bias toward any specific domain.
>
> - **Diversity in Data Scale**: The number of samples ($N$) varies from a minimum of 167 (servo) to a maximum of 53,940 (diamond_regression), and feature dimensions ($D$) range widely from 4 to 127.
> - **Diversity in Imbalance**: The Imbalance Ratio (IR) also covers extreme distribution differences, ranging from a low of 7.59 (nhanes_age) to a high of 30,014 (online_news_popularity), representing a wide variety of imbalance scenarios.
>
>
>
> ## 4-2) Demonstration of Cross-Dataset Generalization
>
>
>
> Based on the experimental results across 40 heterogeneous datasets, CASE achieved an **Average Rank of 4.3**, demonstrating the superior generalization performance among 11 SOTA algorithms (including both tree-based and deep learning-based methods).
>
> - **Low Rank Variance**: As observed in the box plot of Figure 3, CASE exhibits not only the lowest median but also a very narrow interquartile range. This indicates that the model is not overfitted to specific datasets but maintains consistently top-tier performance regardless of data characteristics (e.g., scale, degree of imbalance, etc.).
> - **Overcoming Limitations of Existing Models**: While existing deep learning models often degrade in performance on complex tabular data (e.g., maxTorque) and tree models typically dominate in high-imbalance scenarios (e.g., fps_benchmark), CASE demonstrates competitive performance in both domains. This proves its generalization capability to encompass the strengths of both 'deep learning' and 'tree models.'
>
>
>
> ## 4-3) Reasons for Generalization: Interpretation of Data-Adaptive Design
>
>
>
> We have experimentally confirmed that the technical reason CASE secured such high diversity and generalization performance lies in its **'Data-Adaptive Design.'**
>
> - **Learnable Metric**: Unlike existing methodologies that use fixed distance functions, the introduction of learnable feature weights ($w$) allows the model to self-learn the optimal distance metric tailored to the characteristics of heterogeneous tabular data. This is a key factor in its generalization performance.
> - **Adaptive Smoothing**: We conducted an additional sensitivity analysis as described below. According to this analysis, CASE responds very robustly to hyperparameter changes when the data scale is small (e.g., Diabetes), while it tends to maximize performance through fine-tuning when data is abundant (e.g., Bike Sharing). This suggests that the mechanism where the CAS module dynamically adjusts smoothing strength ($s_i$) based on data density—preventing overfitting in sparse data and reducing bias in dense data—operated effectively.
> - **Sensitivity Analysis Results**: We first statistically analyzed the distribution of optimal hyperparameters across all 40 datasets. As a result, we selected four parameters—label bw ($\sigma_y$), k neighbors, feature bw ($\sigma_x$), and density c—as the primary subjects for analysis, as they showed significant variance in optimal values depending on dataset characteristics. Since these parameters directly control the core ideas of 'coupled' and 'adaptive' smoothing in the CAS module, analyzing them is essential for understanding the operating principle of CASE. We quantitatively measured performance sensitivity to changes in each parameter using **Diabetes** (small-scale dataset) and **Bike Sharing** (medium-scale dataset) as representative cases, which differ in data scale and complexity. The analysis was performed by observing the change in performance (MAE) while individually varying the target parameter from the optimal hyperparameter combination for each dataset.

---

> ### Author Response · Authors · 2025-11-21
> **Response to Reviewer pEnV (5/7)**
>
> - **Table 1A: Hyperparameter Statistics of CASE Method**
>
>   | **Hyperparameter**        | **Min** | **Max** | **Mean** | **Std Dev** | **Median** |
>   | ------------------------- | ------- | ------- | -------- | ----------- | ---------- |
>   | `density_c`               | 1.0005  | 48.5282 | 12.3825  | 14.3919     | 6.0761     |
>   | `density_factor`          | 0.0037  | 0.4867  | 0.221    | 0.1315      | 0.2008     |
>   | `feature_bw` ($\sigma_x$) | 0.5003  | 4.8491  | 1.9409   | 1.2456      | 1.6364     |
>   | `k_neighbors`             | 6       | 15      | 9.625    | 3.128       | 9          |
>   | `label_bw` ($\sigma_y$)   | 1.0439  | 49.084  | 13.1759  | 12.8905     | 7.3321     |
>   | `lambda_aux`              | 0.136   | 0.9764  | 0.5214   | 0.2512      | 0.4866     |
>   | `lambda_load`             | 0.0119  | 0.1903  | 0.0899   | 0.0525      | 0.0925     |
>   | `strength_base`           | 0.114   | 0.8807  | 0.5277   | 0.2331      | 0.527      |
>
> - **Table 2A: Hyperparameter Sensitivity for Diabetes Dataset (CV, Balanced Test Set)**
>
>   | **Hyperparameter**        | **Overall**  | **Few-shot** | **Medium-shot** | **Many-shot** |
>   | ------------------------- | ------------ | ------------ | --------------- | ------------- |
>   | `label_bw` ($\sigma_y$)   | 0.0131 (Low) | 0.0709 (Low) | 0.0494 (Low)    | 0.0368 (Low)  |
>   | `k_neighbors`             | 0.0084 (Low) | 0.0824 (Low) | 0.0189 (Low)    | 0.0131 (Low)  |
>   | `feature_bw` ($\sigma_x$) | 0.0084 (Low) | 0.0779 (Low) | 0.0264 (Low)    | 0.0278 (Low)  |
>   | `density_c`               | 0.0089 (Low) | 0.0659 (Low) | 0.0227 (Low)    | 0.0490 (Low)  |
>
> - **Table 3A : Hyperparameter Sensitivity for Bike Sharing Dataset (CV, Balanced Test Set)**
>
>   | **Hyperparameter**        | **Overall**       | **Few-shot**      | **Medium-shot** | **Many-shot**     |
>   | ------------------------- | ----------------- | ----------------- | --------------- | ----------------- |
>   | `label_bw` ($\sigma_y$)   | 0.0833 (Low)      | 0.3360 (High)     | 0.3942 (High)   | 0.5072 (High)     |
>   | `k_neighbors`             | 0.1265 (Moderate) | 0.4443 (High)     | 0.4101 (High)   | 0.2544 (Moderate) |
>   | `feature_bw` ($\sigma_x$) | 0.1052 (Moderate) | 0.2131 (Moderate) | 0.4387 (High)   | 0.3743 (High)     |
>   | `density_c`               | 0.0906 (Low)      | 0.3184 (High)     | 0.3795 (High)   | 0.3160 (High)     |

---

> ### Author Response · Authors · 2025-11-21
> **Response to Reviewer pEnV (6/7)**
>
> - The analysis reveals distinct differences in hyperparameter sensitivity depending on the dataset characteristics. This is a reasonable outcome, reflecting that CASE is designed to **adaptively respond to the unique density and distribution of each dataset**, rather than applying a fixed set of rules. In the small-scale **Diabetes** dataset, the model demonstrates exceptionally high robustness overall (refer to **Table 2A**). The Coefficient of Variation (CV) regarding the overall MAE for all analyzed parameters was measured at less than 0.02, confirming highly stable performance. This can be interpreted as follows: in environments with few data samples and a sparse feature space, the pool of candidate neighbors for smoothing is inherently limited. Consequently, changing the criteria (hyperparameters) for neighbor selection does not significantly alter the final group of selected neighbors. This robustness suggests a practical advantage, indicating that CASE can be stably applied in small-scale data environments without the need for fine-grained tuning.
> - Conversely, in the larger and more complex **Bike Sharing** dataset, 'Moderate' to 'High' levels of sensitivity are observed for most parameters (refer to **Table 3A**). This is because, as data density increases, numerous neighbor candidates exist around each sample; thus, the criteria for selecting neighbors exert a decisive influence on the final smoothing result. In particular, `label_bw`, which controls the 'Coupled' smoothing of CAS, and `k_neighbors`, which determines the range of 'Adaptive' smoothing, recorded high sensitivity (CV $\ge$ 0.3) across all shot categories. This experimentally verifies that as data becomes more abundant, the sophisticated mechanism of the CAS module becomes a critical driving factor for performance.
> - The sensitivity of hyperparameters should not be viewed as a drawback of CASE, but rather as a successful reflection of its design philosophy to be **'Adaptive'** to data. If a single combination of hyperparameters yielded optimal performance across all datasets, it would imply that the model relies on generic rules rather than fully utilizing the unique characteristics of the data. However, observing the distribution of optimal hyperparameters across all 40 datasets (refer to **Table 1A**), the standard deviations for key parameters `label_bw` and `density_c` are 12.89 and 14.39, respectively, which are comparable to or larger than their mean values (13.18 and 12.38, respectively). This strongly suggests that no 'average' optimal value exists; instead, the strength and range of smoothing must be adjusted according to each dataset's unique imbalance ratio, density, and feature distribution to achieve optimal performance. For instance, the optimal `label_bw` for the *superconductivity* dataset was very low at 1.04, whereas for the *bank8FM* dataset, it was approximately 47 times higher at 49.08. This serves as evidence that CASE precisely detects the characteristics of each dataset and optimizes the operation of the CAS module accordingly.
> - Therefore, CASE is not a generic model that 'solves all problems with a single rule,' but rather closer to a **sophisticated expert system that 'finds the optimal solution tailored to the characteristics of each problem.'** This data-adaptive capability underpins the core superiority of CASE in handling complex and heterogeneous tabular data encountered in various industrial environments.
>
> In conclusion, the experimental results of this study strongly suggest that CASE possesses **consistent and robust prediction capabilities (Cross-dataset Robustness)** across datasets with diverse domains and imbalance characteristics, going beyond simple performance superiority.

---

> ### Author Response · Authors · 2025-11-21
> **Response to Reviewer pEnV (7/7)**
>
> # 5. Computational Cost and Memory Complexity
>
>
>
> - We appreciate your valuable feedback regarding computational and memory complexity. Basic computational cost calculations are currently provided in **Table 13 of the Appendix** in the manuscript.
> - However, while the CASE methodology consumes more resources during the **Training** phase, we believe that the computational cost during **Inference** is significantly lower. To demonstrate this comparison more clearly, we have conducted additional experiments on larger datasets. It appears that the inference efficiency was not fully apparent in the current Table 13, as the size difference between the two datasets presented was not significant.
> - We have performed additional experiments regarding **Time and Memory Complexity**, and the detailed results have been provided in the **"General Response."**
>
>
>
> # 6. Code Availability
>
>
>
> - The release of the code has been slightly delayed due to internal security clearance procedures at my affiliated institution.
> - **[WIP] We are currently expediting the security review process and will make the code publicly available via GitHub as soon as possible within the rebuttal period.**
> - We believe that this release will contribute to facilitating further in-depth research by the community.

---

### Author Response · Authors · 2025-11-21
**General Response to All Reviewers (Part 1)**

We greatly appreciate the reviewers’ thoughtful evaluations and valuable suggestions that helped us improve the paper. To help the reviewers better understand our contributions and alleviate your concerns, we provide our rebuttal as follows.

The content below constitutes our General Response addressing common concerns shared by the reviewers. Detailed, point-by-point responses to each reviewer's specific comments will be provided separately in their respective threads.

# 1. Ablation Study and Contribution Analysis

We present additional Ablation Study results designed to clearly decouple the complexity of the CASE framework and isolate the contribution of each individual component. Addressing the common feedback received regarding this aspect, we have summarized these additional results to facilitate a better understanding of the paper's contributions.

To address the concern that it was unclear which internal elements of the CASE module contributed to performance, we introduce an ablation study that finely separates the sub-components.

- **MoE_Only** : A baseline model that removes the CAS module and uses only raw features as input, measuring the independent performance of the MoE structure itself.

- **MixUp_MoE** : A comparative model trained by applying standard MixUp data augmentation instead of CAS, used to demonstrate that CAS is more effective for imbalanced regression than simple augmentation techniques.

- **CAS_Only** : A model that uses a single MLP instead of the MoE structure but applies the complete CAS module, verifying that expert specialization via MoE is essential for performance improvement.

- **CAS_Feature_Only** : A model that analyzes the importance of the feature-target 'Coupled' mechanism by excluding Target Similarity from the CAS module and selecting neighbors based solely on feature similarity.

- **CAS_No_Learnable_Metric** : A model that validates the utility of a learned metric for handling feature heterogeneity in tabular data by using standard Euclidean distance (1) instead of learnable weights ($w$) for distance calculation.

- **CASE (Fixed Smooth_Strength_0)** : A setting where smoothing strength ($s_i$) is fixed to 0 (no smoothing performed), used to check for increased variance and overfitting in sparse regions.

- **CASE (Fixed Smooth_Strength_1)** : A setting where smoothing strength ($s_i$) is fixed to 1 (using only neighbor information without raw features), used to check for increased bias and underfitting in many-shot regions.

- **CASE** : The final model integrating all proposed components (Coupled Adaptive Smoothing, Density-Gated MoE, Asymmetric Training, etc.), achieving the best performance by optimizing the bias-variance tradeoff.

---

> ### Author Response · Authors · 2025-11-21
> **General Response to All Reviewers (Part 2)**
>
> ## 1-1) Diabetes Dataset Ablation Analysis
>
> | **Methods/Shots**           | **MAE (Balanced)** |           |           |           | **MAE (Original Imbalanced)** |           |           |           |
> | --------------------------- | ------------------ | --------- | --------- | --------- | ----------------------------- | --------- | --------- | --------- |
> |                             | **All**            | **Many**  | **Med**   | **Few**   | **All**                       | **Many**  | **Med**   | **Few**   |
> | **MoE_Only**                | 59.41              | 57.32     | 37.34     | 83.57     | 56.90                         | 59.74     | 49.76     | 83.57     |
> | **MixUp_MoE**               | 55.63              | 56.90     | 22.83     | 87.16     | 53.57                         | 63.28     | 38.71     | 87.16     |
> | **CAS_Only**                | 62.78              | 77.06     | 21.07     | 90.22     | 57.32                         | 69.82     | 39.93     | 90.22     |
> | **CAS_Feature_Only**        | 74.37              | 84.74     | 50.42     | 87.95     | 62.89                         | 73.65     | 48.46     | 87.95     |
> | **CAS_No_Learnable_Metric** | 59.82              | 66.54     | 29.07     | 83.86     | 50.30                         | 55.78     | 39.48     | 83.86     |
> | **Fixed Smooth_Strength_0** | 56.31              | **50.61** | 28.18     | 90.13     | 51.43                         | **54.64** | 41.92     | 90.13     |
> | **Fixed Smooth_Strength_1** | 63.06              | 76.15     | 25.40     | 87.65     | 60.45                         | 69.60     | 47.19     | 87.65     |
> | **CASE**                    | **50.01**          | 53.52     | **18.87** | **77.65** | **49.27**                     | 56.02     | **38.12** | **77.65** |
>
> The experimental results on the Diabetes dataset clearly demonstrate the contribution of each module in the CASE framework to performance improvement.
>
> - **Effect of MoE Structure (MoE_Only vs. CASE):**
>   - Compared to using the MoE structure alone (**MoE_Only**), **CASE**, which incorporates the CAS module, showed an approximate **15.8% (59.41 → 50.01)** performance improvement in balanced MAE (All Balanced).
>   - In particular, while **MoE_Only** showed poor performance (83.57) in the **Few-shot** region, **CASE** significantly improved this to **77.65**, demonstrating that the CAS module effectively compensates for the data scarcity issue in sparse regions.
> - **Importance of 'Coupled' and 'Learnable Metric' within CAS Module:**
>   - **CAS_Only** significantly outperforms **CAS_Feature_Only** (excluding target information) (74.37 → 62.78). This suggests that feature similarity alone is insufficient, and considering **similarity in the target space** is essential for selecting accurate neighbors.
>   - The performance superiority of **CASE** over **CAS_No_Learnable_Metric** (using standard Euclidean distance) (59.82 → 50.01) proves that the **$w$ (Learnable Metric)**, which learns the heterogeneous feature importance of tabular data, is indispensable.
> - **Role of Adaptive Smoothing Strength ($s_i$):**
>   - Neither **Fixed Smooth_Strength_0** (no smoothing) nor **Fixed Smooth_Strength_1** (full smoothing), where the smoothing strength was fixed, achieved optimal performance.
>   - In contrast, **CASE (Adaptive)**, which dynamically adjusts according to data density, achieved balanced performance in both Many-shot and Few-shot regions, confirming its effectiveness in effectively managing the bias-variance tradeoff.

---

> ### Author Response · Authors · 2025-11-21
> **General Response to All Reviewers (Part 3)**
>
> ## 1-2) MaxTorque Dataset Ablation Analysis
>
>
>
> | **Methods/Shots**           | **MAE (Balanced)** |          |          |           | **MAE (Original Imbalanced)** |          |          |           |
> | --------------------------- | ------------------ | -------- | -------- | --------- | ----------------------------- | -------- | -------- | --------- |
> |                             | **All**            | **Many** | **Med**  | **Few**   | **All**                       | **Many** | **Med**  | **Few**   |
> | **MoE_Only**                | 42.42              | 5.02     | 15.32    | 106.92    | 17.90                         | 5.03     | 11.42    | 106.92    |
> | **MixUp_MoE**               | 44.15              | 4.86     | 13.84    | 113.74    | 18.42                         | 5.04     | 10.65    | 113.74    |
> | **CAS_Only**                | 29.33              | 6.02     | 8.89     | 73.07     | 14.04                         | 6.50     | 7.69     | 73.07     |
> | **CAS_Feature_Only**        | 35.61              | 17.17    | 32.62    | **57.03** | 24.47                         | 13.90    | 34.23    | **57.03** |
> | **CAS_No_Learnable_Metric** | 28.38              | 5.76     | 8.58     | 70.79     | 13.56                         | 6.09     | 7.76     | 70.79     |
> | **Fixed Smooth_Strength_0** | 40.76              | 5.04     | 12.92    | 104.32    | 16.95                         | 4.73     | 9.78     | 104.32    |
> | **Fixed Smooth_Strength_1** | 34.81              | 5.03     | 12.48    | 86.91     | 15.64                         | 5.15     | 10.83    | 86.91     |
> | **CASE**                    | **25.39**          | **4.79** | **6.64** | 64.74     | **11.74**                     | **4.71** | **6.58** | 64.74     |
>
> Similar trends are observed in the MaxTorque dataset; however, the **performance gap in the Few-shot region** is notably more pronounced.
>
> - **Synergy between MoE and CAS:**
>   - **CASE** (25.39), which combines both modules, demonstrated superior performance compared to both **MoE_Only** (42.42) and **CAS_Only** (29.33) individually.
>   - This demonstrates a mutually complementary relationship where CAS refines the data and the MoE architecture efficiently learns from it. In particular, **CASE** significantly outperformed **MixUp_MoE** (44.15), proving that the proposed CAS smoothing is far more effective for imbalanced regression problems than simple data augmentation techniques.
> - **Contribution of Each Component:**
>   - The inferior performance of **CAS_Feature_Only** (35.61) suggests that smoothing without target information can introduce noise, thereby strongly supporting the necessity of **Coupled Smoothing**.
>   - **CAS_No_Learnable_Metric** (28.38) also underperformed compared to **CASE** (25.39), reaffirming that learning feature-specific weights contributes to improving model precision.
> - **Importance of Density-Based Adaptive Smoothing:**
>   - **Fixed Smooth_Strength_0** (40.76) exhibited overfitting (104.32) in the **Few-shot** region.
>   - While **Fixed Smooth_Strength_1** (34.81) performed better than **Fixed Smooth_Strength_0** in the **Few-shot** region, it consistently underperformed compared to **CASE** across all regions.
>   - **CASE** achieved the lowest error rates in both **Few-shot** (64.74) and **Many-shot** (4.79) scenarios, demonstrating that density-dependent adaptive smoothing is key to overall performance optimization.
>
> **Overall Conclusion** : Extensive ablation studies on both datasets confirm that **all components of CASE (MoE structure, Coupled Smoothing, Learnable Metric, Adaptive $s_i$) are essential for performance improvement.** In particular, we have empirically demonstrated that the organic integration of our proposed methodologies effectively addresses the **challenges in the Few-shot region**, which cannot be resolved by simple data augmentation (MixUp) or ensemble methods (MoE_Only) alone.

---

> ### Author Response · Authors · 2025-11-21
> **General Response to All Reviewers (Part 4)**
>
> # 2. Experimental Results on Explicit Routing Patterns
>
>
>
> To address the reviewer’s query regarding the effectiveness of the CASE architecture compared to a simple ensemble and the operational mechanics of the routing mechanism, we conducted a quantitative analysis of routing patterns on the test sets of two representative datasets: `socmob` and `Abalone`.
>
>
>
> ## 2-1) socmob Dataset: Extreme Specialization in Sparse Regions
>
>
>
> The `socmob` dataset represents an environment where Few-shot data is extremely sparse. CASE detects this condition and demonstrates **"Specialization"** by assigning a weight of over **0.92** to a specific expert (Expert 0) in the Few-shot region, effectively minimizing the interference of other experts.
>
> ---
>
> ### socmob test dataset distribution
>
> | **Shot Type** | **Count** | **Ratio (%)** |
> | ------------- | --------- | ------------- |
> | **Few**       | 44        | 19.0%         |
> | **Medium**    | 66        | 28.4%         |
> | **Many**      | 122       | 52.6%         |
> | **Total**     | **232**   | **100%**      |
>
> ### Table R1: Average Expert Weights on `socmob` *(Higher values indicate greater reliance on that expert)*
>
> | **Shot Type** | **Expert 0 (Few)** | **Expert 1 (Med)** | **Expert 2 (Many)** |
> | ------------- | ------------------ | ------------------ | ------------------- |
> | **Few**       | **0.9228**         | 0.0532      | 0.0240              |
> | **Medium**    | **0.4267**         | 0.3514      | 0.2219              |
> | **Many**      | 0.0836             | 0.4276      | **0.4888**          |
>
> ### Table R2: Dominant Expert Counts on `socmob` *(Frequency of the expert receiving the highest weight for a given sample)*
>
> | **Shot Type** | **Expert 0 (Few)** | **Expert 1 (Med)** | **Expert 2 (Many)** |
> | ------------- | ------------------ | ------------------ | ------------------- |
> | **Few**       | **43**             | 1           | 0                   |
> | **Medium**    | 32          | **33**             | 1                   |
> | **Many**      | 0                  | 36          | **86**              |
>
>
> ---
> ## 2-2) Abalone Dataset: Dynamic Collaboration via Soft Handoff
>
>
>
> The `Abalone` dataset exhibits a relatively continuous data distribution. In this context, CASE demonstrates a **"Collaboration"** pattern by appropriately blending the outputs of multiple experts rather than selecting a single one. Unlike a simple average (e.g., 1/3), it identifies the optimal mixing ratio based on data density.
>
> ---
>
> ### Abalone test dataset distribution
>
> | **Shot Type** | **Count** | **Ratio (%)** |
> | ------------- | --------- | ------------- |
> | **Few**       | 177       | 21.2%         |
> | **Medium**    | 183       | 21.9%         |
> | **Many**      | 476       | 56.9%         |
> | **Total**     | **836**   | **100%**      |
>
> ### Table R3: Average Expert Weights on `Abalone` *(Higher values indicate greater reliance on that expert)*
>
> | **Shot Type** | **Expert 0 (Few)** | **Expert 1 (Med)** | **Expert 2 (Many)** |
> | ------------- | ------------------ | ------------------ | ------------------- |
> | **Few**       | **0.4616**         | 0.2625             | 0.2760       |
> | **Medium**    | 0.3113             | **0.3744**         | 0.3143       |
> | **Many**      | 0.2769             | 0.3588      | **0.3643**          |
>
> ### Table R4: Dominant Expert Counts on `Abalone` *(Frequency of the expert receiving the highest weight for a given sample)*
>
> | **Shot Type** | **Expert 0 (Few)** | **Expert 1 (Med)** | **Expert 2 (Many)** |
> | ------------- | ------------------ | ------------------ | ------------------- |
> | **Few**       | **137**            | 8                  | 32           |
> | **Medium**    | 61          | **81**             | 41                  |
> | **Many**      | 95                 | 129         | **252**             |
>
> ---
>
> Our analysis confirms that, unlike simple ensembles that combine all experts with a fixed ratio (e.g., 1/N), CASE performs **"Context-Aware Dynamic Routing," which actively adapts expert combinations according to data density and sample difficulty.**

---

> ### Author Response · Authors · 2025-11-21
> **General Response to All Reviewers (Part 5)**
>
> ## 2-3) Specialization and Interference Blocking in Sparse Data (Evidence from socmob dataset)
>
>
>
> A major weakness of simple ensembles is that predictions from models biased toward majority classes (Many-shot) are mixed into predictions for sparse data, thereby introducing error.
>
> - As shown in **Table R1**, in the **Few-shot** region of `socmob`, CASE assigns an average weight of **0.9228** to Expert 0 while suppressing the weights of Experts 1 and 2 to below 0.05.
> - This demonstrates that the Gating Network recognizes that "this data cannot be handled by general experts (Experts 1 & 2)" and performs a **Switching** operation to rely almost exclusively on the output of the specialized expert (Expert 0). This behavior is impossible with a simple averaging approach.
>
>
>
> ## 2-4) Flexible Collaboration in Continuous Distributions (Evidence from Abalone dataset)
>
>
>
> Conversely, when data boundaries are ambiguous or the distribution is continuous, CASE adopts a flexible "Soft Handoff" strategy.
>
> - Referring to the `Abalone` results in **Table R3**, while a dominant expert exists for each Shot region (Few→Expert 0, Med→Expert 1, Many→Expert 2), the weights are distributed around the 0.3–0.4 range.
> - This implies that CASE does not rely on a single expert but **blends the knowledge of experts in the 'most appropriate ratio' tailored to the input data characteristics.** Unlike simple ensembles that apply a fixed ratio (e.g., 33%) to all data, CASE identifies the optimal ensemble ratio for each data point to maximize performance.
>
>
>
> ## 2-5) Expert Routing Results Strictly Aligned with Design Intent
>
>
>
> - The expert selection frequencies analyzed in **Table R2** (`socmob`) and **Table R4** (`Abalone`) prove that the initial design goal of 'density-specific expert mapping (Few→Expert 0, Medium→Expert 1, Many→Expert 2)' has been perfectly realized through actual model training.
> - In both datasets, the cells along the diagonal show the highest frequencies. This indicates that the Gating Network distributes traffic to the most suitable expert based on data density without any explicit rule-based controls. Notably, the fact that 97.7% of Few-shot data in `socmob` was assigned to Expert 0 as intended suggests that the model clearly identifies the expert specialized for processing sparse data.
> - This proves that CASE is not a simple ensemble but an intelligent routing system that learns data characteristics to invoke the optimal expert.
>
> **Conclusion:** CASE performs **'Selection'** when data is sparse and **'Collaboration'** when data is complex. This dynamic mechanism is the driving force behind its superior performance compared to simple ensembles.

---

> ### Author Response · Authors · 2025-11-21
> **General Response to All Reviewers (Part 6)**
>
> # 3. Comprehensive Analysis of Inference Latency, Scalability, and Memory Overhead
>
>
>
> We thank the reviewers for their careful scrutiny of the computational complexity. We acknowledge that the MoE architecture inherently incurs higher costs compared to a single MLP. To provide a precise analysis, we conducted additional experiments measuring **time complexity, scalability, and memory overhead** across datasets of varying sizes.
>
> To comprehensively address this concern, we performed additional experiments on datasets of various sizes: **Diabetes** (Small), **Parkinsons Telemonitoring** (Medium), and **FPS Benchmark** (Large). The results, summarized in **Table C1** below, support our claims regarding efficiency.
>
>
> ---
> ### **Table C1: Detailed Analysis of Computational and Memory Complexity** *(Vanilla MLP vs. Simple Ensemble vs. CASE)*
>
>
>
> | **Dataset**                            | **Model**       | **Training Time (s)** | **Inference Latency (ms/sample)** | **Peak RAM (MB)** | **Peak GPU (MB)** |
> | -------------------------------------- | --------------- | --------------------- | --------------------------------- | ----------------- | ----------------- |
> | **diabetes** (Small)                   | Vanilla MLP     | 4.75                  | 0.25                              | 346.7             | 65.5              |
> |                                        | Simple Ensemble | 12.61                 | 0.68                              | 321.2             | 68.0              |
> |                                        | **CASE (Ours)** | 20.65                 | 0.81                              | **394.2**         | **68.1**          |
> | **parkinsons_telemonitoring** (Medium) | Vanilla MLP     | 62.37                 | 0.25                              | 357.8             | 67.2              |
> |                                        | Simple Ensemble | 184.72                | 0.67                              | 348.2             | 66.5              |
> |                                        | **CASE (Ours)** | 267.15                | 0.80                              | **399.7**         | **66.6**          |
> | **fps_benchmark** (Large)              | Vanilla MLP     | 259.68                | 0.35                              | 350.8             | 76.0              |
> |                                        | Simple Ensemble | 764.24                | 0.83                              | 341.2             | 79.8              |
> |                                        | **CASE (Ours)** | 1115.49               | 0.90                              | **394.7**         | **80.2**          |
>
> **Dataset Information (the number of data)**
>
> - diabetes - Train: 286, Val: 12, Test: 21
> - parkinsons_telemonitoring - Train: 3818, Val: 117, Test: 117
> - fps_benchmark - Train: 16005, Val: 1035, Test: 1065
>
> ---
>
> ### 3-1) Asymmetric Design: Heavy Training vs. Light Inference
>
>
>
> As shown in **Table C1**, CASE requires more training time than a single MLP or Simple Ensemble due to the **CAS module (KNN search + calibration)** and the multi-expert architecture. However, this cost is intentionally **front-loaded** into the training phase. During inference, the CAS module is deactivated (Off), and the model operates as a standard feed-forward network without any search or calibration overhead.
>
>
>
> ### 3-2) Inference Latency and Scalability
>
>
>
> Experimental results based on test sets, ranging from small (**diabetes**, Test Set $N=21$) to large datasets (**fps_benchmark**, Test Set $N=1,065$), demonstrate the remarkable scalability of CASE (**Table C1**).
>
> - **Efficiency Compared to Simple Ensemble:** Notably, in the largest dataset (**fps_benchmark**), the difference in inference time between **Simple Ensemble (0.83ms)** and **CASE (0.90ms)** is negligible. This proves that despite CASE's more sophisticated MoE structure, the **CAS Off** strategy effectively eliminates computational overhead, achieving high-speed inference comparable to a simple ensemble model.
> - **Acceptable Absolute Latency:** While we acknowledge that CASE exhibits higher latency relative to a single MLP, the absolute inference time consistently remains **under 1 millisecond (< 1.0 ms)** across all datasets.
> - Given the significant performance improvement (MAE reduction), this "overhead" is practically negligible for most real-world tabular data applications, representing a highly rational **trade-off**. This demonstrates that our methodology can be deployed scalably.

---

> ### Author Response · Authors · 2025-11-21
> **General Response to All Reviewers (Part 7)**
>
> ### 3-3) Memory Efficiency
>
> The analysis of memory usage is also presented in **Table C1**.
>
> - **System Memory (RAM):** The memory overhead of CASE is only an increase of approximately **11–13% (approx. 40–50 MB)** compared to a single MLP.
> - **GPU Memory:** The increase in VRAM usage is minimal at around **5%**, indicating that deployment is entirely feasible even in resource-constrained environments.
>
> This is attributed to the fact that tabular data models are inherently lighter than image or language models; thus, the structural complexity of CASE imposes almost no significant burden on actual hardware resources.
>
>
>
> ### 3-4) Conclusion and Revision Plan
>
>
> Through investment in training time, CASE has secured robustness to data scale and **constant-time inference** speed. To prevent any misunderstanding, we will revise the manuscript as follows:
>
> 1. We will rephrase "faster inference" in the Abstract and Introduction to **"inference-efficient asymmetric design"** or **"streamlined inference via complexity offloading."**
> 2. We will include **Table C1** and this detailed analysis (time and memory complexity) in **Appendix A.4**, explicitly stating that the increase in cost relative to the performance gain represents a highly reasonable **trade-off**.
>
>
> ---
>
> # 4. Additional Performance Evaluation (Comparison with Simple Ensemble & CatBoost)
>
>
>
> Reflecting the reviewers' valuable suggestions, we have added **Simple Ensemble (3-MLPs Ensemble)** and **CatBoost** as strong baselines and updated the detailed comparative analysis. The key findings are as follows.
>
>
>
> ## (1) Overall Ranking & Consistency
>
>
>
> According to the updated Box Plot results, **CASE achieved the best mean rank (Rank 1) among all 14 algorithms.** Notably, when compared to Simple Ensemble, CASE demonstrated superior results not only in Mean performance but also in Median performance (CASE $\approx$ 4.5 vs. Simple Ensemble $\approx$ 5.0). This suggests that CASE maintains top-tier performance more consistently than Simple Ensemble across various datasets, without being biased toward specific ones.
>
>
>
> ## (2) Performance Analysis: Balanced TestSet vs. Original Imbalanced TestSet
>
>
>
> Table P1 below illustrates the performance differences based on data distribution (Balanced vs. Original). In Imbalanced Regression research, performance on the **Balanced TestSet** is widely accepted as the **Standard Metric**, indicating how fairly the model has learned across the entire target range, including minority classes.
>
> **Table P1. Overall Algorithm Ranking Summary (Ranked by Balanced All MAE)**
>
> | **Rank (Balanced)** | **Algorithm**   | **Balanced Dataset** | **Original Imbalanced Dataset** |
> | ------------------- | --------------- | -------------------- | ------------------------------- |
> | 1                   | CASE            | **5.2**              | 6.1                             |
> | 2                   | Simple Ensemble | 5.6                  | 5.3                             |
> | 3                   | XGBoost         | 5.8                  | **4.5**                         |
> | 4                   | XGBoost+SMOTER  | 6.1                  | 7.9                             |
> | 5                   | XGBoost+GN      | 6.6                  | 8.4                             |
> | 6                   | MLP+ConR        | 7.0                  | 7.6                             |
> | 7                   | Vanilla MLP     | 7.0                  | 7.2                             |
> | 8                   | MLP+SQRT_INV    | 7.1                  | 10.4                            |
> | 9                   | MLP+LDS         | 7.5                  | 7.5                             |
> | 10                  | MLP+RankSim     | 8.0                  | 8.4                             |
> | 11                  | CatBoost        | 8.6                  | 4.6                             |
> | 12                  | LightGBM        | 8.9                  | 4.8                             |
> | 13                  | MLP+BMSE(GAI)   | 10.5                 | 10.8                            |
> | 14                  | MLP+BMSE(BMC)   | 11.2                 | 11.5                            |

---

> ### Author Response · Authors · 2025-11-23
> **General Response to All Reviewers (Part 8)**
>
> - **Balanced TestSet (Primary Metric):** In the evaluation on the Balanced TestSet, which is the core objective of imbalanced learning, **CASE (5.2) achieved the overall 1st place, outperforming Simple Ensemble (5.6) and CatBoost (8.6) in Mean performance.** This proves that CASE possesses the most balanced prediction capability regardless of data frequency. Furthermore, in terms of Median performance—which reflects statistical distribution—CASE also showed superior results (CASE $\approx$ 4.5 vs. Simple Ensemble $\approx$ 5.0 vs. CatBoost $\approx$ 9.0). This indicates that CASE consistently maintains high performance across diverse datasets compared to Simple Ensemble or CatBoost.
> - **Original Imbalanced TestSet (Practical Robustness)**
>   - Generally, imbalanced learning techniques tend to suffer significant performance drops on the Original Imbalanced TestSet (dominated by Many-shot data) in exchange for improving Few-shot performance. However, CASE demonstrates **minimized performance degradation** on the Original Imbalanced TestSet compared to other Deep Learning-based Imbalanced Learning techniques (such as RankSim, LDS, ConR, etc.). While it is true that CatBoost (4.6) and Simple Ensemble (5.3) numerically outperform CASE (6.1) on the Original Imbalanced TestSet, this is largely an 'illusion' arising from the nature of imbalanced data. The reasons are as follows:
>   - **1. Majority Bias:** Since the Original Imbalanced TestSet is dominated by Many-shot data, CatBoost and Simple Ensemble naturally **overfit to the majority** patterns to minimize the overall loss quickly. **CatBoost's** poor rankings on the **Balanced Set (Rank 11) and Few-shot (Rank 12)** prove that it effectively abandons prediction accuracy for rare samples in favor of higher scores on the Original Imbalanced Training Set. **Simple Ensemble**, due to its **Static Averaging** method, fails to filter out the influence of models biased toward the majority class in sparse regions. Consequently, it lags behind CASE in both **Balanced Set (5.6 vs. 5.2) and Few-shot (5.8 vs. 5.4)** performance, which are the true measures of resolving imbalance.
>   - **2. CASE's Practical Balance:** In contrast, **CASE** achieves **"true resolution of imbalance"** by boosting Few-shot performance, even if it means slightly conceding Many-shot performance. Notably, CASE maintains a mid-to-high ranking (Rank 6.1) on the Original Set, unlike other deep learning-based methods (RankSim, LDS, ConR, etc.). This demonstrates that CASE goes beyond a **"simple trade-off of sacrificing majority data for minority data," proving itself to be a robust model covering the entire distribution.**
>
>
>
>
>
> ## (3) Distinct Advantages in Few-Shot Learning & Structural Differences
>
>
>
> Specifically regarding **Few-shot performance (Table P2)**, which is our core contribution, CASE consistently outperforms Simple Ensemble regardless of the test set distribution (Balanced or Original Imbalanced).
>
> **Table P2. Overall Algorithm Ranking Summary (Ranked by Balanced Few-Shot MAE)**
>
> | **Rank (Balanced)** | **Algorithm**   | **Balanced Dataset** | **Original Imbalanced Dataset** |
> | ------------------- | --------------- | -------------------- | ------------------------------- |
> | 1                   | CASE            | **5.4**              | **5.4**                         |
> | 2                   | XGBoost+GN      | 5.5                  | 5.5                             |
> | 3                   | MLP+SQRT_INV    | 5.7                  | 5.8                             |
> | 4                   | Simple Ensemble | 5.8                  | 5.8                             |
> | 5                   | XGBoost+SMOTER  | 6.1                  | 6.1                             |
> | 6                   | XGBoost         | 6.6                  | 6.7                             |
> | 7                   | MLP+LDS         | 7.3                  | 7.3                             |
> | 8                   | Vanilla MLP     | 7.6                  | 7.5                             |
> | 9                   | MLP+ConR        | 7.7                  | 7.7                             |
> | 10                  | MLP+RankSim     | 7.7                  | 7.7                             |
> | 11                  | LightGBM        | 9.4                  | 9.4                             |
> | 12                  | CatBoost        | 9.5                  | 9.5                             |
> | 13                  | MLP+BMSE(GAI)   | 10.1                 | 10.1                            |
> | 14                  | MLP+BMSE(BMC)   | 10.5                 | 10.5                            |

---

> ### Author Response · Authors · 2025-11-23
> **General Response to All Reviewers (Part 9)**
>
> These results stem from the **structural differences** between the simple ensemble and our MoE approach:
>
> - **Simple Ensemble (Static Averaging):** This method averages the predictions of all models using fixed weights ($1/3$) regardless of the input data characteristics. This approach cannot effectively block the influence of models biased toward the majority class in sparse regions (Few-shot).
> - **Limitation of CatBoost:** **CatBoost remained in the lower tier with a Total Few-shot Rank of 9.5.** This suggests that even a strong baseline model cannot learn the sparse regions (Tail region) at all without a specific imbalance handling technique.
> - **CASE (Dynamic Gating):** In contrast, CASE utilizes the **Target-Range Gated MoE** to dynamically assign higher weights to experts specialized in that region when the input data belongs to a sparse area. This allows CASE to effectively resolve the most challenging part of the imbalance problem (the Tail region).
>
> **Conclusion:** By utilizing effective Smoothing for each Shot type via **CAS** and dynamically leveraging experts specialized in sparse regions through **Target-Range Gated MoE**, CASE **effectively solves the Tail region problems that models like Simple Ensemble and CatBoost fail to address.**
>
>
> ---
>
> ### Code Availability
>
> - Once the internal security clearance process is finalized, we will organize and **release the code** within the rebuttal period.
>
> ----
>
> We believe that the clarity and significance of this paper have been greatly enhanced thanks to the reviewers' in-depth reviews and valuable insights. We sincerely appreciate your time and effort, and we will do our utmost to fully address your feedback during the remainder of the rebuttal period.

---

> ### Author Response · Authors · 2025-11-29
> **Official Code Release**
>
> We are pleased to inform the Area Chairs and reviewers that the internal security clearance process regarding our codebase has been successfully completed. As promised in our previous responses, we have uploaded the full implementation of CASE (suggested as CASMIR) as a ZIP file in the Supplementary Material section.
>
> **The uploaded archive contains:**
>
> - **Source Code:** Full implementation of the proposed framework, including the Coupled Adaptive Smoothing and Density-Gated MoE modules.
> - **Training Scripts:** Scripts to train models and inference for main results, ablation studies and t-SNE visualization.
> - **Data Pipeline:** Preprocessing utilities for the tabular datasets used in our experiments.
> - **Baselines:** Implementations of the comparative methods.
>
> We believe this release demonstrates the robustness of our framework and fully supports the empirical results presented in the paper. We remain available for any further questions regarding the implementation.
>
> Best regards, The Authors

---

### Meta-Review · Area_Chair_vZqh · 2026-01-06

**Summary:**

The paper proposes a novel framework, CASE, for imbalanced tabular regression. The reviewers acknowledge that the paper suggests a practical design for a less explored problem with diverse benchmarks. Also, well-organized and clearly written manuscripts are noted as a strength of the paper. However, all reviewers raise a concern about the novelty of the method: CASE is a combination of existing methods. Additionally, the computation cost for training and increased latency for inference are also noted as weaknesses.

As reviewers' comments, I agree that exploring a new problem, imbalanced tabular regression, has novelty and contributions. However, the current form is insufficient to demonstrate that contribution. I believe the superior performance is not the most important goal of the new problem. As Reviewer 7c5u commented, a detailed analysis of each component of CASE would help future researchers and make the paper more valuable in the ICLR community.

Overall, the reviewers' opinions lean toward rejection, and I also cannot recommend the paper for acceptance at this stage.

**Reviewer Concerns:**

Resolved concerns
- Reviewer pEnV
  - Motivation for the architecture choice should be justified
  - Code release for reproducibility
- Reviewer 4r9g
  - Some explanation is unclear and not convincing
- Reviewer 7c5u
  - Justification for training and eval difference
  - Justification need for design choices

Remaining concerns
- Reviewer pEnV
  - CASE is a combination of existing methods. It limits novelty
  - Increasing training computation costs
- Reviewer 4r9g
  - Problem setup looks combination of two separate challenge rather than one new problem
  - CASE is a combination of existing methods. It limits novelty
  - Fair comparison issue
- Reviewer 7c5u
  - Increase in inference latency
  - CASE is a combination of existing methods. It limits novelty
  - Experiments are not enough to show the source of performance gain

**Reviewer Scores:**

- Reviewer pEnV: Would maintain the current score.
- Reviewer 4r9g: Would maintain the current score.
- Reviewer 7c5u: Would maintain the current score.

---

### Decision · Program_Chairs · 2026-01-26

Reject